# Approximating Full Conformal Prediction for Neural Network Regression with Gauss-Newton Influence

**Dharmesh Tailor**[1]*, **Alvaro H.C. Correia**[2], **Eric Nalisnick**[3], **Christos Louizos**[2]
[1]University of Amsterdam  [2]Qualcomm AI Research[†]  [3]Johns Hopkins University

## Abstract

Uncertainty quantification is an important prerequisite for the deployment of deep learning models in safety-critical areas. Yet, this hinges on the uncertainty estimates being *useful* to the extent the prediction intervals are well-calibrated and sharp. In the absence of inherent uncertainty estimates (*e.g.*, pretrained models predicting only point estimates), popular approaches that operate post-hoc include Laplace's method and split conformal prediction (split-CP). However, Laplace's method can be miscalibrated when the model is misspecified and split-CP requires sample splitting, and thus comes at the expense of statistical efficiency. In this work, we construct prediction intervals for neural network regressors post-hoc without held-out data. This is achieved by approximating the *full* conformal prediction method (full-CP). Whilst full-CP nominally requires retraining the model for every test point and candidate label, we propose to train just once and locally perturb model parameters using Gauss-Newton influence to approximate the effect of retraining. Coupled with linearization of the network, we express the absolute residual nonconformity score as a piecewise linear function of the candidate label allowing for an efficient procedure that avoids the exhaustive search over the output space. On standard regression benchmarks and bounding box localization, we show the resulting prediction intervals are locally-adaptive and often tighter than those of split-CP.

## 1 Introduction

Despite impressive advancements in machine learning, most models, particularly neural networks, are still designed and trained to provide only point estimates, lacking the ability to rigorously quantify uncertainty in their predictions. This poses a significant challenge, as reliable decision-making depends on trustworthy uncertainty representation. This need has drawn increased attention to uncertainty quantification in machine learning research, especially as the use of these models becomes more widespread and permeate safety-sensitive fields such as healthcare and autonomous driving.

Conformal Prediction (CP) (Vovk et al., 2005) is a class of uncertainty quantification methods that represent uncertainty via prediction intervals. These intervals intuitively convey the degree of uncertainty—the larger the interval, the greater the uncertainty. What sets CP apart is a rigorous, *distribution-free coverage* guarantee: conformal prediction intervals contain the true label with a probability of at least $1 - \alpha$, where $\alpha$ is a user-defined miscoverage rate. In recent years, a variant known as split (or inductive) CP (Papadopoulos et al., 2002) has gained traction in the machine learning community due to its ease of use and low computational cost. Split-CP, as the name suggests, divides the available data into training and calibration sets, using the former to fit a model and the latter to construct prediction intervals. Yet, split-CP is statistically inefficient as it cannot leverage all available data for model fitting, which hinders overall model performance. In contrast, full (or transductive) CP uses all data for both training and calibration but incurs high computational costs. It requires retraining the model multiple times—once for each test point and possible label—which is prohibitively expensive for deep learning applications.

---

*Work done while at Qualcomm AI Research. [†]Qualcomm AI Research is an initiative of Qualcomm Technologies, Inc. and/or its subsidiaries.

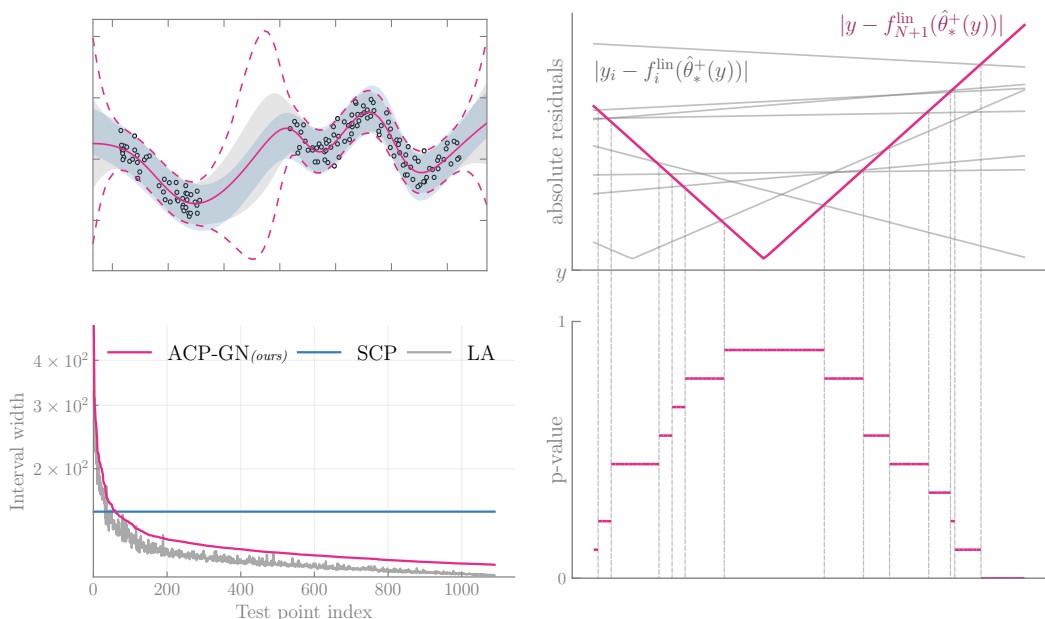

Figure 1: Our approx. full-CP via Gauss-Newton influence (ACP-GN) produces *adaptive* intervals (bottom left figure)—similar to Bayes via Laplace approximation (LA)—while satisfying coverage as seen in the high-overlap with split-CP (SCP) close to the data (white dots with black edge in top left figure). In the top right, we show the absolute residuals for the training points (in gray) and test point (in pink) as function of the postulated label for the test point $\mathbf{x}_{N+1}$. The conformal p-values, in the bottom right, only change when the pink line crosses a gray line, the so-called "changepoints", reducing the search space over candidate labels to incorporate in the prediction set.

In this work, we scale full-CP to neural network regression by (i) eliminating the need for retraining through a Gauss-Newton influence approximation, and (ii) avoiding an exhaustive search over the label space via network linearization. This gives a new, scalable full-CP method for neural network regression we dub *approximate full-CP via Gauss-Newton influence* (ACP-GN). As a second contribution, we show the same tools can be applied to enhance split-CP via normalization, yielding a new adaptable split-CP method we call SCP-GN. We validate ACP-GN and SCP-GN in several regression and bounding box prediction tasks, showing they satisfy coverage in most cases and produce tight, adaptable prediction intervals (see Fig. 1 for results with ACP-GN).

## 2 BACKGROUND ON CONFORMAL PREDICTION

Neural network regression follows the canonical empirical risk minimization framework. Given a dataset $\mathcal{D}_N := \{(\mathbf{x}_i, y_i)\}_{i=1}^N$ consisting of $N$ examples with inputs $\mathbf{x}_i \in \mathbb{R}^I$ and targets $y_i \in \mathbb{R}$, we minimize the (regularized) empirical risk:

$$\boldsymbol{\theta}_* = \arg\min_{\boldsymbol{\theta}} \left( \sum_{i=1}^N \ell(y_i, f_i(\boldsymbol{\theta})) + \frac{1}{2}\delta \|\boldsymbol{\theta}\|^2 \right) \tag{1}$$

where $f_i(\boldsymbol{\theta})$ is shorthand for $f(\mathbf{x}_i; \boldsymbol{\theta})$, the output of a deep neural network (DNN) at $\mathbf{x}_i$ with parameters $\boldsymbol{\theta} \in \mathbb{R}^D$, $\ell(y, f)$ is a loss function, and $\delta$ is an $L_2$ regularizer (*i.e.* weight decay). In this work, we restrict our attention to the squared-error loss: $\ell(y, f) = \frac{1}{2}(y - f)^2$. As is standard practice, the problem in Eq. (1) is approached using stochastic-gradient methods. For an unseen input $\mathbf{x}_{N+1}$, this gives us a point prediction $f_{N+1}(\boldsymbol{\theta}_*)$. However, in an ideal scenario, we would like a prediction *interval* whose width reflects the uncertainty associated with that input. This is where conformal prediction (Vovk et al., 2005) comes into play. It provides a framework for constructing prediction intervals while satisfying the following frequentist coverage guarantee known as *marginal coverage*

$$\mathcal{P}(y_{N+1} \in C_\alpha(\mathbf{x}_{N+1})) \geq 1 - \alpha, \tag{2}$$

where $y_{N+1}$ is the unseen target, $C_\alpha(\mathbf{x}_{N+1})$ is the prediction interval with desired miscoverage rate $\alpha \in (0, 1)$, and the probability is over all samples $\{(\mathbf{x}_i, y_i)\}_{i=1}^{N+1}$, hence the name marginal coverage.

The prediction interval $C_\alpha(\mathbf{x}_{N+1})$ is constructed using *nonconformity* scores, which quantify how unusual a sample $(\mathbf{x}_i, y_i)$ is compared to other samples in a set. In the context of regression, which is the focus of this paper, the absolute residual $R_i = R(\mathbf{x}_i, y_i) = |y_i - f_i(\boldsymbol{\theta})|$ is the most common score (Kato et al., 2023). Given the data $\mathcal{D}_N$ and a score function, $C_\alpha(\mathbf{x}_{N+1})$ is defined as

$$C_\alpha(\mathbf{x}_{N+1}) = \{y \in \mathbb{R} : \pi(y) \leq \lceil (1-\alpha)(N+1) \rceil\}, \tag{3}$$

where $\pi(y) = \sum_{i=1}^{N+1} \mathbb{1}\{R_i \leq R(\mathbf{x}_{N+1}, y)\}$ is the rank of $R(\mathbf{x}_{N+1}, y)$ among the other $N$ residuals. Remarkably, the only requirement for prediction intervals constructed as in Eq. (3) to satisfy marginal coverage is that the set of scores $\{R(\mathbf{x}_i, y_i)\}_{i=1}^{N+1}$ be exchangeable. This is equivalent to assuming the data is exchangeable and the score function (and consequently the regressor) preserves this exchangeability by treating data points symmetrically. Thus, Eq. (2) is a distribution-free guarantee that remains valid even if the model is misspecified. This contrasts with Bayesian methods, the dominant approach for uncertainty quantification in deep learning, which are often poorly calibrated under model misspecification (Dawid, 1982; Fraser, 2011; Grünwald & van Ommen, 2017).

## 2.1 Split Conformal Prediction

As stated in the introduction, CP methods can generally be categorized into split-CP and full-CP variants. The primary distinction between these two families of methods lies in how they ensure the exchangeability of the set of scores $\{R(\mathbf{x}_i, y_i)\}_{i=1}^{N} \cup \{R(\mathbf{x}_{N+1}, y)\}$. Essentially, this means that all samples, including the test sample $(\mathbf{x}_{N+1}, y_{N+1})$, must exert the same influence over the prediction model $f(\cdot; \boldsymbol{\theta})$. Split-CP offers the simplest and most computationally efficient solution. By keeping the model fixed when computing the scores of each of the $N + 1$ samples, it ensures the scores are exchangeable, as the score function is applied elementwise. However, this approach is data inefficient because it prevents using the first $N$ samples to fit the model, requiring a separate training dataset. This statistical inefficiency can negatively impact the quality of the final prediction intervals in two key ways. Since any conformal prediction method is designed to achieve coverage as described in Eq. (2), different approaches are compared in terms of their *efficiency*—how small the prediction intervals are—and *adaptability*—how much the size of the prediction intervals varies across samples. To be informative about both the true label and predictive uncertainty, prediction intervals must be both efficient and adaptable. However, in split-CP, efficiency is compromised because not all data is used for calibration, while fixing the model parameters reduces adaptability.

## 2.2 Full Conformal Prediction

Full-CP uses all available data for both training the model and computing prediction intervals. The main challenge in full-CP is that exchangeability of scores requires treating all data points symmetrically, meaning the model should be fit on $\{(\mathbf{x}_i, y_i)\}_{i=1}^{N}$ as well as $(\mathbf{x}_{N+1}, y_{N+1})$. Since $y_{N+1}$ is unknown a priori, this necessitates retraining the model for all possible values $y_{N+1}$ can take. To be precise, full-CP requires the following modification to the optimization problem in Eq. (1)

$$\boldsymbol{\theta}_*^+(y) = \arg\min_{\boldsymbol{\theta}} \left( \sum_{i=1}^{N} \ell(y_i, f_i(\boldsymbol{\theta})) + \ell(y, f_{N+1}(\boldsymbol{\theta})) + \tfrac{1}{2}\delta \|\boldsymbol{\theta}\|^2 \right), \tag{4}$$

where $\boldsymbol{\theta}_*^+(y)$ is the optimal model parameters for the augmented training set $\mathcal{D}_{N+1}(y) := \mathcal{D}_N \cup \{(\mathbf{x}_{N+1}, y)\}$ that includes the test point $\mathbf{x}_{N+1}$ plus a candidate label $y$ for $y_{N+1}$. With a slight abuse of notation, we use $R_i(y)$ to denote the residual with model parameters $\boldsymbol{\theta}_*^+(y)$, that is,

$$R_i(y) = |y_i - f_i(\boldsymbol{\theta}_*^+(y))| \ \ \forall i = 1, \ldots, N \ \text{ and } \ R_{N+1}(y) = |y - f_{N+1}(\boldsymbol{\theta}_*^+(y))|. \tag{5}$$

From here we can construct prediction intervals as in Eq. (3), but now the residuals vary for each test point $\mathbf{x}_{N+1}$ and candidate label $y \in \mathbb{R}$. As mentioned before, this has two major limitations. Firstly it requires retraining the model for every candidate label $y$ which is infeasible for DNNs. Secondly in theory the method asks to consider an uncountable set (*i.e.* all real numbers). Therefore, in practice a finite grid of possible labels for $y$ is used, typically delimited by the training targets. Evidently, the grid imposes computation-precision trade-off and has implications for the coverage if a valid candidate happens to lie between two grid points (Chen et al., 2018). In a few cases, the

prediction set can be computed efficiently and exactly without the need to try candidate labels of $y$. In addition, this procedure only depends on a single fit on the original (unaugmented) dataset which can be efficiently updated not only for variations in $y$ but also for different inputs $\mathbf{x}_{N+1}$. These include the Lasso (Lei, 2019), k-Nearest Neighbours Regression (Papadopoulos et al., 2011), and ridge regression (Nouretdinov et al., 2001; Burnaev & Vovk, 2014). There are also approaches based on homotopy continuation methods (Ndiaye & Takeuchi, 2019) and algorithmic stability (Ndiaye, 2022) that hold in more general settings. In the following section, we review conformalized ridge regression which is the basis of our approximate full-CP procedure.

## 2.3 Conformalized Ridge Regression (CRR)

Ridge regression is a special case of Eq. (1) where we have a linear model $f_i(\boldsymbol{\theta}) := \mathbf{x}_i^\top \boldsymbol{\theta}$. In this case, the absolute residual can be written as a piecewise linear function of the candidate $y$ with $R_i(y) = |a_i + b_i y|$, where $a_i$ and $b_i$ are coefficients capturing information from the training data and test point, resp. It is then convenient to express the rank as $\pi(y) = \sum_{i=1}^{N+1} \mathbb{1}\{y \in S_i\}$, with

$$S_i = \{y : R_i(y) \leq R_{N+1}(y)\} = \{y : |a_i + b_i y| \leq |a_{N+1} + b_{N+1} y|\}. \tag{6}$$

Each set $S_i$ can be an interval, a point, a ray, a union of two rays, the real line, or the empty set. Eq. (6) suggests that the rank for a given $y$ can only change at points where $R_i(y) = R_{N+1}(y)$ to which we refer as "changepoints". For the absolute residual, these changepoints fall into one of the following two cases, where we assume $b_i \geq 0$ (if needed, multiplying $a_i$ and $b_i$ by $-1$):

- If $b_i \neq b_{N+1}$, then $S_i$ is an interval or a union of two rays, and we have two changepoints, $-(a_i - a_{N+1})/(b_i - b_{N+1})$ and $-(a_i + a_{N+1})/(b_i + b_{N+1})$.
- If $b_i = b_{N+1} \neq 0$, then $S_i$ is a ray, unless $a_i = a_{N+1}$, in which case $S_i = \mathbb{R}$. Here we have single changepoint $-(a_i + a_{N+1})/2b_i$.

This leads to an exact form of the prediction set by taking the union of finitely many intervals and rays whose endpoints are given by the changepoints sorted in increasing order. Given the changepoints, different implementations have been proposed with varying time complexity. For the smaller datasets, we use the ridge regression confidence machine algorithm (Nouretdinov et al., 2001; Vovk et al., 2005), which uses the absolute residual and changepoints described above.

There is also a simpler, asymmetric version of CRR that uses the signed residuals (Burnaev & Vovk, 2014). It allows us to compute the lower and upper bounds of $C_\alpha(\mathbf{x}_{N+1})$ separately, using residuals $f_i(\boldsymbol{\theta}) - y_i$ for the lower bound, and $y_i - f_i(\boldsymbol{\theta})$ for the upper bound. This is the version we use for the larger datasets and that appears in the depiction of our method in Alg. 2, where we use $l_i$ and $u_i$ to denote the changepoints for the lower and upper bounds, resp. When using signed residuals, $S_i$ is either a ray with changepoint $l_i = u_i = (a_i - a_{N+1})/(b_{N+1} - b_i)$ if $b_{N+1} - b_i > 0$ or otherwise, $S_i = \mathbb{R}$ with $l_i = -\infty$ and $u_i = \infty$.

Finally, we need to write down the expression for coefficients $a_i$ and $b_i$. Using the Sherman-Morrison formula, a widely-used tool of the regressions diagnostics literature (Cook, 1977), Burnaev & Vovk (2014) showed that the required coefficients $a_1, \ldots, a_{N+1}$ and $b_1, \ldots, b_{N+1}$ for the CRR procedure can be efficiently computed for different $\mathbf{x}_{N+1}$ by a simple rank-1 update or *perturbation* to the ridge solution on $\mathcal{D}_N$ (see App. A for derivation):

$$y_i - \mathbf{x}_i^T \boldsymbol{\theta}_*^+(y) = \underbrace{y_i - \mathbf{x}_i^T \boldsymbol{\theta}_* + \frac{h_{i,N+1}}{1 + h_{N+1}} \mathbf{x}_{N+1}^\top \boldsymbol{\theta}_*}_{a_i} - \underbrace{\frac{h_{i,N+1}}{1 + h_{N+1}} y}_{b_i} \tag{7}$$

$$y - \mathbf{x}_{N+1}^T \boldsymbol{\theta}_*^+(y) = \underbrace{-\frac{1}{1 + h_{N+1}} \mathbf{x}_{N+1}^\top \boldsymbol{\theta}_*}_{a_{N+1}} + \underbrace{\frac{1}{1 + h_{N+1}} y}_{b_{N+1}} \tag{8}$$

where $h_{N+1} = \mathbf{x}_{N+1}^\top \mathbf{H}_*^{-1} \mathbf{x}_{N+1}$ with Hessian matrix $\mathbf{H}_* = \sum_{i=1}^N \mathbf{x}_i \mathbf{x}_i^\top + \delta \mathbf{I}$.

## 2.4 Normalized Nonconformity Scores

In the previous section, we derived CRR using the nonconformity score given in Eq. (5), which is often referred to as add-one-in (AOI). In this section, we also consider the leave-one-out (LOO) and

studentized scores, which lead to the variants CRR-deleted and CRR-studentized and their corresponding extensions to neural network regression given by our approximate full-CP method. All those variants are valid choices (Vovk et al., 2005; Shafer & Vovk, 2008), and the literature is not conclusive regarding which one is to be preferred (Fong & Holmes, 2021; Fontana et al., 2023). Yet, in our neural network regression experiments, the studentized variant outperformed the other two.

In the leave-one-out (LOO) variety (also known as jackknife), scores $R_i^{\text{LOO}}$ are computed by excluding the $i^{\text{th}}$ data point from the augmented data $\mathcal{D}_{N+1}(y)$ before retraining the model. The only exception being $R_{N+1}^{\text{LOO}}$, which requires no retraining, as $\boldsymbol{\theta}_*$ already ignores the $(N+1)^{\text{th}}$ data point. In the case of ridge regression with absolute residuals, we can derive the jackknife score from the standard one as (Vovk et al., 2005)

$$R_i^{\text{LOO}} = {R_i}/{(1-\bar{h}_i)}, \tag{9}$$

where we have included the leverage score with respect to the augmented problem $\bar{h}_i = \mathbf{x}_i^\top \bar{\mathbf{H}}_*^{-1} \mathbf{x}_i$, with $\bar{\mathbf{H}}_* = \sum_{i=1}^{N+1} \mathbf{x}_i \mathbf{x}_i^\top + \delta \mathbf{I}$. The leverage score (Chatterjee & Hadi, 1986) can be viewed as a diagnostics measure for measuring the sensitivity of the prediction to changes in the target. The formula is a consequence of the exact rank-1 updates available in ridge regression. This leads to the deleted-CRR method which proceeds in the same way as standard CRR except it uses the normalized coefficients: $a_i \leftarrow a_i/(1 - \bar{h}_i)$ and $b_i \leftarrow b_i/(1 - \bar{h}_i)$ for $i = 1, \ldots, N+1$. Such relations can also be used to recover the standard score starting from the jackknife one:

$$R_i = {R_i^{\text{LOO}}}/{\left(1+\bar{h}_i^{\backslash i}\right)}, \tag{10}$$

where $\bar{h}_i^{\backslash i} = \mathbf{x}_i^\top \bar{\mathbf{H}}_*^{\backslash i} \mathbf{x}_i$ with $\bar{\mathbf{H}}_*^{\backslash i} = \sum_{j=1,j\neq i}^{N+1} \mathbf{x}_j \mathbf{x}_j^\top + \delta \mathbf{I}$. Papadopoulos (2024) gave an interpretation of this relation as locally-weighted conformal prediction (Papadopoulos et al., 2008) where the expression in the denominator can be seen as the leave-one-out predictive variance from a Bayesian perspective. This can be seen as a measure of the difficulty of the $i^{\text{th}}$ example. Similar normalizations can also be applied to nonconformity scores in a split-CP framework (Vovk et al., 2005). Finally, the studentized-CRR can be interpreted as a compromise between standard and deleted-CRR. It uses a similar normalization and defines nonconformity scores as

$$R_i^{\text{student}} = {R_i}/{\sqrt{1-\bar{h}_i}}. \tag{11}$$

This transformation is applied analogously to the CRR coefficients with $a_i$ and $b_i$.

## 3 Approximate Full-CP for Neural Network Regression

In previous work, (Martinez et al., 2023) used influence function (Jaeckel, 1972; Cook & Weisberg, 1980) to approximate the retraining step in full-CP by locally perturbing the solution on the unaugmented dataset. This ensures that a neural network is trained only once on the original data. However, this was restricted to classification problems and so their approach requires discretizing the (continuous) target space. This introduces an additional computational burden and must be done in a careful way to avoid further increasing the coverage gap (Chen et al., 2018). We propose to use a local perturbation closely related to influence function alongside network linearization in order to express the residual nonconformity score as a linear function of the candidate label that recovers Eqs. (7) and (8) as a special case. Then by leveraging the conformalized ridge regression framework (Nouretdinov et al., 2001), we can eliminate the need to specify a grid of candidate labels.

We propose *Gauss-Newton influence* to approximate the solution to the augmented problem $\hat{\boldsymbol{\theta}}_*^+(y) \approx \boldsymbol{\theta}_*^+(y)$,

$$\hat{\boldsymbol{\theta}}_*^+(y) = \boldsymbol{\theta}_* + \frac{\hat{e}_{N+1}(y)}{1 + \hat{h}_{N+1}} \mathbf{H}_{\text{GN}}^{-1} \boldsymbol{\phi}_{N+1}, \tag{12}$$

where $\boldsymbol{\phi}_i := \nabla_\theta f_i(\boldsymbol{\theta}_*)^\top$ is the Jacobian of the neural network at $\boldsymbol{\theta}_*$, $\mathbf{H}_{\text{GN}} = \sum_{i=1}^N \boldsymbol{\phi}_i \boldsymbol{\phi}_i^\top + \delta \mathbf{I}$ is the Gauss-Newton approximation to the Hessian, $\hat{e}_{N+1}(y) = y - f_{N+1}(\boldsymbol{\theta}_*)$ is the residual for the $(N+1)^{\text{th}}$ example and $\hat{h}_{N+1} = \boldsymbol{\phi}_{N+1}^\top \mathbf{H}_{\text{GN}}^{-1} \boldsymbol{\phi}_{N+1}$ can be interpreted as a kind of *generalized* leverage score (Wei et al., 1998). This is a specialization of Newton-step (NS) influence (Pregibon, 1981; Beirami et al., 2017), a technique closely related to influence function (Jaeckel, 1972; Koh & Liang, 2017), by approximating the Hessian by the Gauss-Newton matrix (Martens, 2010) followed by a rank-one update (see App. B for the derivation). Whilst such influence measures are commonly

used for estimating the effect of *removing* a single example (or group of examples) on the model, we extend this for *adding* an example, that is add-one-in (AOI) estimation. Previous work has shown that NS influence more accurately estimates the solution to the modified problem than influence function (Pregibon, 1981; Rad & Maleki, 2020). However, at present influence function is the more common choice since in its standard form NS influence requires recomputing and inverting the Hessian matrix whenever a different target effect (*e.g.* change in removed example(s)) is desired, thus incurring a higher computational cost. Our use of the Gauss-Newton approximation leads to a form amenable to a rank-one update, similar to generalized linear models (Pregibon, 1981), bringing down the complexity to that of influence function.

Next we show that using Eq. (12) along with linearization of the neural network, we can approximate the residual as a linear function of the postulated label $y$ recovering an identical form to that of conformalized ridge regression in Eqs. (7) and (8),

$$y_i - f_i(\boldsymbol{\theta}_*^+(y)) \approx \underbrace{y_i - f_i(\boldsymbol{\theta}_*) + \frac{\hat{h}_{i,N+1}}{1 + \hat{h}_{N+1}} f_{N+1}(\boldsymbol{\theta}_*)}_{a_i} - \underbrace{\frac{\hat{h}_{i,N+1}}{1 + \hat{h}_{N+1}} y}_{b_i} \tag{13}$$

$$y - f_{N+1}(\boldsymbol{\theta}_*^+(y)) \approx \underbrace{-\frac{1}{1 + \hat{h}_{N+1}} f_{N+1}(\boldsymbol{\theta}_*)}_{a_{N+1}} + \underbrace{\frac{1}{1 + \hat{h}_{N+1}} y}_{b_{N+1}} \tag{14}$$

where $\hat{h}_{i,N+1} = \boldsymbol{\phi}_i^\top \mathbf{H}_{\mathrm{GN}}^{-1} \boldsymbol{\phi}_{N+1}$ (see App. C for the derivation and App. D for the extension to the multi-output setting). This is not surprising as it is known that NS influence recovers the exact leave-one-out diagnostics in linear regression (Allen, 1974; Cook, 1977), which is not the case for influence function. The coefficients can be readily adapted for normalized nonconformity scores as outlined in Sec. 2.4. Our use of network linearization is akin to the "direct" approach used in Martinez et al. (2023) to approximate the nonconformity scores on the augmented solution directly.

In Alg. 2, we outline a simplified but complete algorithmic depiction of our method, which we dub *approximate full-CP via Gauss-Newton influence* (ACP-GN). We contrast it with standard full-CP in Alg. 1, highlighting the costly grid search and retraining steps (see App. G for a time complexity analysis). Note that in ACP-GN there is no grid search and the model is fit only once on all available data $\mathcal{D}_N$, as indicated in blue. At test time, instead of retraining the model for each $\mathbf{x}_{N+1}$, ACP-GN only computes the CRR coefficients $\{(a_i, b_i)\}_{i=1}^{N+1}$ using Gauss-Newton influence as in Eq. (13) and Eq. (14). These are then used to derive the prediction set in closed form either using the method of (Nouretdinov et al., 2001) or the asymmetric one of (Burnaev & Vovk, 2014) that we show in Alg. 2.

| **Algorithm 1:** Standard Full-CP. | **Algorithm 2:** ACP-GN (ours). |
|---|---|
| **for** each test point $\mathbf{x}_{N+1}$ **do** | optimize $\boldsymbol{\theta}_*$ as in Eq. (1) |
|   **for** each $y$ in a given grid **do** | **for** each test point $\mathbf{x}_{N+1}$ **do** |
|     optimize $\boldsymbol{\theta}_*^+(y)$ as in Eq. (4) |   compute $a_{N+1}, b_{N+1}$ as in Eq. (14) |
|     $R_{N+1}(y) = \|y - f_{N+1}(\boldsymbol{\theta}_*^+(y))\|$ |   **for** $i \in \{1, \dots N\}$ **do** |
|     **for** $i \in \{1, \dots N\}$ **do** |     compute $a_i, b_i$ as in Eq. (13) |
|       $R_i(y) = \|y_i - f_i(\boldsymbol{\theta}_*^+(y))\|$ |     **if** $b_{N+1} - b_i > 0$ **then** |
|     $\pi(y) = \sum_{i=1}^{N+1} \mathbb{1}\{R_i(y) \le R_{N+1}(y)\}$ |       $l_i = u_i = (a_i - a_{N+1})/(b_{N+1} - b_i)$ |
|     **if** $\pi(y) \le \lceil(1-\alpha)(N+1)\rceil$ **then** |     **else** |
|       include $y$ in $C_\alpha(\mathbf{x}_{N+1})$ |       $l_i = -\infty$ and $u_i = \infty$ |
| |   sort $\{l_i\}_{i=1}^N$ and $\{u_i\}_{i=1}^N$ in ascending order |
| |   $C_\alpha(\mathbf{x}_{N+1}) =$ |
| |   $\left[l_{(\lfloor(N+1)(\alpha/2)\rfloor)}, u_{(\lceil(N+1)(1-\alpha/2)\rceil)}\right]$ |

**Conformalizing Linearized Laplace**  We can show that our ACP-GN method can be interpreted as conformalizing Linearized Laplace (MacKay, 1992; Khan et al., 2019; Immer et al., 2021b) for regression. This relates to the result of Burnaev & Vovk (2014) who showed the CRR procedure can be viewed as conformalizing or "de-Bayesing" Bayesian Linear Regression (with Gaussian assumptions) (Burnaev & Vovk, 2014). They provide asymptotic results indicating that in well-specified settings, the conformal prediction intervals and Bayesian credible intervals closely align. This places

our method within the wider context of *Conformal Bayes* (Melluish et al., 2001; Wasserman, 2011) for recalibrating Bayesian intervals in case of model misspecification.

The Laplace approximation constructs a Gaussian posterior approximation centered around the point estimate $\boldsymbol{\theta}_*$ and covariance given by the inverse of the Hessian (local curvature) of the empirical risk evaluated at $\boldsymbol{\theta}_*$. When the Hessian is approximated by the generalized Gauss-Newton matrix, we refer to the resulting Laplace approximation as the Laplace-GGN posterior, $q_*(\boldsymbol{\theta}) = N(\boldsymbol{\theta}|\boldsymbol{\theta}_*, \boldsymbol{\Sigma}_*)$ where $\boldsymbol{\Sigma}_* = \mathbf{H}_{\mathrm{GN}}^{-1}$. This is often accompanied by linearizing the output of the neural network about $\boldsymbol{\theta}_*$ (Foong et al., 2019; Immer et al., 2021b):

$$f_i(\boldsymbol{\theta}) \approx f_i^{\mathrm{lin}}(\boldsymbol{\theta}) = f_i(\boldsymbol{\theta}_*) + \nabla_\theta f_i(\boldsymbol{\theta}_*)^\top (\boldsymbol{\theta} - \boldsymbol{\theta}_*). \tag{15}$$

The overall method is referred to as Linearized Laplace and retains the original NN point prediction as the mean of the posterior predictive. Analogous to the add-one-in estimate in Eq. (12), we can derive an approximation to the add-one-in posterior $\hat{q}_*^+(\boldsymbol{\theta})$ by perturbing the Laplace-GGN posterior (see App. E for derivation) whose mean is equal to $\hat{\boldsymbol{\theta}}_*^+(y)$:

$$\hat{q}_*^+(\boldsymbol{\theta}) = \mathbb{N}(\boldsymbol{\theta}|\hat{\boldsymbol{\theta}}_*^+(y), \hat{\boldsymbol{\Sigma}}_*^+) \quad \text{where } \hat{\boldsymbol{\Sigma}}_*^+ = \left(\mathbf{H}_{\mathrm{GN}} + \boldsymbol{\phi}_{N+1}\boldsymbol{\phi}_{N+1}^\top\right)^{-1} \tag{16}$$

This is a simple extension of the leave-one-out results in (Nickl et al., 2023) to the add-one-in case. Using this perturbed posterior in combination with the linearized predictor in Eq. (15), we recover Eqs. (13) and (14). From the perspective of Linearized Laplace, the Gauss-Newton influence gives the exact AOI solution with respect to the linearized network and in particular its feature expansion given the Jacobian of the network. However, it is often the case in practice that $\boldsymbol{\theta}_*$ is not a minimum of the empirical risk (*i.e.* neural network not trained to convergence). Thus, $\boldsymbol{\theta}_*$ is not a minima of the linearized network's objective. One can correct this by solving for the following objective,

$$\tilde{\boldsymbol{\theta}} = \arg\min_{\boldsymbol{\theta}} \left( \sum_{i=1}^N \tfrac{1}{2}(\tilde{y}_i - \boldsymbol{\phi}_i^\top \boldsymbol{\theta})^2 + \tfrac{1}{2}\delta \|\boldsymbol{\theta}\|^2 \right) \tag{17}$$

where $\tilde{y}_i := \boldsymbol{\phi}_i^\top \boldsymbol{\theta}_* + e_i$ with residual $e_i = y_i - f_i(\boldsymbol{\theta}_*)$. This is a linear-Gaussian system and hence can be solved for in a single step. This process is often referred to as "refinement" in the Linearized Laplace literature and has been shown to improve predictions (Immer et al., 2021b). The exact AOI solution with respect to Eq. (17) is then given by,

$$\tilde{\boldsymbol{\theta}}_*^+(y) = \tilde{\boldsymbol{\theta}} + \frac{y - f_{N+1}^{\mathrm{lin}}(\tilde{\boldsymbol{\theta}})}{1 + \hat{h}_{N+1}} \mathbf{H}_{\mathrm{GN}}^{-1}\boldsymbol{\phi}_{N+1} \tag{18}$$

which reduces to Eq. (12) when $\tilde{\boldsymbol{\theta}} = \boldsymbol{\theta}_*$. We can derive analogous expressions to Eqs. (13) and (14) that differ only in the use of linearized versions of the neural network prediction and hold exactly.

**Normalized Split-CP with Gauss-Newton Influence**    It turns out the tools we used to derive our approximate full-CP method are also effective to improve the adaptability and efficiency of split-CP. In vanilla split-CP, all prediction intervals have the same width because the model remains fixed. One way to alleviate this issue is to *normalize* the scores as $R_i = R_i/\sigma_i$, where $\sigma_i$ estimates the difficulty in predicting the $i^{\mathrm{th}}$ data point correctly (Papadopoulos et al., 2008). As observed in (Papadopoulos, 2024), the normalized scores described in Sec. 2.4 are scaled by the predictive variance, which is closely related to how difficult $y_i$ is to predict. This motivates our Gauss-Newton split-CP variant, with scores,

$$R_i = |y_i - f_i(\boldsymbol{\theta}_*)|/\sqrt{1+\hat{h}_i}, \tag{19}$$

where $\hat{h}_i = \boldsymbol{\phi}_i^\top \mathbf{H}_{\mathrm{GN}}^{-1}\boldsymbol{\phi}_i$ is the marginal variance given by Linearized Laplace. This is similar to Eq. 4.10 in (Vovk et al., 2005) and is analogous to the studentized scores in the full-CP case.

**Validity of ACP-GN**    Similar to Martinez et al. (2023), we cannot assure that our ACP-GN satisfies the coverage guarantee of full-CP. This is simply because the retraining step is locally approximated. However, bounds on the approximation error have previously been derived for Newton-step influence under fairly standard assumptions (Beirami et al., 2017; Koh et al., 2019). Whilst not shown here, we anticipate that such bounds can be extended to Gauss-Newton influence, as it is just a specialization of Newton-step influence for a certain choice of Hessian approximation. That being said in a majority of settings (datasets and target coverage levels) we empirically observe that the validity of ACP-GN does hold in practice. To address concerns about validity, in Sec. 4 we propose a variant "ACP-GN (split+refine)" that is guaranteed to provide correct coverage. This uses a train-calibration split like in split-CP along with the refinement procedure from Linearized Laplace ensuring that the residual nonconformity score expressions are exact with respect to the linearized neural network.

# 4 EXPERIMENTS & RESULTS

We compare our method, approximate full-CP via Gauss-Newton influence (ACP-GN), against Linearized Laplace (LA) (MacKay, 1992; Immer et al., 2021b), a recently popular Bayesian method for post-hoc uncertainty quantification, split conformal prediction (SCP) (Papadopoulos et al., 2002), conformalized residual fitting (CRF) (Papadopoulos et al., 2002) and conformalized quantile regression (CQR) (Romano et al., 2019). We also evaluate two further proposals, "ACP-GN (split + refine)" and "SCP-GN" which we proceed to describe along with the aforementioned methods:

- **LA**: The Laplace approximation with the Hessian approximated by the generalized Gauss-Newton matrix. The linearized predictive in Eq. (15) is used for inference. We use the implementation provided in the `Laplace` PyTorch library (Daxberger et al., 2021a) as well as to implement our ACP-GN method.

- **SCP**: Uses absolute residual nonconformity score in the procedure outlined in Sec. 2.1.

- **CRF**: Trains an additional network to predict the absolute residuals of the original network which is then used to normalize the absolute residual nonconformity score.

- **CQR**: Trains a quantile regression network using the pinball loss. The predicted lower and upper quantile functions are then used in the split conformal quantile regression algorithm.

- **ACP-GN**: Uses the *studentized* nonconformity score of Eq. (11) as we found this score performed the best.

- **SCP-GN**: Normalizes the absolute residual nonconformity score by the posterior predictive standard deviation of the LA method (trained only on the same split as SCP) as in Eq. (19).

- **ACP-GN** (split + refine): Uses a train-calibration split like in SCP, where we pretrain the model on the training set before running ACP-GN on the calibration set. More precisely, it solves the linearized network's objective in Eq. (17) but defined on the calibration set. It then uses the linearized network prediction in Eq. (15) in lieu of the original network to evaluate the CRR coefficients, again on the calibration set.

## 4.1 UCI REGRESSION

We conduct experiments on popular benchmark datasets for regression taken from the UCI Machine Learning repository (Nottingham et al., 2024). These vary in size and we adapt the experimental setup accordingly, placing the datasets into 3 groups for easy referencing: *small* (boston, concrete, energy, wine, yacht); *medium* (kin8nm, power); *large* (bike, community, protein, facebook_1, facebook_2). A subset of these are shown in Table 1 and the remainder are discussed in App. I.1. For the small datasets, the reported metrics result from 10 repeats of a 10-fold cross-validation process. For the other datasets, we perform 20 different train-test splits. In all cases, 90% of the examples are used for training and 10% for testing. When a calibration set is needed, the training set is divided into two chunks of equal size. We show results for a desired miscoverage rate of $\alpha \in \{0.1, 0.05, 0.01\}$.

All methods use neural networks trained to convergence with the Adam optimizer (Kingma & Ba, 2015). Throughout, we use fully-connected layers with 50 units and GeLU activations. The small and medium datasets have a single hidden layer whereas the large datasets use 3 hidden layers. For these architectures, it is feasible to evaluate the Gauss-Newton matrix without any approximations. However, we expect this to be prohibitive for larger architectures—inversion of the Gauss-Newton matrix scales cubically in the number of parameters. For this reason in App. J.1, we repeat the experiments using two scalable approximations: Kronecker-factored approximate curvature (KFAC) (Martens & Grosse, 2015) and last-layer approximation (Daxberger et al., 2021b) (*i.e.* neural linear model approach (Ober & Rasmussen, 2019)). As described in Sec. 2.3, to construct the predictive intervals from the coefficients in ACP-GN we use the ridge regression confidence machine algorithm (Nouretdinov et al., 2001) on the small datasets, and the asymmetric version (Burnaev & Vovk, 2014) for the medium and large datasets. See App. I.1 for further details on the experimental setup.

To assess the efficiency (tightness) and well-calibratedness of our proposed methods for obtaining prediction intervals, we report their average prediction interval width and coverage against the baselines in Table 1. A method is reported as satisfying validity if its empirical coverage lies within the 1% and 99% quantiles of the exact marginal coverage distribution as given by the train/calibration

Table 1: Our approximate full-CP via Gauss-Newton influence (ACP-GN) almost always gives the tightest intervals in limited-data regimes whilst satisfying the target coverage (cf. `yacht`, `boston`, `energy`). On larger datasets, ACP-GN remains competitive on efficiency compared with other conformal methods but can sometimes miscover. As a remedy, we propose two variants inspired by ACP-GN that generally fix the miscoverage issue. Average prediction interval width and coverage for our proposed approaches (shaded gray) against baselines (non-shaded) for three different settings of the confidence level. The best average widths over well-calibrated approaches (indicated by ✓/✗) appear in bold. Reported metrics are accompanied by standard error from repeated runs.

| | | Avg. Width | | | Avg. Coverage | | |
|---|---|---|---|---|---|---|---|
| | | 90% | 95% | 99% | 90% | 95% | 99% |
| yacht $N$=308 $I$=6 | LA | $1.690_{\pm0.017}$ | $2.014_{\pm0.020}$ | $2.647_{\pm0.027}$ | $88.73_{\pm0.61}$ (✓) | $90.78_{\pm0.59}$ (✗) | $93.89_{\pm0.60}$ (✗) |
| | SCP | $2.553_{\pm0.093}$ | $4.001_{\pm0.115}$ | $10.018_{\pm0.361}$ | $89.56_{\pm0.66}$ (✓) | $94.07_{\pm0.39}$ (✓) | $99.32_{\pm0.08}$ (✓) |
| | CRF | $2.526_{\pm0.092}$ | $3.947_{\pm0.115}$ | $9.674_{\pm0.294}$ | $89.53_{\pm0.64}$ (✓) | $94.10_{\pm0.38}$ (✓) | $99.29_{\pm0.10}$ (✓) |
| | CQR | $4.090_{\pm0.105}$ | $5.845_{\pm0.187}$ | $18.650_{\pm0.484}$ | $89.94_{\pm0.42}$ (✓) | $94.42_{\pm0.32}$ (✓) | $99.02_{\pm0.17}$ (✓) |
| | **ACP-GN** | $\mathbf{1.594}_{\pm0.016}$ | $\mathbf{2.385}_{\pm0.029}$ | $\mathbf{6.915}_{\pm0.067}$ | $87.36_{\pm0.58}$ (✓) | $92.56_{\pm0.68}$ (✓) | $99.03_{\pm0.11}$ (✓) |
| | **SCP-GN** | $2.270_{\pm0.086}$ | $3.349_{\pm0.098}$ | $\mathbf{7.216}_{\pm0.254}$ | $89.85_{\pm0.51}$ (✓) | $94.91_{\pm0.32}$ (✓) | $99.19_{\pm0.15}$ (✓) |
| | **ACP-GN** (split + refine) | $1.993_{\pm0.020}$ | $2.954_{\pm0.037}$ | $7.307_{\pm0.178}$ | $89.35_{\pm0.62}$ (✓) | $94.90_{\pm0.51}$ (✓) | $99.45_{\pm0.08}$ (✓) |
| boston $N$=506 $I$=13 | LA | $9.398_{\pm0.046}$ | $\mathbf{11.199}_{\pm0.055}$ | $14.718_{\pm0.072}$ | $91.24_{\pm0.31}$ (✓) | $94.34_{\pm0.22}$ (✓) | $97.53_{\pm0.11}$ (✗) |
| | SCP | $10.635_{\pm0.123}$ | $14.509_{\pm0.171}$ | $36.272_{\pm1.847}$ | $89.56_{\pm0.42}$ (✓) | $94.64_{\pm0.32}$ (✓) | $99.11_{\pm0.13}$ (✓) |
| | CRF | $11.932_{\pm0.605}$ | $16.073_{\pm0.862}$ | $40.690_{\pm3.333}$ | $90.01_{\pm0.33}$ (✓) | $94.77_{\pm0.22}$ (✓) | $99.30_{\pm0.08}$ (✓) |
| | CQR | $11.692_{\pm0.129}$ | $15.115_{\pm0.213}$ | $31.628_{\pm1.822}$ | $90.10_{\pm0.33}$ (✓) | $95.12_{\pm0.24}$ (✓) | $99.07_{\pm0.14}$ (✓) |
| | **ACP-GN** | $\mathbf{9.182}_{\pm0.046}$ | $12.111_{\pm0.038}$ | $\mathbf{20.512}_{\pm0.057}$ | $90.64_{\pm0.26}$ (✓) | $95.49_{\pm0.16}$ (✓) | $99.11_{\pm0.08}$ (✓) |
| | **SCP-GN** | $10.301_{\pm0.089}$ | $13.418_{\pm0.151}$ | $24.714_{\pm0.865}$ | $89.52_{\pm0.50}$ (✓) | $94.82_{\pm0.32}$ (✓) | $99.05_{\pm0.12}$ (✓) |
| | **ACP-GN** (split + refine) | $13.103_{\pm0.072}$ | $16.729_{\pm0.134}$ | $27.561_{\pm0.445}$ | $90.12_{\pm0.26}$ (✓) | $95.41_{\pm0.20}$ (✓) | $99.27_{\pm0.10}$ (✓) |
| energy $N$=768 $I$=8 | LA | $1.502_{\pm0.006}$ | $1.790_{\pm0.007}$ | $2.353_{\pm0.009}$ | $88.96_{\pm0.35}$ (✓) | $92.92_{\pm0.33}$ (✗) | $96.95_{\pm0.23}$ (✗) |
| | SCP | $1.942_{\pm0.032}$ | $2.486_{\pm0.046}$ | $3.772_{\pm0.093}$ | $89.44_{\pm0.28}$ (✓) | $94.80_{\pm0.20}$ (✓) | $99.18_{\pm0.08}$ (✓) |
| | CRF | $1.923_{\pm0.031}$ | $2.454_{\pm0.046}$ | $3.728_{\pm0.092}$ | $89.39_{\pm0.28}$ (✓) | $94.78_{\pm0.22}$ (✓) | $99.14_{\pm0.08}$ (✓) |
| | CQR | $4.670_{\pm0.030}$ | $5.139_{\pm0.029}$ | $6.438_{\pm0.120}$ | $90.08_{\pm0.26}$ (✓) | $95.24_{\pm0.21}$ (✓) | $98.96_{\pm0.09}$ (✓) |
| | **ACP-GN** | $\mathbf{1.462}_{\pm0.006}$ | $\mathbf{1.884}_{\pm0.008}$ | $\mathbf{3.076}_{\pm0.015}$ | $88.28_{\pm0.33}$ (✓) | $93.69_{\pm0.33}$ (✓) | $98.88_{\pm0.11}$ (✓) |
| | **SCP-GN** | $1.911_{\pm0.029}$ | $2.449_{\pm0.044}$ | $3.609_{\pm0.071}$ | $89.69_{\pm0.29}$ (✓) | $94.79_{\pm0.18}$ (✓) | $99.21_{\pm0.09}$ (✓) |
| | **ACP-GN** (split + refine) | $1.745_{\pm0.016}$ | $2.174_{\pm0.021}$ | $3.300_{\pm0.045}$ | $90.54_{\pm0.25}$ (✓) | $94.96_{\pm0.22}$ (✓) | $99.18_{\pm0.10}$ (✓) |
| bike $N$=10,886 $I$=18 | LA | $100.451_{\pm2.394}$ | $119.694_{\pm2.853}$ | $157.305_{\pm3.749}$ | $89.82_{\pm0.39}$ (✓) | $93.29_{\pm0.33}$ (✗) | $96.83_{\pm0.16}$ (✗) |
| | SCP | $131.138_{\pm0.812}$ | $180.477_{\pm1.244}$ | $324.756_{\pm4.635}$ | $90.33_{\pm0.21}$ (✓) | $95.17_{\pm0.15}$ (✓) | $99.00_{\pm0.07}$ (✓) |
| | CRF | $127.836_{\pm0.894}$ | $174.362_{\pm1.376}$ | $311.580_{\pm5.077}$ | $90.39_{\pm0.19}$ (✓) | $95.26_{\pm0.14}$ (✓) | $99.01_{\pm0.07}$ (✓) |
| | CQR | $141.329_{\pm5.943}$ | $\mathbf{167.682}_{\pm5.835}$ | $\mathbf{244.863}_{\pm4.952}$ | $89.83_{\pm0.23}$ (✓) | $94.80_{\pm0.14}$ (✓) | $98.89_{\pm0.07}$ (✓) |
| | **ACP-GN** | $\mathbf{98.813}_{\pm2.485}$ | $130.893_{\pm3.231}$ | $213.131_{\pm5.630}$ | $89.36_{\pm0.43}$ (✓) | $94.41_{\pm0.27}$ (✗) | $98.67_{\pm0.09}$ (✗) |
| | **SCP-GN** | $122.245_{\pm1.073}$ | $\mathbf{160.505}_{\pm1.761}$ | $254.409_{\pm3.767}$ | $90.34_{\pm0.24}$ (✓) | $95.26_{\pm0.15}$ (✓) | $99.02_{\pm0.08}$ (✓) |
| | **ACP-GN** (split + refine) | $128.336_{\pm4.336}$ | $170.782_{\pm5.859}$ | $281.632_{\pm10.106}$ | $89.98_{\pm0.22}$ (✓) | $94.94_{\pm0.16}$ (✓) | $99.01_{\pm0.06}$ (✓) |
| protein $N$=45,730 $I$=9 | LA | $9.385_{\pm0.022}$ | $11.183_{\pm0.027}$ | $14.697_{\pm0.035}$ | $85.43_{\pm0.18}$ (✗) | $89.69_{\pm0.15}$ (✗) | $94.81_{\pm0.10}$ (✗) |
| | SCP | $13.041_{\pm0.088}$ | $17.161_{\pm0.098}$ | $26.181_{\pm0.119}$ | $89.78_{\pm0.08}$ (✓) | $94.83_{\pm0.06}$ (✓) | $98.94_{\pm0.04}$ (✓) |
| | CRF | $12.645_{\pm0.127}$ | $16.931_{\pm0.146}$ | $26.973_{\pm0.202}$ | $89.86_{\pm0.09}$ (✓) | $94.84_{\pm0.17}$ (✓) | $98.93_{\pm0.04}$ (✓) |
| | CQR | $13.541_{\pm0.144}$ | $\mathbf{14.798}_{\pm0.129}$ | $\mathbf{18.239}_{\pm0.041}$ | $90.07_{\pm0.10}$ (✓) | $95.07_{\pm0.09}$ (✓) | $98.96_{\pm0.04}$ (✓) |
| | **ACP-GN** | $10.243_{\pm0.019}$ | $13.294_{\pm0.027}$ | $20.101_{\pm0.053}$ | $87.54_{\pm0.15}$ (✗) | $93.04_{\pm0.11}$ (✗) | $98.24_{\pm0.05}$ (✗) |
| | **SCP-GN** | $\mathbf{12.426}_{\pm0.085}$ | $16.102_{\pm0.096}$ | $24.032_{\pm0.138}$ | $89.78_{\pm0.10}$ (✓) | $94.86_{\pm0.08}$ (✓) | $98.94_{\pm0.03}$ (✓) |
| | **ACP-GN** (split + refine) | $12.660_{\pm0.028}$ | $16.073_{\pm0.031}$ | $23.445_{\pm0.057}$ | $89.83_{\pm0.09}$ (✓) | $94.90_{\pm0.09}$ (✓) | $98.97_{\pm0.05}$ (✓) |
| facebook_2 $N$=81,311 $I$=53 | LA | $66.088_{\pm2.760}$ | $78.749_{\pm3.289}$ | $103.493_{\pm4.322}$ | $97.47_{\pm0.12}$ (✗) | $98.01_{\pm0.09}$ (✗) | $98.65_{\pm0.06}$ (✗) |
| | SCP | $16.387_{\pm0.208}$ | $35.387_{\pm0.462}$ | $152.706_{\pm1.591}$ | $89.97_{\pm0.07}$ (✓) | $95.00_{\pm0.06}$ (✓) | $99.06_{\pm0.03}$ (✓) |
| | CRF | $\mathbf{15.088}_{\pm0.188}$ | $29.679_{\pm0.396}$ | $102.326_{\pm2.552}$ | $89.94_{\pm0.07}$ (✓) | $94.98_{\pm0.06}$ (✓) | $99.03_{\pm0.02}$ (✓) |
| | CQR | $17.605_{\pm0.645}$ | $\mathbf{21.571}_{\pm0.960}$ | $\mathbf{30.852}_{\pm1.303}$ | $90.16_{\pm0.28}$ (✓) | $95.13_{\pm0.12}$ (✓) | $99.01_{\pm0.03}$ (✓) |
| | **ACP-GN** | $18.396_{\pm0.546}$ | $40.088_{\pm1.091}$ | $166.792_{\pm5.632}$ | $90.47_{\pm0.11}$ (✗) | $95.45_{\pm0.08}$ (✗) | $99.35_{\pm0.04}$ (✗) |
| | **SCP-GN** | $16.287_{\pm0.202}$ | $33.655_{\pm0.563}$ | $118.489_{\pm4.305}$ | $89.99_{\pm0.07}$ (✓) | $94.98_{\pm0.06}$ (✓) | $99.00_{\pm0.03}$ (✓) |
| | **ACP-GN** (split + refine) | $21.469_{\pm0.906}$ | $42.184_{\pm0.788}$ | $152.460_{\pm2.499}$ | $90.14_{\pm0.08}$ (✓) | $95.10_{\pm0.06}$ (✓) | $99.13_{\pm0.02}$ (✗) |

set size (depending on the method) (Angelopoulos & Bates, 2021; Vovk, 2012). In the case of limited data regimes (yacht, boston, energy), ACP-GN gives the tighest intervals, with the exception of boston at 95% target coverage where LA is the most efficient (being one of the few cases when LA does not miscover). On the larger datasets, ACP-GN remains competitive on efficiency with the exception of facebook_2 at the higher values of target coverage, but we find it miscovers. Our proposed variant, "ACP-GN (split+refine)", generally gives the correct coverage albeit incurring a trade-off in efficiency due to sample splitting. We also observe that our novel normalization strategy inspired by ACP-GN, SCP-GN, improves over CRF for most datasets and settings of the target coverage.

## 4.2 BOUNDING BOX LOCALIZATION

We consider single-object localization and in particular adapt the task from Phan et al. (2018), which predicts bounding boxes localizing the face of different breeds of cats and dogs in varying conditions. Conformal methods have recently been adapted for this task (De Grancey et al., 2022). We construct two-sided intervals similar to Timans et al. (2024) but without considering uncertainty in the classifier, following De Grancey et al. (2022). All images with ground-truth bounding box annotations in the Oxford-IIIT Pet dataset (Parkhi et al., 2012) are extracted resulting in 3 686 images overall. The experiment is repeated with 20 different train-test splits where 20% of the data is used for testing. When needed, 25% of the training set is reserved as calibration data. The VGG-19 ar-

Table 2: On a bounding box regression task using a deep convolutional neural network, our split and refine variant of ACP-GN results in the most efficient confidence regions whilst achieving comparable coverage to the split-CP baselines. We target coverage rates of $\{85\%, 90\%, 95\%\}$ and reported metrics are accompanied by standard error from repeated runs.

| | Avg. Volume ($\times 10^{-2}$) | | | Avg. Coverage | | |
|---|---|---|---|---|---|---|
| | 85% | 90% | 95% | 85% | 90% | 95% |
| **LA** | $0.710_{\pm0.009}$ | $0.945_{\pm0.013}$ | $1.405_{\pm0.019}$ | $92.11_{\pm0.18}$ (✗) | $94.49_{\pm0.18}$ (✗) | $96.65_{\pm0.16}$ (✗) |
| **SCP** | $0.809_{\pm0.025}$ | $1.166_{\pm0.035}$ | $2.261_{\pm0.106}$ | $87.80_{\pm0.42}$ (✓) | $91.82_{\pm0.41}$ (✓) | $95.85_{\pm0.32}$ (✓) |
| **CRF** | $0.713_{\pm0.021}$ | $1.098_{\pm0.044}$ | $2.130_{\pm0.087}$ | $87.47_{\pm0.44}$ (✓) | $91.61_{\pm0.34}$ (✓) | $95.85_{\pm0.23}$ (✓) |
| **CQR** | $0.947_{\pm0.028}$ | $1.451_{\pm0.046}$ | $3.680_{\pm0.113}$ | $87.45_{\pm0.51}$ (✓) | $91.29_{\pm0.38}$ (✓) | $96.06_{\pm0.26}$ (✓) |
| **ACP-GN** | $0.384_{\pm0.004}$ | $0.605_{\pm0.007}$ | $1.225_{\pm0.017}$ | $90.97_{\pm0.20}$ (✗) | $94.09_{\pm0.20}$ (✗) | $97.41_{\pm0.16}$ (✗) |
| **SCP-GN** | $0.768_{\pm0.020}$ | $1.071_{\pm0.030}$ | $1.854_{\pm0.066}$ | $87.17_{\pm0.45}$ (✓) | $91.43_{\pm0.40}$ (✓) | $95.58_{\pm0.28}$ (✓) |
| **ACP-GN** (split + refine) | $\mathbf{0.311}_{\pm0.006}$ | $\mathbf{0.472}_{\pm0.014}$ | $\mathbf{1.031}_{\pm0.046}$ | $87.34_{\pm0.41}$ (✓) | $91.31_{\pm0.35}$ (✓) | $96.12_{\pm0.27}$ (✓) |

chitecture is used as the object detection backbone but with the original output layer removed. The network is trained jointly with two heads, a regression head predicting 2D bounding box coordinates (4 outputs) and a binary classification head. We only consider the regression head for constructing predictive intervals. Without a calibration split, the model gets $99.6\%$ classification accuracy and $20.1\%$ localization error with 0.5 IoU (Intersection over Union) threshold. With a calibration split, the model achieves $99.5\%$ classification accuracy and $27.5\%$ localization error.

We use the last-layer approximation to the Gauss-Newton matrix throughout for computational reasons. The asymmetric implementation of CRR is used due to its efficiency and we target miscoverage rates $\alpha \in \{0.15, 0.1, 0.05\}$. See App. I.2 for further experimental details. Since bounding box localization is a type of multi-output regression, we obtain confidence *regions* given by hyperrectangles except for LA that results in a hyperellipsoid. All conformal prediction methods are run for each output dimension independently and we evaluate the confidence region volumes by taking the product over interval widths per output dimension. We apply a multiple testing correction, the Bonferroni correction, to mitigate the miscoverage that arises when conformalizing the outputs separately. However, we find that all methods still consistently overcover suggesting further improvements are possible.

In Table 2, we observe that ACP-GN gives the tightest intervals as compared with the baseline conformal methods despite having a greater empirical coverage than those methods. Surprisingly our split and refine variant of ACP-GN gives even tighter intervals whilst matching the coverage of the conformal baselines. We observed a similar effect in our ablation of the previous UCI experiments with the last-layer approximation in App. J.1.

## 5 CONCLUSION

In this work, we show how to efficiently construct prediction intervals with full conformal prediction (CP) for neural networks in regression tasks. While full-CP requires retraining the model from scratch on all of the training data and for each postulated label for the test point, we show one can efficiently approximate the full-CP predictive intervals with a specialization of Newton-step influence (Pregibon, 1981; Beirami et al., 2017) and the ridge regression confidence machine of Nouretdinov et al. (2001) without retraining the model. In doing so, we also avoid relying on a grid over all possible labels for the test point, which incurs an undesirable accuracy/precision trade-off due to the uncountable set of real numbers. We demonstrate how this approach corresponds to exact full-CP on a linearized version of the neural network and further show how it corresponds to "conformalizing" the Linearized Laplace method (Khan et al., 2019; Immer et al., 2021b), a popular Bayesian approach for post-hoc uncertainty estimation in deep learning. Finally, through the lens of normalized nonconformity scores, we recover the leave-one-out variety of full-CP and easily extend to studentized scores that we find performs the best empirically and was left as future work in Martinez et al. (2023). This also leads us to propose a novel adaptive split-CP method similar to conformalized residual fitting (Papadopoulos et al., 2002) but without the need to train an additional network. Empirically, we see that our approximate full-CP methods typically provide tighter prediction intervals in limited data regimes across the well-calibrated approaches. For future work, we would like to extend our method to other real-world tasks such as pose estimation and tracking, and develop further the theoretical analysis.

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

## A  DERIVATION OF CRR COEFFICIENTS

The ridge solution on $\mathcal{D}_N$ is given by $\boldsymbol{\theta}_* = \mathbf{H}_*^{-1}\mathbf{X}^\top\mathbf{y}$ where $\mathbf{X}$ is the $(N \times I)$ feature matrix with $\mathbf{x}_i^\top$ as rows and $\mathbf{y}$ is the $N$-dim vector of targets. We can express the ridge solution on $\mathcal{D}_{N+1}(y)$, referred to as the add-one-in (AOI) solution, as a deviation from $\boldsymbol{\theta}_*$:

$$\boldsymbol{\theta}_*^+(y) = (\mathbf{H}_* + \mathbf{x}_{N+1}\mathbf{x}_{N+1}^\top)^{-1}\left(\mathbf{X}^\top\mathbf{y} + \mathbf{x}_{N+1}y\right) \tag{20}$$

▷ use Sherman-Morrison formula

$$= \left(\mathbf{H}_*^{-1} - \frac{\mathbf{H}_*^{-1}\mathbf{x}_{N+1}\mathbf{x}_{N+1}^\top\mathbf{H}_*^{-1}}{1 + \mathbf{x}_{N+1}^\top\mathbf{H}_*^{-1}\mathbf{x}_{N+1}}\right)\left(\mathbf{X}^\top\mathbf{y} + \mathbf{x}_{N+1}y\right) \tag{21}$$

$$= \mathbf{H}_*^{-1}\mathbf{X}^\top\mathbf{y} + \mathbf{H}_*^{-1}\mathbf{x}_{N+1}\left(y - \frac{\mathbf{x}_{N+1}^\top\mathbf{H}_*^{-1}\mathbf{X}^\top\mathbf{y}}{1 + \mathbf{x}_{N+1}^\top\mathbf{H}_*^{-1}\mathbf{x}_{N+1}} - \frac{y\mathbf{x}_{N+1}^\top\mathbf{H}_*^{-1}\mathbf{x}_{N+1}}{1 + \mathbf{x}_{N+1}^\top\mathbf{H}_*^{-1}\mathbf{x}_{N+1}}\right) \tag{22}$$

▷ substitute $\boldsymbol{\theta}_* = \mathbf{H}_*^{-1}\mathbf{X}^\top\mathbf{y}$ and $h_{N+1} = \mathbf{x}_{N+1}^\top\mathbf{H}_*^{-1}\mathbf{x}_{N+1}$

$$= \boldsymbol{\theta}_* + \mathbf{H}_*^{-1}\mathbf{x}_{N+1}\left(y - \frac{\mathbf{x}_{N+1}^\top\boldsymbol{\theta}_*}{1 + h_{N+1}} - \frac{yh_{N+1}}{1 + h_{N+1}}\right) \tag{23}$$

$$= \boldsymbol{\theta}_* + \mathbf{H}_*^{-1}\mathbf{x}_{N+1}\left(\frac{y - \mathbf{x}_{N+1}^\top\boldsymbol{\theta}_*}{1 + h_{N+1}}\right) \tag{24}$$

Using this, it is easy to show the residuals can be expressed in terms of the postulated label $y$:

$$y_i - \mathbf{x}_i^\top\boldsymbol{\theta}_*^+(y) = y_i - \mathbf{x}_i^\top\boldsymbol{\theta}_* - \mathbf{x}_i^\top\mathbf{H}_*^{-1}\mathbf{x}_{N+1}\left(\frac{y - \mathbf{x}_{N+1}^\top\boldsymbol{\theta}_*}{1 + h_{N+1}}\right) \tag{25}$$

▷ substitute $h_{i,N+1} = \mathbf{x}_i^\top\mathbf{H}_*^{-1}\mathbf{x}_{N+1}$

$$= y_i - \mathbf{x}_i^T\boldsymbol{\theta}_* + \frac{h_{i,N+1}}{1 + h_{N+1}}\mathbf{x}_{N+1}^\top\boldsymbol{\theta}_* - \frac{h_{i,N+1}}{1 + h_{N+1}}y \tag{26}$$

$$y - \mathbf{x}_{N+1}^\top\boldsymbol{\theta}_*^+(y) = y - \mathbf{x}_{N+1}^\top\boldsymbol{\theta}_* - \mathbf{x}_{N+1}^\top\mathbf{H}_*^{-1}\mathbf{x}_{N+1}\left(\frac{y - \mathbf{x}_{N+1}^\top\boldsymbol{\theta}_*}{1 + h_{N+1}}\right) \tag{27}$$

▷ substitute $h_{N+1} = \mathbf{x}_{N+1}^\top\mathbf{H}_*^{-1}\mathbf{x}_{N+1}$

$$= -\frac{1}{1 + h_{N+1}}\mathbf{x}_{N+1}^\top\boldsymbol{\theta}_* + \frac{1}{1 + h_{N+1}}y \tag{28}$$

## B  DERIVATION OF GAUSS-NEWTON INFLUENCE

The first two steps are similar to the usual derivation of Newton-step influence (Beirami et al., 2017) except that a single Newton step is performed on the *augmented* problem defined in Eq. (4) starting from $\boldsymbol{\theta}_*$. For brevity, we define $L(\boldsymbol{\theta}) = \sum_{i=1}^N \ell(y_i, f_i(\boldsymbol{\theta})) + \frac{1}{2}\delta\|\boldsymbol{\theta}\|^2$ as the empirical risk on $\mathcal{D}_N$.

$$\boldsymbol{\theta}_*^+(y) \approx \boldsymbol{\theta}_* - \left(\nabla_\theta^2 L(\boldsymbol{\theta}_*) + \nabla_\theta^2\ell(y, f_{N+1}(\boldsymbol{\theta}_*))\right)^{-1}\left(\nabla_\theta L(\boldsymbol{\theta}_*) + \nabla_\theta\ell(y, f_{N+1}(\boldsymbol{\theta}_*))\right) \tag{29}$$

▷ substitute first-order stationarity condition $\nabla_\theta L(\boldsymbol{\theta}_*) = 0$

$$= \boldsymbol{\theta}_* - \left(\nabla_\theta^2 L(\boldsymbol{\theta}_*) + \nabla_\theta^2\ell(y, f_{N+1}(\boldsymbol{\theta}_*))\right)^{-1}\nabla_\theta\ell(y, f_{N+1}(\boldsymbol{\theta}_*)) \tag{30}$$

▷ use Gauss-Newton approximation to Hessian

$$\approx \boldsymbol{\theta}_* - \left(\mathbf{H}_{\text{GN}} + \boldsymbol{\phi}_{N+1}\boldsymbol{\phi}_{N+1}^\top\right)^{-1}\nabla_\theta\ell(y, f_{N+1}(\boldsymbol{\theta}_*)) \tag{31}$$

▷ expand the loss gradient on the $(N+1)^{\text{th}}$ example by chain rule

$$= \boldsymbol{\theta}_* + \left(\mathbf{H}_{\text{GN}} + \boldsymbol{\phi}_{N+1}\boldsymbol{\phi}_{N+1}^\top\right)^{-1}\boldsymbol{\phi}_{N+1}\hat{e}_{N+1}(y) \tag{32}$$

▷ use Sherman-Morrison formula

$$= \boldsymbol{\theta}_* + \frac{\hat{e}_{N+1}(y)}{1 + \hat{h}_{N+1}}\mathbf{H}_{\text{GN}}^{-1}\boldsymbol{\phi}_{N+1} \tag{33}$$

## C    DERIVATION OF ACP-GN COEFFICIENTS

We present the derivation for expressing the residual in the neural network regression setting as a linear function of the postulated label $y$,

$$y_i - f_i(\boldsymbol{\theta}_*^+(y)) \tag{34}$$

▷ linearize neural network about $\boldsymbol{\theta}_*$

$$\approx y_i - \left[ f_i(\boldsymbol{\theta}_*) + \nabla_{\boldsymbol{\theta}} f_i(\boldsymbol{\theta}_*) \left( \boldsymbol{\theta}_*^+(y) - \boldsymbol{\theta}_* \right) \right] \tag{35}$$

▷ approximate $\boldsymbol{\theta}_*^+(y) \approx \hat{\boldsymbol{\theta}}_*^+(y)$ and substitute Gauss-Newton influence

$$= y_i - f_i(\boldsymbol{\theta}_*) - \boldsymbol{\phi}_i^\top \left( \frac{\hat{e}_{N+1}(y)}{1 + \hat{h}_{N+1}} \mathbf{H}_{\mathrm{GN}}^{-1} \boldsymbol{\phi}_{N+1} \right) \tag{36}$$

▷ definition of residual $\hat{e}_{N+1}(y)$ to reveal postulated label $y$ and substitute $\hat{h}_{i,N+1} = \boldsymbol{\phi}_i^\top \mathbf{H}_{\mathrm{GN}}^{-1} \boldsymbol{\phi}_{N+1}$

$$= \underbrace{y_i - f_i(\boldsymbol{\theta}_*) + \frac{\hat{h}_{i,N+1}}{1 + \hat{h}_{N+1}} f_{N+1}(\boldsymbol{\theta}_*)}_{a_i} - \underbrace{\frac{\hat{h}_{i,N+1}}{1 + \hat{h}_{N+1}} y}_{b_i} \tag{37}$$

The residual of the postulated point proceds in an almost identical manner.

## D    EXTENSION OF ACP-GN FOR MULTI-OUTPUT REGRESSION

In the case of multi-output regression with vector-valued targets $\mathbf{y}_i \in \mathbb{R}^O$ and outputs of a DNN $\mathbf{f}_i(\boldsymbol{\theta}) \in \mathbb{R}^O$, the coefficients needed in the CRR procedure $\{\mathbf{a}_i, \mathbf{B}_i\}$ now correspond to a $O$-dim vector and a $O \times O$ matrix respectively. Since CRR was only proposed for single-output regression, we also need to adapt the procedure. Considering the asymmetric version of CRR (Burnaev & Vovk, 2014), we simply need to solve for the $O$-dim changepoints which are given by $(\mathbf{B}_{N+1} - \mathbf{B}_i)^{-1}(\mathbf{a}_i - \mathbf{a}_{N+1})$ as long as $\mathbf{B}_{N+1} - \mathbf{B}_i$ is positive-definite (generalizing the positivity constraint in the single-output case). Then we propose to sort the set of changepoints followed by taking the quantile component-wise analogous to the existing algorithm. Now we proceed to derive $\{\mathbf{a}_i, \mathbf{B}_i\}$. Firstly the multi-output analogue of the AOI estimator in Eq. (12) is given by,

$$\hat{\boldsymbol{\theta}}_*^+(\mathbf{y}) = \boldsymbol{\theta}_* + \mathbf{H}_{\mathrm{GN}}^{-1} \boldsymbol{\Phi}_{N+1}^\top \left( \mathbf{I} + \hat{\mathbf{V}}_{N+1} \right)^{-1} \hat{\mathbf{e}}_{N+1}(\mathbf{y}) \tag{38}$$

where $\boldsymbol{\Phi}_i := \nabla_{\boldsymbol{\theta}} \mathbf{f}_i(\boldsymbol{\theta}_*) \in \mathbb{R}^{O \times D}$, $\mathbf{H}_{\mathrm{GN}} = \sum_{i=1}^N \boldsymbol{\Phi}_i^\top \boldsymbol{\Phi}_i + \delta \mathbf{I}$, $\hat{\mathbf{e}}_{N+1}(\mathbf{y}) = \mathbf{y} - \mathbf{f}_{N+1}(\boldsymbol{\theta}_*)$ and $\hat{\mathbf{V}}_{N+1} = \boldsymbol{\Phi}_{N+1} \mathbf{H}_{\mathrm{GN}}^{-1} \boldsymbol{\Phi}_{N+1}^\top$. There is a change in notation in the last expression of the multi-output leverage score to avoid confusion with the Hessian. Using this we can derive analogous expressions to Eqs. (13) and (14):

$$\mathbf{y}_i - \mathbf{f}_i(\boldsymbol{\theta}_*^+(\mathbf{y})) \approx \mathbf{y}_i - \left[ \mathbf{f}_i(\boldsymbol{\theta}_*) + \nabla_{\boldsymbol{\theta}} \mathbf{f}_i(\boldsymbol{\theta}_*) \left( \hat{\boldsymbol{\theta}}_*^+(\mathbf{y}) - \boldsymbol{\theta}_* \right) \right] \tag{39}$$

$$= \mathbf{y}_i - \mathbf{f}_i(\boldsymbol{\theta}_*) - \boldsymbol{\Phi}_i \mathbf{H}_{\mathrm{GN}}^{-1} \boldsymbol{\Phi}_{N+1}^\top \left( \mathbf{I} + \hat{\mathbf{V}}_{N+1} \right)^{-1} \hat{\mathbf{e}}_{N+1}(\mathbf{y}) \tag{40}$$

$$= \underbrace{\mathbf{y}_i - \mathbf{f}_i(\boldsymbol{\theta}_*) + \hat{\mathbf{V}}_{i,N+1} \left( \mathbf{I} + \hat{\mathbf{V}}_{N+1} \right)^{-1} \mathbf{f}_{N+1}(\boldsymbol{\theta}_*)}_{\mathbf{a}_i} \tag{41}$$

$$\underbrace{- \hat{\mathbf{V}}_{i,N+1} \left( \mathbf{I} + \hat{\mathbf{V}}_{N+1} \right)^{-1} \mathbf{y}}_{\mathbf{B}_i}$$

and

$$\mathbf{y} - \mathbf{f}_{N+1}(\boldsymbol{\theta}_*^+(\mathbf{y})) \approx \underbrace{- \left( \mathbf{I} + \hat{\mathbf{V}}_{N+1} \right)^{-1} \mathbf{f}_{N+1}(\boldsymbol{\theta}_*)}_{\mathbf{a}_{N+1}} + \underbrace{\left( \mathbf{I} + \hat{\mathbf{V}}_{N+1} \right)^{-1} \mathbf{y}}_{\mathbf{B}_{N+1}} \tag{42}$$

where $\hat{\mathbf{V}}_{i,N+1} = \boldsymbol{\Phi}_i \mathbf{H}_{\mathrm{GN}}^{-1} \boldsymbol{\Phi}_{N+1}^\top$. The normalized nonconformity scores can also be extended to the multi-output setting. In the case of deleted-CRR, we have:

$$\mathbf{a}_i \leftarrow \left( \mathbf{I} - \bar{\mathbf{V}}_i \right)^{-1} \mathbf{a}_i, \quad \mathbf{B}_i \leftarrow \left( \mathbf{I} - \bar{\mathbf{V}}_i \right)^{-1} \mathbf{B}_i \qquad \forall i = 1, \ldots, N+1 \tag{43}$$

where,

$$\bar{\mathbf{V}}_i = \hat{\mathbf{V}}_i - \hat{\mathbf{V}}_{i,N+1}\big(\mathbf{I} + \hat{\mathbf{V}}_{N+1}\big)^{-1}\hat{\mathbf{V}}_{i,N+1}^{\top} \quad \forall i = 1,\ldots,N \tag{44}$$

$$\bar{\mathbf{V}}_{N+1} = \hat{\mathbf{V}}_{N+1}\big(\mathbf{I} + \hat{\mathbf{V}}_{N+1}\big)^{-1} \tag{45}$$

and we introduced $\hat{\mathbf{V}}_i = \boldsymbol{\Phi}_i \mathbf{H}_{\mathrm{GN}}^{-1} \boldsymbol{\Phi}_i^{\top}$.

# E   DERIVATION OF APPROXIMATE ADD-ONE-IN POSTERIOR WITH LAPLACE-GGN

Khan et al. (2019) show that the Laplace-GGN posterior can be equivalently stated as exact inference in the following linear regression model (see their Theorem 1):

$$q_*(\boldsymbol{\theta}) \propto \prod_{i=1}^{N} e^{-\frac{1}{2}\big(\tilde{y}_i - \boldsymbol{\phi}_i^{\top}\boldsymbol{\theta}\big)^2} p(\boldsymbol{\theta}) \tag{46}$$

where $p(\boldsymbol{\theta}) \propto \exp(\frac{1}{2}\delta\,\|\boldsymbol{\theta}\|^2)$, $\boldsymbol{\phi}_i := \nabla_{\boldsymbol{\theta}} f_i(\boldsymbol{\theta}_*)^{\top}$ and $\tilde{y}_i := \boldsymbol{\phi}_i^{\top}\boldsymbol{\theta}_* + e_i$ with $e_i = y_i - f_i(\boldsymbol{\theta}_*)$. This can be viewed as approximating the original non-conjugate terms by conjugate factors (see Sec. 5.4 in (Khan & Rue, 2023)): $e^{-\ell(y_i, f_i(\boldsymbol{\theta}))} \approx e^{-\frac{1}{2}\big(\tilde{y}_i - \boldsymbol{\phi}_i^{\top}\boldsymbol{\theta}\big)^2}$, which take a similar interpretation to site functions in expectation propagation. This is used along with the standard formula for online Bayesian updating to derive the approximate AOI posterior $\hat{q}_*^+(\boldsymbol{\theta}) \approx p(\boldsymbol{\theta}|\mathcal{D}_{N+1})$,

$$p(\boldsymbol{\theta}|\mathcal{D}_{N+1}) \propto \prod_{i=1}^{N+1} e^{-\ell(y_i, f_i(\boldsymbol{\theta}))} p(\boldsymbol{\theta}) \tag{47}$$

$\triangleright$ split off the $(N+1)^{\text{th}}$ likelihood from the others

$$= e^{-\ell(y, f_{N+1}(\boldsymbol{\theta}))} \prod_{i=1}^{N} e^{-\ell(y_i, f_i(\boldsymbol{\theta}))} p(\boldsymbol{\theta}) \tag{48}$$

$\triangleright$ apply the Laplace-GGN posterior approximation

$$\approx e^{-\ell(y, f_{N+1}(\boldsymbol{\theta}))} q_*(\boldsymbol{\theta}) \tag{49}$$

$\triangleright$ approximate the $(N+1)^{\text{th}}$ likelihood by its site function

$$\approx e^{-\frac{1}{2}\big(\tilde{y} - \boldsymbol{\phi}_{N+1}^{\top}\boldsymbol{\theta}\big)^2} q_*(\boldsymbol{\theta}) \tag{50}$$

$\triangleright$ this corresponds to an unnormalized Gaussian distribution

$$\propto \mathbb{N}(\boldsymbol{\theta}|\hat{\boldsymbol{\theta}}_*^+(y), \hat{\boldsymbol{\Sigma}}_*^+) \tag{51}$$

# F   MULTI-OUTPUT REGRESSION PREDICTION INTERVALS

For a Bayesian posterior predictive distribution (under Gaussian assumptions) in the multi-output case, $p(\mathbf{y}_*|\mathbf{x}_*, \mathcal{D}) = \mathbb{N}(\mathbf{y}_*|\hat{\mathbf{y}}_*, \boldsymbol{\Sigma}_{y_*})$, the confidence *region* is given by an ellipsoid,

$$C_\alpha(\mathbf{x}_*) = \{\mathbf{y} \in \mathbb{R}^O : (\mathbf{y} - \hat{\mathbf{y}}_*)^{\top} \boldsymbol{\Sigma}_{y_*}^{-1} (\mathbf{y} - \hat{\mathbf{y}}_*) \leq \chi_{O,\alpha}^2\} \tag{52}$$

where $\chi_{O,\alpha}^2$ is the quantile function for the chi-squared distribution with $O$ degrees of freedom. Using this definition, it is straightforward to evaluate empirical coverage. Efficiency is then given by the volume of the corresponding ellipsoid,

$$\mathrm{Vol}\big[C_\alpha(\mathbf{x}_*)\big] = (\chi_{O,\alpha}^2)^{\frac{O}{2}} \det(\boldsymbol{\Sigma}_{y_*})^{\frac{1}{2}} \mathrm{Vol}[B_O] \tag{53}$$

where $B_O$ is the unit ball with $O$ dimensions.

Table 3: We have number of test points ($M$), number of grid points ($K$), number of train points ($N$), parameter count ($D$), total epochs ($E$), and cost of forward pass/gradient computation/Jacobian computation ($[FP]$). We highlight the additional complexity of FCP over ACP-GN (purple), ACP over ACP-GN (green), ACP-GN over ACP (blue) and ACP/ACP-GN over FCP (red).

|  | **Train** | **Predict** |
|---|---|---|
| FCP | — | $MKN^2E[FP]$ |
| ACP | $NE[FP] + ND^2 + D^3$ | $MK([FP] + ND^2)$ |
| ACP-GN | $NE[FP] + ND^2 + D^3$ | $M([FP] + ND^2 + N\log N)$ |

## G  TIME COMPLEXITY: FCP VS. ACP-GN

We write the time complexity for our ACP-GN and compare it against full conformal prediction (FCP) and approximate full conformal prediction (ACP) (Martinez et al., 2023) in Table 3. This is shown for the deleted nonconformity score similar to Sec. A.2 in Martinez et al. (2023) and for scalar targets only. For the standard score, ACP and ACP-GN are unchanged but the factor of $N$ is dropped in FCP. "Train" refers to the upfront time complexity that can be re-used when constructing prediction intervals ("predict") for new batches of test points.

We state the best time complexity of the ridge regression confidence machine routine given the coefficients $\{(a_i, b_i)\}_{i=1}^{N+1}$ as $\mathcal{O}(N\log N)$ (see Sec. 2.3 in (Vovk et al., 2005)). This is the same as the asymmetric implementation of Burnaev & Vovk (2014) shown in Alg. 2 (sorting the $N$ change-points). The cost of network prediction (single forward pass) is architecture-dependent but since this is constant across the methods we do not expand on this here. We also regard the cost of gradient and Jacobian computation as comparable to a forward pass (see the discussion in App. D in Novak et al. (2022)).

FCP has a complexity multiplicative in the number of test points, grid points, train points (twice) and epochs. Whereas for ACP-GN, it is just multiplicative in the number of test points, given an upfront cost which is cubic in the number of parameters. We can further reduce the time complexity for ACP-GN when scalable approximations to the Gauss-Newton Hessian are used (as investigated in App. J.1). These extensions could also be adapted for ACP which was in fact left as future work in Martinez et al. (2023). This is shown in Table 4 for the cases of Kronecker-factored approximate curvature (KFAC) and last-layer approximation (LL). Crucially, both these approximations relax the cubic dependence on the parameter count $P$ which often exceeds millions of parameters for modern architectures. Instead, the cubic dependence shifts to the input and output dimensionality of the network layers in the case of KFAC or just the last layer, which are typically far smaller than $P$.

Table 4: We have number of network layers ($L$), layer input/output dimensionality ($I_{l,\text{in}}/I_{l,\text{out}}$), cost of Gauss-Newton Hessian evaluation/inversion ($H_N$) along with its KFAC ($H_N^{kfac}$) and last-layer ($H_N^{LL}$) approximation, and cost of operations related to Gauss-Newton influence in the KFAC case $[INF]^{kfac}$. Other terms are defined in Table 3.

|  | **Train** | **Predict** |
|---|---|---|
| ACP-GN | $NE[FP] + H_N$[a] | $M([FP] + ND^2 + N\log N)$ |
| ACP-GN$(kfac)$ | $NE[FP] + H_N^{kfac}$[b] | $M([FP] + [INF]^{kfac}$[d] $+ N\log N)$ |
| ACP-GN$(LL)$ | $NE[FP] + H_N^{LL}$[c] | $M([FP] + NI_{L,\text{in}}^2 + N\log N)$ |

[a] $H_N = N[FP] + ND^2 + D^3$
[b] $H_N^{kfac} = N[FP] + N\sum_{l=1}^{L}(I_{l,\text{in}}^2 + I_{l,\text{out}}^2) + \sum_{l=1}^{L}(I_{l,\text{in}}^3 + I_{l,\text{out}}^3)$
[c] $H_N^{LL} = N[FP] + NI_{L,\text{in}}^2 + I_{L,\text{in}}^3$
[d] $[INF]^{kfac} = N\left(D + \sum_{l=1}^{L}(I_{l,\text{out}}I_{l,\text{in}}^2 + I_{l,\text{in}}I_{l,\text{out}}^2)\right)$

## H    RELATED WORK

**Computationally efficient versions of conformal prediction**    There are approaches based on data splitting such as split-CP (Papadopoulos et al., 2002; Lei et al., 2018) and cross-conformal prediction (Vovk, 2015; Barber et al., 2021). Such approaches greatly reduce the number of models that need to be trained (in the case of split-CP just one), but this comes at the expense of statistical efficiency. There are also normalized variants of split-CP (Papadopoulos et al., 2008) and other extensions in the regression setting such as conformalized quantile regression (Romano et al., 2019) and regression-to-classification CP (Guha et al., 2024). For certain model classes, computationally efficient implementations of full-CP exist such as ridge regression (Nouretdinov et al., 2001; Burnaev & Vovk, 2014), Lasso regression (Lei, 2019) and k-Nearest Neighbours Regression (Papadopoulos et al., 2011). For more general settings, there are approaches based on homotopy continuation method (Ndiaye & Takeuchi, 2019) and algorithmic stability Ndiaye (2022). The method of Martinez et al. (2023) is closely related to ours where they use influence function (Jaeckel, 1972; Cook & Weisberg, 1980) to approximate the retraining step in the full-CP algorithm. However, they only consider classification problems and their solution still requires an exhaustive search over possible labels, which is unfeasible for the regression setting we consider in this paper. A related approach is Alaa & Van Der Schaar (2020) that used higher-order influence functions to approximate leave-one-out retraining in jackknife+ (Barber et al., 2021) and Schulam & Saria (2019) that approximates the bootstrap (Efron & Tibshirani, 1986) using similar sensitivity-based techniques.

**Influence functions**    These are a family of techniques that estimate the effect on the model of deleting a single example (or group of examples) from the training set, without the computationally prohibitive cost of retraining the model. There are two main approaches, the first is the infinitesimal jackknife (IJ) or simply *influence function* that the wider family of techniques takes its namesake. This was first proposed in robust statistics (Jaeckel, 1972; Hampel, 1974) and then was introduced as an influence measure for regression diagnostics and outlier detection in Cook & Weisberg (1980). It was popularized in deep learning by Koh & Liang (2017) focusing on diagnosing model errors and data attribution. The other approach, that we use in this work, is the Newton-step (NS) influence which was first proposed in Pregibon (1981) for logistic regression and generalized to more general losses and regularizers in Beirami et al. (2017). Closely related to our use case, influence functions have been used to approximate cross-validation: IJ in Giordano et al. (2019b) and NS influence in le Cessie & van Houwelingen (1992); Beirami et al. (2017); Rad & Maleki (2020). They have also been proposed for machine unlearning: IJ in Guo et al. (2020) and NS influence in Sekhari et al. (2021). Influence functions derived using higher-order Taylor approximations have been proposed (Giordano et al., 2019a; Basu et al., 2020). Many recent works have focused on scaling IJ such as Fisher information matrix (Teso et al., 2021) or generalized Gauss-Newton (Bae et al., 2022) approximations to the Hessian, or measures defined using only gradients (Pruthi et al., 2020). Similar techniques to our Gauss-Newton influence (in the leave-one-out setting) have appeared in past works (Laurent & Cook, 1992; Hansen & Larsen, 1996; Monari & Dreyfus, 2000).

**Network linearization**    The use of gradients or Jacobians as features first appeared in Jaakkola & Haussler (1998) which they called "Fisher vectors". Such features have been demonstrated for fast adaptation (Maddox et al., 2021), transfer learning (Mu et al., 2020) and improving predictions in Bayesian neural networks (Immer et al., 2021b). This local linearization has received much attention in recent years due to the neural tangent kernel (Jacot et al., 2018) of which its finite-width variety can be viewed as the kernel analogue of Fisher-vectors.

## I    EXPERIMENTAL DETAILS

### I.1    UCI REGRESSION

For every dataset, both the input features and targets are standardized to have zero mean and unit variance. We use a batch size of 256 in SGD training with an initial learning rate of $10^{-2}$ that is decayed to $10^{-5}$ using a cosine schedule. All methods require tuning the L2 regularizer/prior precision. Additionally, for Linearized Laplace we have the observation noise. For the small and medium datasets, these are tuned using online marginal likelihood optimization (Immer et al., 2021a) that alternates between standard neural network training and gradient-based updates to the hyperparam-

Table 5: We repeat the bounding box experiment in Table 2 but with 50% calibration split.

| | Avg. Volume ($\times 10^{-2}$) | | | Avg. Coverage | | |
| | 85% | 90% | 95% | 85% | 90% | 95% |
|---|---|---|---|---|---|---|
| **LA** | $0.710_{\pm 0.009}$ | $0.945_{\pm 0.013}$ | $1.405_{\pm 0.019}$ | $92.11_{\pm 0.18}$ (✗) | $94.49_{\pm 0.18}$ (✗) | $96.65_{\pm 0.16}$ (✗) |
| **SCP** | $1.177_{\pm 0.023}$ | $1.723_{\pm 0.037}$ | $3.137_{\pm 0.081}$ | $87.36_{\pm 0.27}$ (✗) | $91.87_{\pm 0.27}$ (✗) | $95.91_{\pm 0.19}$ (✓) |
| **CRF** | $1.248_{\pm 0.030}$ | $1.900_{\pm 0.044}$ | $3.781_{\pm 0.092}$ | $87.48_{\pm 0.34}$ (✗) | $91.96_{\pm 0.25}$ (✗) | $96.02_{\pm 0.20}$ (✓) |
| **CQR** | $1.627_{\pm 0.027}$ | $2.739_{\pm 0.052}$ | $7.971_{\pm 0.159}$ | $87.46_{\pm 0.31}$ (✗) | $91.42_{\pm 0.28}$ (✓) | $95.97_{\pm 0.14}$ (✓) |
| **ACP-GN** | $0.384_{\pm 0.004}$ | $0.605_{\pm 0.007}$ | $1.225_{\pm 0.017}$ | $90.97_{\pm 0.20}$ (✗) | $94.09_{\pm 0.20}$ (✗) | $97.41_{\pm 0.16}$ (✗) |
| **SCP-GN** | $1.149_{\pm 0.025}$ | $1.615_{\pm 0.034}$ | $2.643_{\pm 0.054}$ | $86.84_{\pm 0.29}$ (✓) | $91.35_{\pm 0.31}$ (✓) | $95.56_{\pm 0.21}$ (✓) |
| **ACP-GN** (split + refine) | $0.365_{\pm 0.006}$ | $\mathbf{0.556}_{\pm 0.010}$ | $\mathbf{1.104}_{\pm 0.028}$ | $87.78_{\pm 0.37}$ (✗) | $91.75_{\pm 0.31}$ (✓) | $95.62_{\pm 0.17}$ (✓) |

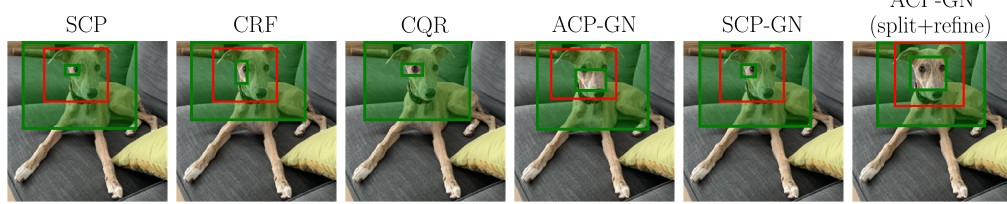

SCP   CRF   CQR   ACP-GN   SCP-GN   ACP-GN (split+refine)

Figure 2: Demonstration of our method (starting 3rd from right) for object localization alongside common conformal prediction baselines. The two-sided prediction regions are shown in green and neural network prediction is shown in red.

eters using the differentiable marginal likelihood estimate. We use a layerwise structure in the prior precision. The marginal likelihood estimate is also used for early stopping. The prior precision and observation noise are initialized to 1. Overall, using Adam optimizer we run for 5000 epochs with a hyperparameter learning rate of $10^{-2}$ decayed to $10^{-3}$ using a cosine schedule, 100 burn-in epochs, and take 50 hyperparameter steps on single marginal likelihood evaluation every 50 epochs.

For the large datasets, the (scalar) prior precision is tuned via grid-search for each order of magnitude from $10^{-2}$ to $10^4$. 10% of the training set is used for validation and once the best prior precision is found, the network is retrained on full training set. The observation noise is fit to the training data via maximum likelihood after training.

We follow the above procedure for all methods except "ACP-GN (split + refine)" where we first train with L2 regularizer fixed to $\delta/N = 10^{-4}$. For the small datasets, the hyperparameters are tuned via post-hoc marginal likelihood training with 5000 steps and the same learning rate schedule described earlier. In Fig. 3 we demonstrate the trade-off between coverage and efficiency in the split variant of ACP-GN. We set the sensitivity hyperparameter ($\beta$) of CRF (see Eq. 16 in (Papadopoulos & Haralambous, 2011)) to 1 as used in Romano et al. (2019). The additional network is identical to the original one and re-uses the same hyperparameters and training configuration. It is worth mentioning that the original proposal of CRF in the context of neural networks (Papadopoulos et al., 2002) used a ridge regression model to predict residuals.

The quantile regression network in CQR uses the same architecture as that used in the other methods except there are two output units. As done in CRF, hyperparameters from SCP are re-used along with the training configuration. Rather than retraining for each desired significance level, the output layer is appended with additional units and trained jointly. We do not perform tuning of the quantiles so it is expected the intervals can be made more efficient (Romano et al., 2019).

Results on additional UCI datasets are reported in Table 8. Furthermore, in Table 9 and 10 we repeat all experiments with 25% calibration split.

## I.2 BOUNDING BOX LOCALIZATION

The training setup is adapted from Girshick (2015) using SGD optimizer and simple data augmentation involving random horizontal flips of probability $0.5$. The robust L1 loss is used for the bounding box regression head and logistic loss for the classification head. With L1 loss, Eqs. (12) to (14) no longer hold but we demonstrate the efficacy of our procedure when the objectives for training and predictive interval construction differ.

Images are resized to $224 \times 224$ followed by scaling the pixel values to $[0, 1]$ and then normalizing to statistics computed from the ImageNet dataset as required by the pretrained VGG-19 backbone (Simonyan & Zisserman, 2015). Bounding box coordinates denote the top-left and bottom-right corners and are scaled to $[0, 1]$. The reported volumes use the standardized targets. A demonstration is shown in Fig. 2.

For SGD training we use a batch size of 128 for 200 epochs and an initial learning rate of $10^{-2}$. A learning rate schedule is used that takes incremental steps towards the base learning rate for the first 5 epochs (warm-up) and then decays to 0 using a cosine schedule. The SGD optimizer uses nesterov momentum $0.9$ and weight decay $5 \times 10^{-4}$. At inference with squared-error loss, we perform *post-hoc* finetuning of hyperparameters (regularization coefficient, observation noise) using the marginal likelihood. This is optimized using Adam for 250 epochs with a learning rate of $10^{-1}$. For LA, we use the expressions outlined in App. F to evaluate the volume and coverage.

Due to the last-layer approximation, the off-diagonal entries of the $O \times O$ multi-output leverage scores are zero. We can understand this by realizing the Jacobian of an output with respect to a parameter tied to a different output is zero. Due to this inherent output independence, we can simply use the expressions given in the single-output case in parallel over the output dimensions rather than use the expressions derived for the multi-output setting in App. D.

When training the additional network in CRF and the quantile regression network in CQR, the backbone parameters are initialized to those of the original network and not updated. They are trained in a similar fashion except for 100 epochs, without warm-up in the learning rate schedule and without data augmentation. The sensitivity hyperparameter ($\beta$) of CRF is set to $0.01$ after trying a few different values on a single seed.

In Table 5, we repeat the experiment but with 50% calibration split. In this case the model achieves 99.4% classification accuracy and 37.0% localization error.

## J    ADDITIONAL EXPERIMENTS

### J.1    ABLATION WITH SCALABLE APPROXIMATIONS TO THE GAUSS-NEWTON MATRIX

For most deep architectures, storing and inverting the full Gauss-Newton matrix is infeasible. We investigate two choices for scalable approximations to the Gauss-Newton matrix in the context of our experiments involving the large UCI datasets. These are inspired by two popular choices for scalable Laplace approximations.

The first is the use of Kronecker-factored approximate curvature (KFAC) (Martens & Grosse, 2015) as an approximation to the GN. By approximating each layer's Gauss Newton independently as a Kronecker product, this leads to a block-diagonal factorization for the overall GN matrix enabling efficient storage and computation of inverses. We use the specific form proposed in Immer et al. (2021b) that performs an eigendecomposition of the Kronecker factors and avoids the "dampening" approximation of Ritter et al. (2018). With regards to refinement, we observed that the one-step solve with KFAC approximation performs much worse than without refinement. Hence, we instead take a gradient-based approach as outlined in Antorán et al. (2022) optimizing the linearized network's objective using a similar configuration to the original NN training. The linearized network's loss gradients are evaluated without explicitly instantiating Jacobians via vector-Jacobian and Jacobian-vector products.

The results are shown in Table 11. For LA, we observe that KFAC leads to slightly tighter intervals but the miscoverage increases. In the case of SCP-GN, we find KFAC to be very competitive to the full Gauss-Newton with little change to the interval width or coverage. For ACP-GN we show results for the standard (*i.e.* AOI) nonconformity score as opposed to the studentized nonconformity score reported in Table 1 and 8. This is because we found that the deleted and studentized nonconformity scores combined with the KFAC approximation performed poorly. However for the reported standard nonconformity score, we observe KFAC leads to much tighter intervals with just a slight increase in miscoverage. For the "ACP-GN (split + refine)", on all datasets except bike, we observe tighter intervals with KFAC however the miscoverage is often slightly greater.

Table 6: We compare against Approximate full Conformal Prediction (ACP) that uses a target discretization strategy.

| | | Avg. Width | | | Avg. Coverage | | |
| --- | --- | --- | --- | --- | --- | --- | --- |
| | | 90% | 95% | 99% | 90% | 95% | 99% |
| energy | **ACP** | $1.706_{\pm0.045}$ | $2.219_{\pm0.055}$ | $3.694_{\pm0.070}$ | $87.41_{\pm0.28}$ (✓) | $93.44_{\pm0.31}$ (✓) | $98.58_{\pm0.16}$ (✓) |
| $N$=768 | **ACP-GN** | $\mathbf{1.467}_{\pm0.010}$ | $\mathbf{1.882}_{\pm0.013}$ | $\mathbf{3.097}_{\pm0.015}$ | $88.32_{\pm0.40}$ (✓) | $93.87_{\pm0.29}$ (✓) | $99.01_{\pm0.09}$ (✓) |
| $I$=8 | **ACP-GN** (split + refine) | $1.745_{\pm0.016}$ | $2.174_{\pm0.021}$ | $3.300_{\pm0.045}$ | $90.54_{\pm0.25}$ (✓) | $94.96_{\pm0.22}$ (✓) | $99.18_{\pm0.10}$ (✓) |
| concrete | **ACP** | $17.854_{\pm0.104}$ | $21.966_{\pm0.119}$ | $31.068_{\pm0.139}$ | $90.36_{\pm0.15}$ (✓) | $94.97_{\pm0.13}$ (✓) | $98.43_{\pm0.07}$ (✓) |
| $N$=1,030 | **ACP-GN** | $\mathbf{15.690}_{\pm0.112}$ | $\mathbf{19.856}_{\pm0.141}$ | $\mathbf{29.179}_{\pm0.190}$ | $89.95_{\pm0.18}$ (✓) | $94.96_{\pm0.13}$ (✓) | $98.91_{\pm0.06}$ (✓) |
| $I$=8 | **ACP-GN** (split + refine) | $18.907_{\pm0.103}$ | $24.012_{\pm0.129}$ | $35.949_{\pm0.387}$ | $90.26_{\pm0.20}$ (✓) | $95.23_{\pm0.27}$ (✓) | $99.17_{\pm0.06}$ (✓) |
| wine | **ACP** | $2.158_{\pm0.003}$ | $2.676_{\pm0.005}$ | $\mathbf{3.630}_{\pm0.008}$ | $90.46_{\pm0.13}$ (✓) | $94.88_{\pm0.08}$ (✓) | $98.22_{\pm0.07}$ (✓) |
| $N$=1,599 | **ACP-GN** | $\mathbf{2.091}_{\pm0.005}$ | $\mathbf{2.651}_{\pm0.006}$ | $3.797_{\pm0.013}$ | $90.91_{\pm0.11}$ (✓) | $95.50_{\pm0.10}$ (✓) | $99.14_{\pm0.05}$ (✓) |
| $I$=11 | **ACP-GN** (split + refine) | $2.474_{\pm0.007}$ | $3.054_{\pm0.008}$ | $4.324_{\pm0.020}$ | $89.56_{\pm0.27}$ (✓) | $94.81_{\pm0.13}$ (✓) | $98.99_{\pm0.09}$ (✓) |
| kin8nm | **ACP** | $0.207_{\pm0.001}$ | $\mathbf{0.255}_{\pm0.001}$ | $\mathbf{0.355}_{\pm0.001}$ | $88.83_{\pm0.28}$ (✗) | $94.16_{\pm0.19}$ (✓) | $98.77_{\pm0.10}$ (✓) |
| $N$=8,192 | **ACP-GN** | $\mathbf{0.213}_{\pm0.001}$ | $0.262_{\pm0.001}$ | $0.365_{\pm0.002}$ | $89.68_{\pm0.28}$ (✓) | $94.58_{\pm0.20}$ (✓) | $98.85_{\pm0.08}$ (✓) |
| $I$=8 | **ACP-GN** (split + refine) | $0.232_{\pm0.001}$ | $0.284_{\pm0.001}$ | $0.406_{\pm0.003}$ | $90.45_{\pm0.17}$ (✓) | $95.02_{\pm0.22}$ (✓) | $99.10_{\pm0.07}$ (✓) |
| power | **ACP** | $\mathbf{12.131}_{\pm0.022}$ | $\mathbf{14.840}_{\pm0.017}$ | $\mathbf{20.804}_{\pm0.050}$ | $89.28_{\pm0.25}$ (✓) | $94.68_{\pm0.18}$ (✓) | $98.82_{\pm0.10}$ (✓) |
| $N$=9,568 | **ACP-GN** | $12.526_{\pm0.024}$ | $15.248_{\pm0.020}$ | $21.592_{\pm0.062}$ | $90.16_{\pm0.26}$ (✓) | $95.13_{\pm0.19}$ (✓) | $98.89_{\pm0.08}$ (✓) |
| $I$=4 | **ACP-GN** (split + refine) | $12.744_{\pm0.045}$ | $15.442_{\pm0.039}$ | $22.043_{\pm0.151}$ | $90.17_{\pm0.29}$ (✓) | $95.26_{\pm0.17}$ (✓) | $98.98_{\pm0.09}$ (✓) |
| community | **ACP** | $\mathbf{0.401}_{\pm0.001}$ | $\mathbf{0.502}_{\pm0.003}$ | $0.700_{\pm0.005}$ | $89.71_{\pm0.48}$ (✓) | $94.00_{\pm0.36}$ (✓) | $97.71_{\pm0.30}$ (✗) |
| $N$=1,994 | **ACP-GN** | $0.459_{\pm0.003}$ | $0.594_{\pm0.004}$ | $\mathbf{0.936}_{\pm0.007}$ | $90.68_{\pm0.55}$ (✓) | $95.21_{\pm0.38}$ (✓) | $99.18_{\pm0.14}$ (✓) |
| $I$=100 | **ACP-GN** (split + refine) | $0.521_{\pm0.006}$ | $0.654_{\pm0.008}$ | $1.011_{\pm0.017}$ | $90.95_{\pm0.64}$ (✓) | $95.26_{\pm0.43}$ (✓) | $99.16_{\pm0.17}$ (✓) |
| bike | **ACP** | $\mathbf{93.647}_{\pm1.300}$ | $\mathbf{121.850}_{\pm1.603}$ | $186.090_{\pm2.393}$ | $89.03_{\pm0.31}$ (✓) | $94.36_{\pm0.22}$ (✓) | $98.43_{\pm0.14}$ (✗) |
| $N$=10,886 | **ACP-GN** | $97.966_{\pm1.193}$ | $130.732_{\pm1.565}$ | $216.677_{\pm2.917}$ | $89.12_{\pm0.21}$ (✗) | $94.32_{\pm0.18}$ (✗) | $98.72_{\pm0.09}$ (✗) |
| $I$=18 | **ACP-GN** (split + refine) | $130.933_{\pm5.221}$ | $174.190_{\pm7.065}$ | $\mathbf{288.603}_{\pm12.066}$ | $90.09_{\pm0.25}$ (✓) | $94.93_{\pm0.19}$ (✓) | $99.06_{\pm0.07}$ (✓) |

The second choice is a last-layer approximation or neural linear model approach (Ober & Rasmussen, 2019) that can be considered a special case of subnetwork inference for Linearized Laplace (Daxberger et al., 2021b). This makes the AOI estimation exact with respect to a linear model whose basis features are given by the activations of the penultimate layer. This can be recovered as a special case of Gauss-Newton influence. This leads to storing and inverting a much smaller matrix whose size corresponds to the number of parameters in the final layer. In the context of Linearized Laplace, the last-layer approximation can be combined with KFAC for increased scalability but we do not investigate this configuration here.

The results are shown in Table 12. The last-layer approximation has the effect of making the existing undercoverage in LA much worse. In the case of SCP-GN, the approximation leads to the intervals being no more efficient than SCP, that is without normalization, whilst maintaining correct coverage. For ACP-GN, the intervals undercover by a large margin on the bike, community and protein datasets. The split and refine variant is still able to successfully correct this attaining the desired coverage whilst being competitive to the full Gauss-Newton on tightness. We also observed the different varieties of nonconformity score all performed similarly – results reported are those of the studentized variety as in Table 1 and 8.

## J.2 COMPARISON AGAINST APPROXIMATE FULL CONFORMAL PREDICTION

We compare ACP-GN and ACP-GN (split+refine) to Approximate full Conformal Prediction (ACP) (Martinez et al., 2023) on several UCI datasets taken from Table 1 and 8. ACP uses the AOI nonconformity score (referred to as the ordinary scheme in (Martinez et al., 2023)). Whilst the exact Hessian is computed in (Martinez et al., 2023) with a damping term to ensure positive eigenvalues, we approximate the Hessian by a (damped) Gauss-Newton matrix in order to keep approximations consistent in the comparison. The damping term is tuned in the same way as ACP-GN, as described in App. I.1. The Gauss-Newton matrix has been used in previous works when evaluating influence function (Bae et al., 2022; Nickl et al., 2023). Furthermore, we use "direct approach" (see Eq. 4 in (Martinez et al., 2023)). Martinez et al. (2023) only considered classification tasks so ACP needs to be adapted for regression. We define a grid of candidate targets using a simple discretization strategy that constructs a fine, uniform grid of 200 points delimited by the training targets.

Table 6 shows that in some datasets and target coverage levels, ACP-GN is more efficient but in others ACP is more efficient. However, we used 200 grid points which exceeds the upper end of what was investigated in prior work (Chen et al., 2018). We also emphasise that ACP-GN has lower space complexity since it only computes changepoints for each combination of test and train point, whereas ACP computes residuals for each combination of test point, train point and candidate target value. For the datasets considered we did not observe a considerable difference in the running time between the two methods but we do expect for larger datasets ACP to be slower due to the need for batched computation.

## J.3 COMPARISON AGAINST FULL CONFORMAL PREDICTION

We compare full conformal prediction (FCP) against ACP-GN and ACP (Martinez et al., 2023) as well as LA and SCP. This is evaluated on a synthetic dataset with outliers (500 train points and 100 test points) taken from Papadopoulos (2024) (see Sec. 5.2). This generates data from a Gaussian Process prior with RBF kernel. There is a coin flip of probability $0.1$ for which the observation noise standard deviation is increased by a factor of 10. The experiment is repeated 20 times with different seeds also controlling the data generation. A MLP with a single hidden layer, 100 units and Tanh activation function is used throughout. This is trained using Adam for 500 epochs (full-batch training) with an initial learning rate of $10^{-2}$ that decays following a cosine schedule to $10^{-5}$. We use a uniform grid of 50 points for the target discretization in FCP which is on the lower end as suggested by previous works (Chen et al., 2018) and enough to give valid coverage. ACP follows the configuration outlined in App. J.2 with 200 grid points. The AOI nonconformity score is used throughout for FCP, ACP and ACP-GN. SCP uses a 50% calibration split. Hyperparameters are tuned using the online marginal likelihood procedure as described in App. I.1 which we exclude from the running time.

As Table 7 shows, all conformal methods give the correct coverage but FCP is indeed the most efficient. However, the running time is a factor of $10^4$ slower than all other methods including ACP and ACP-GN. This experiment was run on a NVIDIA A100 GPU.

Table 7: We compare against Full Conformal Prediction (FCP) on a synthetically-generated dataset.

| | Avg. Width | | | Avg. Coverage | | | Time $(\times 10^2)$ |
|---|---|---|---|---|---|---|---|
| | 90% | 95% | 99% | 90% | 95% | 99% | |
| **LA** | $1.061_{\pm 0.025}$ | $\mathbf{1.264}_{\pm 0.030}$ | $1.661_{\pm 0.039}$ | $94.45_{\pm 0.56}$ (✗) | $95.15_{\pm 0.52}$ (✓) | $96.60_{\pm 0.45}$ (✗) | $0.034_{\pm 0.001}$ |
| **SCP** | $0.623_{\pm 0.024}$ | $1.466_{\pm 0.090}$ | $3.632_{\pm 0.131}$ | $91.10_{\pm 0.82}$ (✓) | $96.10_{\pm 0.45}$ (✓) | $99.50_{\pm 0.15}$ (✓) | $0.023_{\pm 0.001}$ |
| **FCP** | $\mathbf{0.451}_{\pm 0.011}$ | $1.287_{\pm 0.053}$ | $\mathbf{3.090}_{\pm 0.092}$ | $87.00_{\pm 1.06}$ (✓) | $95.25_{\pm 0.52}$ (✓) | $98.90_{\pm 0.32}$ (✓) | $169.527_{\pm 3.560}$ |
| **ACP** | $0.532_{\pm 0.013}$ | $1.370_{\pm 0.054}$ | $3.177_{\pm 0.094}$ | $90.35_{\pm 0.79}$ (✓) | $95.50_{\pm 0.49}$ (✓) | $99.00_{\pm 0.32}$ (✓) | $0.035_{\pm 0.001}$ |
| **ACP-GN** | $0.581_{\pm 0.021}$ | $1.450_{\pm 0.053}$ | $3.529_{\pm 0.105}$ | $91.00_{\pm 0.81}$ (✓) | $95.60_{\pm 0.48}$ (✓) | $99.25_{\pm 0.27}$ (✓) | $0.034_{\pm 0.000}$ |

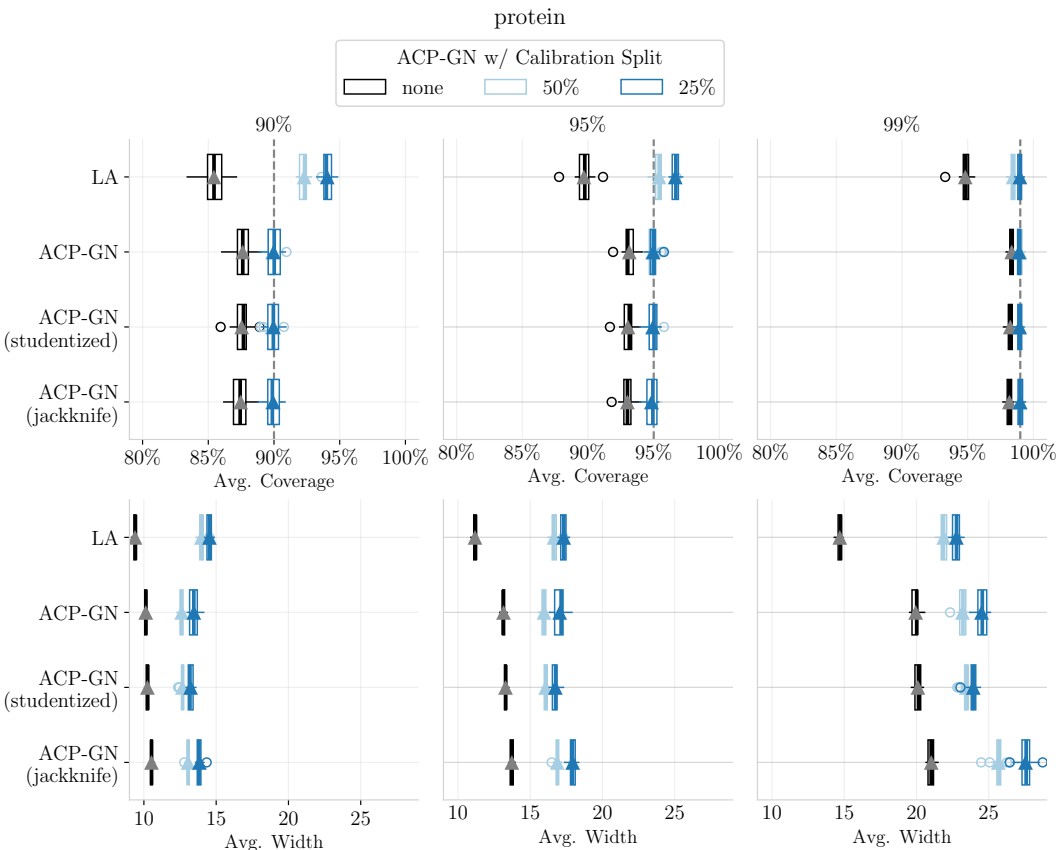

Figure 3: Our extension to ACP-GN that employs a separate calibration set to evaluate nonconformity scores combined with *refinement* of the linearized model shows considerable improvement to the coverage. Unsurprisingly it leads to larger average interval widths due to a smaller training set. This is shown for `protein` dataset at $\{90\%, 95\%, 99\%\}$ confidence levels.

Table 8: Additional results on UCI regression datasets.

| | | Avg. Width | | | Avg. Coverage | | |
|---|---|---|---|---|---|---|---|
| | | 90% | 95% | 99% | 90% | 95% | 99% |
| concrete $N$=1,030 $I$=8 | **LA** | $\mathbf{15.523}_{\pm0.087}$ | $\mathbf{18.497}_{\pm0.103}$ | $24.310_{\pm0.136}$ | $89.30_{\pm0.17}$ (✓) | $93.53_{\pm0.10}$ (✓) | $97.37_{\pm0.11}$ (✗) |
| | **SCP** | $19.216_{\pm0.124}$ | $24.526_{\pm0.194}$ | $42.110_{\pm0.802}$ | $89.92_{\pm0.29}$ (✓) | $94.97_{\pm0.14}$ (✓) | $99.11_{\pm0.05}$ (✓) |
| | **CRF** | $18.623_{\pm0.117}$ | $23.737_{\pm0.167}$ | $40.390_{\pm0.506}$ | $89.75_{\pm0.36}$ (✓) | $94.97_{\pm0.13}$ (✓) | $99.05_{\pm0.06}$ (✓) |
| | **CQR** | $22.486_{\pm0.081}$ | $27.145_{\pm0.091}$ | $39.827_{\pm0.212}$ | $90.51_{\pm0.27}$ (✓) | $95.24_{\pm0.16}$ (✓) | $99.01_{\pm0.08}$ (✓) |
| | **ACP-GN** | $15.727_{\pm0.114}$ | $19.810_{\pm0.147}$ | $\mathbf{29.192}_{\pm0.179}$ | $89.60_{\pm0.20}$ (✓) | $94.77_{\pm0.14}$ (✓) | $98.85_{\pm0.08}$ (✓) |
| | **SCP-GN** | $18.851_{\pm0.118}$ | $23.780_{\pm0.121}$ | $36.686_{\pm0.313}$ | $89.68_{\pm0.28}$ (✓) | $94.87_{\pm0.14}$ (✓) | $99.15_{\pm0.07}$ (✓) |
| | **ACP-GN** (split + refine) | $18.907_{\pm0.103}$ | $24.012_{\pm0.129}$ | $35.949_{\pm0.387}$ | $90.26_{\pm0.20}$ (✓) | $95.23_{\pm0.27}$ (✓) | $99.17_{\pm0.06}$ (✓) |
| wine $N$=1,599 $I$=11 | **LA** | $2.099_{\pm0.001}$ | $\mathbf{2.501}_{\pm0.002}$ | $\mathbf{3.287}_{\pm0.002}$ | $91.12_{\pm0.13}$ (✓) | $94.58_{\pm0.10}$ (✓) | $98.34_{\pm0.05}$ (✓) |
| | **SCP** | $2.183_{\pm0.009}$ | $2.768_{\pm0.012}$ | $3.941_{\pm0.020}$ | $90.41_{\pm0.19}$ (✓) | $95.07_{\pm0.12}$ (✓) | $99.12_{\pm0.04}$ (✓) |
| | **CRF** | $2.304_{\pm0.009}$ | $2.921_{\pm0.015}$ | $4.381_{\pm0.031}$ | $90.14_{\pm0.18}$ (✓) | $95.16_{\pm0.15}$ (✓) | $99.02_{\pm0.05}$ (✓) |
| | **CQR** | $\mathbf{1.977}_{\pm0.005}$ | $2.490_{\pm0.016}$ | $3.867_{\pm0.027}$ | $90.24_{\pm0.12}$ (✓) | $95.05_{\pm0.06}$ (✓) | $99.07_{\pm0.06}$ (✓) |
| | **ACP-GN** | $2.103_{\pm0.003}$ | $2.665_{\pm0.006}$ | $3.827_{\pm0.007}$ | $91.16_{\pm0.12}$ (✓) | $95.80_{\pm0.05}$ (✓) | $99.17_{\pm0.04}$ (✓) |
| | **SCP-GN** | $2.134_{\pm0.007}$ | $2.676_{\pm0.015}$ | $3.753_{\pm0.026}$ | $90.06_{\pm0.24}$ (✓) | $95.09_{\pm0.16}$ (✓) | $99.01_{\pm0.05}$ (✓) |
| | **ACP-GN** (split + refine) | $2.474_{\pm0.007}$ | $3.054_{\pm0.008}$ | $4.324_{\pm0.020}$ | $89.56_{\pm0.27}$ (✓) | $94.81_{\pm0.13}$ (✓) | $98.99_{\pm0.09}$ (✓) |
| kin8nm $N$=8,192 $I$=8 | **LA** | $\mathbf{0.213}_{\pm0.001}$ | $0.254_{\pm0.001}$ | $0.334_{\pm0.001}$ | $89.66_{\pm0.29}$ (✓) | $93.88_{\pm0.23}$ (✗) | $98.09_{\pm0.10}$ (✗) |
| | **SCP** | $0.231_{\pm0.001}$ | $0.285_{\pm0.001}$ | $0.409_{\pm0.003}$ | $90.54_{\pm0.29}$ (✓) | $95.31_{\pm0.20}$ (✓) | $99.18_{\pm0.08}$ (✓) |
| | **CRF** | $0.229_{\pm0.001}$ | $0.282_{\pm0.001}$ | $0.401_{\pm0.002}$ | $90.59_{\pm0.31}$ (✓) | $95.41_{\pm0.23}$ (✓) | $99.18_{\pm0.06}$ (✓) |
| | **CQR** | $0.254_{\pm0.001}$ | $0.304_{\pm0.001}$ | $0.426_{\pm0.003}$ | $90.24_{\pm0.27}$ (✓) | $95.21_{\pm0.25}$ (✓) | $99.02_{\pm0.09}$ (✓) |
| | **ACP-GN** | $\mathbf{0.213}_{\pm0.001}$ | $\mathbf{0.262}_{\pm0.001}$ | $\mathbf{0.365}_{\pm0.002}$ | $89.68_{\pm0.28}$ (✓) | $94.58_{\pm0.20}$ (✓) | $98.85_{\pm0.08}$ (✓) |
| | **SCP-GN** | $0.230_{\pm0.001}$ | $0.282_{\pm0.001}$ | $0.400_{\pm0.002}$ | $90.57_{\pm0.29}$ (✓) | $95.20_{\pm0.22}$ (✓) | $99.18_{\pm0.07}$ (✓) |
| | **ACP-GN** (split + refine) | $0.232_{\pm0.001}$ | $0.284_{\pm0.001}$ | $0.406_{\pm0.003}$ | $90.45_{\pm0.17}$ (✓) | $95.02_{\pm0.22}$ (✓) | $99.10_{\pm0.07}$ (✓) |
| power $N$=9,568 $I$=4 | **LA** | $13.345_{\pm0.028}$ | $15.901_{\pm0.034}$ | $\mathbf{20.898}_{\pm0.045}$ | $92.08_{\pm0.23}$ (✗) | $95.86_{\pm0.17}$ (✗) | $98.86_{\pm0.09}$ (✓) |
| | **SCP** | $12.732_{\pm0.040}$ | $15.359_{\pm0.041}$ | $21.688_{\pm0.143}$ | $90.27_{\pm0.24}$ (✓) | $94.94_{\pm0.17}$ (✓) | $98.98_{\pm0.10}$ (✓) |
| | **CRF** | $\mathbf{12.573}_{\pm0.039}$ | $\mathbf{15.130}_{\pm0.039}$ | $21.548_{\pm0.145}$ | $90.26_{\pm0.27}$ (✓) | $94.93_{\pm0.13}$ (✓) | $98.98_{\pm0.09}$ (✓) |
| | **CQR** | $12.860_{\pm0.026}$ | $\mathbf{15.180}_{\pm0.042}$ | $21.946_{\pm0.154}$ | $90.37_{\pm0.24}$ (✓) | $94.93_{\pm0.14}$ (✓) | $98.89_{\pm0.06}$ (✓) |
| | **ACP-GN** | $\mathbf{12.526}_{\pm0.024}$ | $15.248_{\pm0.020}$ | $21.592_{\pm0.062}$ | $90.16_{\pm0.26}$ (✓) | $95.13_{\pm0.19}$ (✓) | $98.89_{\pm0.08}$ (✓) |
| | **SCP-GN** | $12.736_{\pm0.041}$ | $15.362_{\pm0.044}$ | $21.680_{\pm0.146}$ | $90.25_{\pm0.25}$ (✓) | $94.95_{\pm0.18}$ (✓) | $98.98_{\pm0.10}$ (✓) |
| | **ACP-GN** (split + refine) | $12.744_{\pm0.045}$ | $15.442_{\pm0.039}$ | $22.043_{\pm0.151}$ | $90.17_{\pm0.29}$ (✓) | $95.26_{\pm0.17}$ (✓) | $98.98_{\pm0.09}$ (✓) |
| community $N$=1,994 $I$=100 | **LA** | $0.548_{\pm0.074}$ | $0.653_{\pm0.088}$ | $0.858_{\pm0.116}$ | $90.90_{\pm0.59}$ (✓) | $93.83_{\pm0.50}$ (✓) | $97.05_{\pm0.33}$ (✗) |
| | **SCP** | $0.534_{\pm0.010}$ | $0.731_{\pm0.020}$ | $1.159_{\pm0.028}$ | $90.30_{\pm0.42}$ (✓) | $95.53_{\pm0.24}$ (✓) | $99.12_{\pm0.16}$ (✓) |
| | **CRF** | $0.526_{\pm0.010}$ | $0.721_{\pm0.020}$ | $1.154_{\pm0.029}$ | $90.20_{\pm0.43}$ (✓) | $95.33_{\pm0.25}$ (✓) | $99.12_{\pm0.13}$ (✓) |
| | **CQR** | $0.554_{\pm0.020}$ | $\mathbf{0.701}_{\pm0.034}$ | $\mathbf{1.077}_{\pm0.038}$ | $90.90_{\pm0.57}$ (✓) | $95.70_{\pm0.30}$ (✓) | $99.35_{\pm0.15}$ (✓) |
| | **ACP-GN** | $\mathbf{0.570}_{\pm0.108}$ | $\mathbf{0.755}_{\pm0.158}$ | $\mathbf{1.224}_{\pm0.285}$ | $90.90_{\pm0.62}$ (✓) | $95.30_{\pm0.47}$ (✓) | $99.25_{\pm0.14}$ (✓) |
| | **SCP-GN** | $\mathbf{0.474}_{\pm0.013}$ | $\mathbf{0.656}_{\pm0.024}$ | $1.104_{\pm0.034}$ | $90.58_{\pm0.37}$ (✓) | $94.95_{\pm0.27}$ (✓) | $99.00_{\pm0.15}$ (✓) |
| | **ACP-GN** (split + refine) | $0.519_{\pm0.006}$ | $\mathbf{0.652}_{\pm0.008}$ | $\mathbf{1.010}_{\pm0.016}$ | $90.97_{\pm0.61}$ (✓) | $95.28_{\pm0.41}$ (✓) | $99.12_{\pm0.16}$ (✓) |
| facebook_1 $N$=40,948 $I$=53 | **LA** | $67.580_{\pm2.637}$ | $80.527_{\pm3.142}$ | $105.831_{\pm4.130}$ | $97.63_{\pm0.09}$ (✗) | $98.15_{\pm0.08}$ (✗) | $98.76_{\pm0.06}$ (✗) |
| | **SCP** | $\mathbf{20.771}_{\pm2.452}$ | $44.192_{\pm2.799}$ | $178.890_{\pm4.596}$ | $90.00_{\pm0.10}$ (✓) | $95.03_{\pm0.08}$ (✓) | $99.05_{\pm0.04}$ (✓) |
| | **CRF** | $\mathbf{18.689}_{\pm1.944}$ | $35.765_{\pm1.870}$ | $121.181_{\pm4.148}$ | $89.98_{\pm0.12}$ (✓) | $95.10_{\pm0.09}$ (✓) | $99.08_{\pm0.04}$ (✓) |
| | **CQR** | $19.625_{\pm0.878}$ | $\mathbf{25.970}_{\pm1.322}$ | $\mathbf{59.922}_{\pm12.774}$ | $90.16_{\pm0.24}$ (✓) | $94.91_{\pm0.11}$ (✓) | $98.97_{\pm0.04}$ (✓) |
| | **ACP-GN** | $\mathbf{17.986}_{\pm0.480}$ | $41.063_{\pm0.770}$ | $199.331_{\pm10.821}$ | $90.36_{\pm0.16}$ (✓) | $95.56_{\pm0.17}$ (✗) | $99.37_{\pm0.08}$ (✗) |
| | **SCP-GN** | $\mathbf{20.460}_{\pm2.463}$ | $39.712_{\pm3.148}$ | $125.078_{\pm8.075}$ | $90.00_{\pm0.10}$ (✓) | $94.94_{\pm0.09}$ (✓) | $99.04_{\pm0.03}$ (✓) |
| | **ACP-GN** (split + refine) | $29.006_{\pm4.472}$ | $54.099_{\pm4.576}$ | $172.252_{\pm6.758}$ | $90.06_{\pm0.10}$ (✓) | $95.20_{\pm0.08}$ (✓) | $99.14_{\pm0.04}$ (✓) |

Table 9: We repeat the UCI regression experiment in Table 1 but with 25% calibration split. In the case of yacht, the desired coverage of 99% was too small a significance level to accurately evaluate the quantile of the calibration scores.

| | | Avg. Width | | | Avg. Coverage | | |
|---|---|---|---|---|---|---|---|
| | | 90% | 95% | 99% | 90% | 95% | 99% |
| yacht $N=308$ $I=6$ | **LA** | $1.690_{\pm0.017}$ | $2.014_{\pm0.020}$ | $2.647_{\pm0.027}$ | $88.73_{\pm0.61}$ (✓) | $90.78_{\pm0.59}$ (✗) | $93.89_{\pm0.60}$ (✗) |
| | **SCP** | $2.306_{\pm0.117}$ | $3.866_{\pm0.133}$ | — | $90.11_{\pm0.70}$ (✓) | $95.88_{\pm0.43}$ (✓) | — |
| | **CRF** | $2.281_{\pm0.112}$ | $3.818_{\pm0.130}$ | — | $90.11_{\pm0.67}$ (✓) | $95.82_{\pm0.46}$ (✓) | — |
| | **CQR** | $3.236_{\pm0.151}$ | $4.895_{\pm0.211}$ | — | $90.33_{\pm0.65}$ (✓) | $95.85_{\pm0.39}$ (✓) | — |
| | **ACP-GN** | $\mathbf{1.594}_{\pm0.016}$ | $\mathbf{2.385}_{\pm0.029}$ | $\mathbf{6.915}_{\pm0.067}$ | $87.36_{\pm0.58}$ (✓) | $92.56_{\pm0.68}$ (✓) | $99.03_{\pm0.11}$ (✓) |
| | **SCP-GN** | $2.070_{\pm0.084}$ | $3.111_{\pm0.110}$ | — | $90.53_{\pm0.54}$ (✓) | $95.95_{\pm0.51}$ (✓) | — |
| | **ACP-GN** (split + refine) | $3.931_{\pm0.083}$ | $5.926_{\pm0.144}$ | — | $91.66_{\pm0.46}$ (✓) | $96.66_{\pm0.41}$ (✓) | — |
| boston $N=506$ $I=13$ | **LA** | $9.398_{\pm0.046}$ | $\mathbf{11.199}_{\pm0.055}$ | $14.718_{\pm0.072}$ | $91.24_{\pm0.31}$ (✓) | $94.34_{\pm0.22}$ (✓) | $97.53_{\pm0.11}$ (✗) |
| | **SCP** | $10.086_{\pm0.184}$ | $14.746_{\pm0.345}$ | $38.574_{\pm1.914}$ | $90.12_{\pm0.51}$ (✓) | $96.03_{\pm0.34}$ (✓) | $99.19_{\pm0.12}$ (✓) |
| | **CRF** | $9.874_{\pm0.194}$ | $14.204_{\pm0.270}$ | $37.421_{\pm2.132}$ | $90.03_{\pm0.45}$ (✓) | $95.52_{\pm0.24}$ (✓) | $99.28_{\pm0.11}$ (✓) |
| | **CQR** | $10.918_{\pm0.121}$ | $14.618_{\pm0.208}$ | $31.130_{\pm0.823}$ | $90.09_{\pm0.31}$ (✓) | $95.65_{\pm0.21}$ (✓) | $99.17_{\pm0.10}$ (✓) |
| | **ACP-GN** | $\mathbf{9.182}_{\pm0.046}$ | $12.111_{\pm0.038}$ | $\mathbf{20.512}_{\pm0.057}$ | $90.64_{\pm0.26}$ (✓) | $95.49_{\pm0.16}$ (✓) | $99.11_{\pm0.08}$ (✓) |
| | **SCP-GN** | $9.787_{\pm0.152}$ | $13.372_{\pm0.247}$ | $27.274_{\pm1.221}$ | $90.21_{\pm0.44}$ (✓) | $95.49_{\pm0.29}$ (✓) | $99.13_{\pm0.14}$ (✓) |
| | **ACP-GN** (split + refine) | $17.372_{\pm0.252}$ | $22.852_{\pm0.371}$ | $42.081_{\pm1.170}$ | $90.68_{\pm0.44}$ (✓) | $95.79_{\pm0.28}$ (✓) | $98.95_{\pm0.15}$ (✓) |
| energy $N=768$ $I=8$ | **LA** | $1.502_{\pm0.006}$ | $1.790_{\pm0.007}$ | $2.353_{\pm0.009}$ | $88.96_{\pm0.35}$ (✓) | $92.92_{\pm0.33}$ (✗) | $96.95_{\pm0.23}$ (✗) |
| | **SCP** | $1.824_{\pm0.037}$ | $2.365_{\pm0.046}$ | $4.313_{\pm0.129}$ | $90.05_{\pm0.45}$ (✓) | $95.04_{\pm0.25}$ (✓) | $99.39_{\pm0.09}$ (✓) |
| | **CRF** | $1.807_{\pm0.037}$ | $2.341_{\pm0.046}$ | $4.316_{\pm0.133}$ | $89.97_{\pm0.47}$ (✓) | $95.02_{\pm0.23}$ (✓) | $99.36_{\pm0.10}$ (✓) |
| | **CQR** | $4.779_{\pm0.032}$ | $5.181_{\pm0.035}$ | $7.692_{\pm0.171}$ | $90.55_{\pm0.26}$ (✓) | $95.51_{\pm0.22}$ (✓) | $99.35_{\pm0.06}$ (✓) |
| | **ACP-GN** | $\mathbf{1.462}_{\pm0.006}$ | $\mathbf{1.884}_{\pm0.008}$ | $\mathbf{3.076}_{\pm0.015}$ | $88.28_{\pm0.33}$ (✓) | $93.69_{\pm0.33}$ (✓) | $98.88_{\pm0.11}$ (✓) |
| | **SCP-GN** | $1.812_{\pm0.034}$ | $2.348_{\pm0.047}$ | $4.226_{\pm0.120}$ | $90.16_{\pm0.43}$ (✓) | $95.19_{\pm0.24}$ (✓) | $99.31_{\pm0.11}$ (✓) |
| | **ACP-GN** (split + refine) | $2.498_{\pm0.056}$ | $3.251_{\pm0.072}$ | $5.601_{\pm0.168}$ | $90.19_{\pm0.53}$ (✓) | $95.28_{\pm0.37}$ (✓) | $99.43_{\pm0.07}$ (✓) |
| bike $N=10{,}886$ $I=18$ | **LA** | $100.451_{\pm2.394}$ | $119.694_{\pm2.853}$ | $157.305_{\pm3.749}$ | $89.82_{\pm0.39}$ (✓) | $93.29_{\pm0.33}$ (✗) | $96.83_{\pm0.16}$ (✗) |
| | **SCP** | $113.594_{\pm1.370}$ | $159.114_{\pm2.297}$ | $287.486_{\pm5.425}$ | $90.48_{\pm0.22}$ (✓) | $95.11_{\pm0.20}$ (✓) | $98.87_{\pm0.08}$ (✓) |
| | **CRF** | $112.104_{\pm1.653}$ | $156.161_{\pm2.486}$ | $283.911_{\pm6.247}$ | $90.38_{\pm0.22}$ (✓) | $95.17_{\pm0.19}$ (✓) | $98.93_{\pm0.09}$ (✓) |
| | **CQR** | $\mathbf{107.141}_{\pm3.598}$ | $\mathbf{130.756}_{\pm3.259}$ | $\mathbf{212.513}_{\pm5.131}$ | $90.34_{\pm0.26}$ (✓) | $95.18_{\pm0.18}$ (✓) | $98.97_{\pm0.09}$ (✓) |
| | **ACP-GN** | $\mathbf{98.813}_{\pm2.485}$ | $130.893_{\pm3.231}$ | $213.131_{\pm5.630}$ | $89.36_{\pm0.43}$ (✓) | $94.41_{\pm0.27}$ (✗) | $98.67_{\pm0.09}$ (✗) |
| | **SCP-GN** | $105.020_{\pm1.209}$ | $139.391_{\pm1.711}$ | $228.656_{\pm4.493}$ | $90.17_{\pm0.21}$ (✓) | $95.10_{\pm0.16}$ (✓) | $98.99_{\pm0.07}$ (✓) |
| | **ACP-GN** (split + refine) | $137.114_{\pm0.759}$ | $178.639_{\pm1.307}$ | $285.997_{\pm3.834}$ | $90.20_{\pm0.18}$ (✓) | $95.05_{\pm0.16}$ (✓) | $99.16_{\pm0.10}$ (✓) |
| protein $N=45{,}730$ $I=9$ | **LA** | $9.385_{\pm0.022}$ | $11.183_{\pm0.027}$ | $14.697_{\pm0.035}$ | $85.43_{\pm0.18}$ (✗) | $89.69_{\pm0.15}$ (✗) | $94.81_{\pm0.10}$ (✗) |
| | **SCP** | $12.188_{\pm0.036}$ | $16.069_{\pm0.050}$ | $24.828_{\pm0.126}$ | $89.85_{\pm0.11}$ (✓) | $94.91_{\pm0.07}$ (✓) | $98.92_{\pm0.05}$ (✓) |
| | **CRF** | $\mathbf{11.532}_{\pm0.047}$ | $15.398_{\pm0.071}$ | $25.006_{\pm0.147}$ | $89.89_{\pm0.09}$ (✓) | $94.93_{\pm0.08}$ (✓) | $98.92_{\pm0.03}$ (✓) |
| | **CQR** | $13.174_{\pm0.162}$ | $\mathbf{14.422}_{\pm0.148}$ | $\mathbf{17.924}_{\pm0.078}$ | $90.10_{\pm0.09}$ (✓) | $95.04_{\pm0.05}$ (✓) | $98.92_{\pm0.04}$ (✓) |
| | **ACP-GN** | $10.243_{\pm0.019}$ | $13.294_{\pm0.027}$ | $20.101_{\pm0.053}$ | $87.54_{\pm0.15}$ (✗) | $93.04_{\pm0.11}$ (✗) | $98.24_{\pm0.05}$ (✗) |
| | **SCP-GN** | $11.743_{\pm0.035}$ | $15.312_{\pm0.035}$ | $23.243_{\pm0.097}$ | $89.87_{\pm0.09}$ (✓) | $94.91_{\pm0.07}$ (✓) | $98.95_{\pm0.04}$ (✓) |
| | **ACP-GN** (split + refine) | $13.229_{\pm0.042}$ | $16.741_{\pm0.055}$ | $23.927_{\pm0.068}$ | $89.96_{\pm0.11}$ (✓) | $94.93_{\pm0.10}$ (✓) | $98.94_{\pm0.05}$ (✓) |
| facebook_2 $N=81{,}311$ $I=53$ | **LA** | $66.088_{\pm2.760}$ | $78.749_{\pm3.289}$ | $103.493_{\pm4.322}$ | $97.47_{\pm0.21}$ (✗) | $98.01_{\pm0.09}$ (✗) | $98.65_{\pm0.06}$ (✗) |
| | **SCP** | $\mathbf{15.739}_{\pm0.240}$ | $34.114_{\pm0.646}$ | $145.543_{\pm1.431}$ | $89.85_{\pm0.09}$ (✓) | $94.96_{\pm0.06}$ (✓) | $99.01_{\pm0.03}$ (✓) |
| | **CRF** | $\mathbf{15.257}_{\pm0.663}$ | $29.889_{\pm1.398}$ | $100.709_{\pm6.136}$ | $89.82_{\pm0.08}$ (✓) | $94.97_{\pm0.05}$ (✓) | $99.01_{\pm0.03}$ (✓) |
| | **CQR** | $17.929_{\pm0.703}$ | $\mathbf{22.314}_{\pm1.070}$ | $\mathbf{32.257}_{\pm1.604}$ | $89.85_{\pm0.06}$ (✓) | $95.00_{\pm0.04}$ (✓) | $99.02_{\pm0.03}$ (✓) |
| | **ACP-GN** | $18.396_{\pm0.546}$ | $40.088_{\pm1.091}$ | $166.792_{\pm5.632}$ | $90.47_{\pm0.11}$ (✗) | $95.45_{\pm0.08}$ (✗) | $99.35_{\pm0.04}$ (✗) |
| | **SCP-GN** | $\mathbf{15.633}_{\pm0.220}$ | $33.192_{\pm0.708}$ | $122.079_{\pm4.142}$ | $89.81_{\pm0.07}$ (✓) | $94.96_{\pm0.06}$ (✓) | $99.00_{\pm0.02}$ (✓) |
| | **ACP-GN** (split + refine) | $20.487_{\pm0.502}$ | $41.051_{\pm0.844}$ | $152.090_{\pm6.966}$ | $90.18_{\pm0.08}$ (✓) | $95.08_{\pm0.07}$ (✓) | $99.11_{\pm0.03}$ (✓) |

Table 10: We repeat the UCI regression experiment in Table 8 but with 25% calibration split.

| | | Avg. Width | | | Avg. Coverage | | |
|---|---|---|---|---|---|---|---|
| | | 90% | 95% | 99% | 90% | 95% | 99% |
| concrete $N$=1,030 $I$=8 | LA | $15.523_{\pm0.087}$ | $18.497_{\pm0.103}$ | $24.310_{\pm0.136}$ | $89.30_{\pm0.17}$ (✓) | $93.53_{\pm0.10}$ (✓) | $97.37_{\pm0.11}$ (✗) |
| | SCP | $17.563_{\pm0.209}$ | $22.787_{\pm0.277}$ | $39.972_{\pm1.300}$ | $89.35_{\pm0.43}$ (✓) | $94.93_{\pm0.27}$ (✓) | $99.02_{\pm0.12}$ (✓) |
| | CRF | $16.946_{\pm0.176}$ | $21.898_{\pm0.271}$ | $38.480_{\pm1.287}$ | $89.44_{\pm0.33}$ (✓) | $94.88_{\pm0.30}$ (✓) | $99.03_{\pm0.12}$ (✓) |
| | CQR | $20.063_{\pm0.092}$ | $24.620_{\pm0.144}$ | $37.035_{\pm0.329}$ | $89.88_{\pm0.27}$ (✓) | $94.90_{\pm0.19}$ (✓) | $99.11_{\pm0.07}$ (✓) |
| | ACP-GN | $15.727_{\pm0.114}$ | $19.810_{\pm0.147}$ | $29.192_{\pm0.179}$ | $89.60_{\pm0.20}$ (✓) | $94.77_{\pm0.14}$ (✓) | $98.85_{\pm0.08}$ (✓) |
| | SCP-GN | $17.373_{\pm0.177}$ | $22.170_{\pm0.233}$ | $34.104_{\pm0.566}$ | $89.47_{\pm0.45}$ (✓) | $94.94_{\pm0.25}$ (✓) | $99.11_{\pm0.12}$ (✓) |
| | ACP-GN (split + refine) | $25.331_{\pm0.182}$ | $32.279_{\pm0.244}$ | $47.494_{\pm0.795}$ | $90.30_{\pm0.40}$ (✓) | $95.45_{\pm0.21}$ (✓) | $99.13_{\pm0.11}$ (✓) |
| wine $N$=1,599 $I$=11 | LA | $2.099_{\pm0.001}$ | $2.501_{\pm0.002}$ | $3.287_{\pm0.002}$ | $91.12_{\pm0.13}$ (✓) | $94.58_{\pm0.10}$ (✓) | $98.34_{\pm0.05}$ (✓) |
| | SCP | $2.112_{\pm0.010}$ | $2.686_{\pm0.018}$ | $4.032_{\pm0.028}$ | $90.11_{\pm0.15}$ (✓) | $95.00_{\pm0.11}$ (✓) | $99.27_{\pm0.05}$ (✓) |
| | CRF | $2.186_{\pm0.013}$ | $2.753_{\pm0.013}$ | $4.397_{\pm0.075}$ | $89.92_{\pm0.15}$ (✓) | $94.92_{\pm0.13}$ (✓) | $99.20_{\pm0.06}$ (✓) |
| | CQR | $1.966_{\pm0.004}$ | $2.421_{\pm0.016}$ | $3.920_{\pm0.031}$ | $89.97_{\pm0.22}$ (✓) | $94.95_{\pm0.09}$ (✓) | $99.12_{\pm0.08}$ (✓) |
| | ACP-GN | $2.103_{\pm0.003}$ | $2.665_{\pm0.006}$ | $3.827_{\pm0.007}$ | $91.16_{\pm0.12}$ (✓) | $95.80_{\pm0.05}$ (✓) | $99.17_{\pm0.04}$ (✓) |
| | SCP-GN | $2.068_{\pm0.009}$ | $2.621_{\pm0.018}$ | $3.760_{\pm0.035}$ | $90.07_{\pm0.18}$ (✓) | $95.00_{\pm0.11}$ (✓) | $99.09_{\pm0.06}$ (✓) |
| | ACP-GN (split + refine) | $2.908_{\pm0.016}$ | $3.527_{\pm0.023}$ | $5.089_{\pm0.037}$ | $90.14_{\pm0.20}$ (✓) | $94.93_{\pm0.21}$ (✓) | $99.09_{\pm0.09}$ (✓) |
| kin8nm $N$=8,192 $I$=8 | LA | $0.213_{\pm0.001}$ | $0.254_{\pm0.001}$ | $0.334_{\pm0.001}$ | $89.66_{\pm0.29}$ (✓) | $93.88_{\pm0.23}$ (✗) | $98.09_{\pm0.10}$ (✗) |
| | SCP | $0.223_{\pm0.001}$ | $0.275_{\pm0.001}$ | $0.394_{\pm0.003}$ | $90.10_{\pm0.27}$ (✓) | $95.35_{\pm0.20}$ (✓) | $99.15_{\pm0.09}$ (✓) |
| | CRF | $0.221_{\pm0.001}$ | $0.271_{\pm0.001}$ | $0.381_{\pm0.003}$ | $90.15_{\pm0.27}$ (✓) | $95.36_{\pm0.19}$ (✓) | $99.12_{\pm0.09}$ (✓) |
| | CQR | $0.241_{\pm0.001}$ | $0.286_{\pm0.001}$ | $0.392_{\pm0.003}$ | $90.41_{\pm0.24}$ (✓) | $95.18_{\pm0.20}$ (✓) | $98.98_{\pm0.09}$ (✓) |
| | ACP-GN | $0.213_{\pm0.001}$ | $0.262_{\pm0.001}$ | $0.365_{\pm0.002}$ | $89.68_{\pm0.28}$ (✓) | $94.58_{\pm0.20}$ (✓) | $98.85_{\pm0.08}$ (✓) |
| | SCP-GN | $0.222_{\pm0.001}$ | $0.273_{\pm0.001}$ | $0.386_{\pm0.003}$ | $90.05_{\pm0.27}$ (✓) | $95.36_{\pm0.20}$ (✓) | $99.12_{\pm0.08}$ (✓) |
| | ACP-GN (split + refine) | $0.243_{\pm0.001}$ | $0.298_{\pm0.001}$ | $0.423_{\pm0.004}$ | $90.60_{\pm0.27}$ (✓) | $95.27_{\pm0.16}$ (✓) | $99.17_{\pm0.09}$ (✓) |
| power $N$=9,568 $I$=4 | LA | $13.345_{\pm0.028}$ | $15.901_{\pm0.034}$ | $20.898_{\pm0.045}$ | $92.08_{\pm0.23}$ (✗) | $95.86_{\pm0.17}$ (✗) | $98.86_{\pm0.09}$ (✓) |
| | SCP | $12.620_{\pm0.051}$ | $15.212_{\pm0.069}$ | $21.720_{\pm0.245}$ | $90.29_{\pm0.22}$ (✓) | $95.01_{\pm0.19}$ (✓) | $98.99_{\pm0.10}$ (✓) |
| | CRF | $12.457_{\pm0.049}$ | $15.019_{\pm0.069}$ | $21.553_{\pm0.189}$ | $90.32_{\pm0.24}$ (✓) | $95.06_{\pm0.17}$ (✓) | $98.96_{\pm0.10}$ (✓) |
| | CQR | $12.818_{\pm0.041}$ | $15.081_{\pm0.058}$ | $21.903_{\pm0.170}$ | $90.52_{\pm0.22}$ (✓) | $94.90_{\pm0.18}$ (✓) | $99.02_{\pm0.08}$ (✓) |
| | ACP-GN | $12.526_{\pm0.024}$ | $15.248_{\pm0.020}$ | $21.592_{\pm0.062}$ | $90.16_{\pm0.26}$ (✓) | $95.13_{\pm0.19}$ (✓) | $98.89_{\pm0.08}$ (✓) |
| | SCP-GN | $12.627_{\pm0.050}$ | $15.215_{\pm0.070}$ | $21.724_{\pm0.242}$ | $90.32_{\pm0.22}$ (✓) | $95.01_{\pm0.18}$ (✓) | $99.00_{\pm0.09}$ (✓) |
| | ACP-GN (split + refine) | $12.834_{\pm0.054}$ | $15.561_{\pm0.067}$ | $22.548_{\pm0.255}$ | $90.18_{\pm0.26}$ (✓) | $95.18_{\pm0.19}$ (✓) | $99.05_{\pm0.10}$ (✓) |
| community $N$=1,994 $I$=100 | LA | $0.548_{\pm0.074}$ | $0.653_{\pm0.088}$ | $0.858_{\pm0.116}$ | $90.90_{\pm0.59}$ (✓) | $93.83_{\pm0.50}$ (✓) | $97.05_{\pm0.33}$ (✗) |
| | SCP | $0.471_{\pm0.009}$ | $0.642_{\pm0.013}$ | $1.033_{\pm0.027}$ | $90.12_{\pm0.55}$ (✓) | $95.20_{\pm0.46}$ (✓) | $98.97_{\pm0.20}$ (✓) |
| | CRF | $0.464_{\pm0.010}$ | $0.617_{\pm0.015}$ | $0.992_{\pm0.029}$ | $90.35_{\pm0.53}$ (✓) | $95.03_{\pm0.45}$ (✓) | $98.90_{\pm0.18}$ (✓) |
| | CQR | $0.659_{\pm0.034}$ | $0.926_{\pm0.058}$ | $1.349_{\pm0.067}$ | $91.28_{\pm0.49}$ (✓) | $96.00_{\pm0.37}$ (✓) | $99.65_{\pm0.18}$ (✓) |
| | ACP-GN | $0.460_{\pm0.002}$ | $0.593_{\pm0.005}$ | $0.932_{\pm0.006}$ | $90.45_{\pm0.46}$ (✓) | $95.05_{\pm0.42}$ (✓) | $99.21_{\pm0.14}$ (✓) |
| | SCP-GN | $0.448_{\pm0.008}$ | $0.621_{\pm0.013}$ | $1.069_{\pm0.048}$ | $90.10_{\pm0.57}$ (✓) | $95.35_{\pm0.42}$ (✓) | $99.03_{\pm0.19}$ (✓) |
| | ACP-GN (split + refine) | $0.561_{\pm0.009}$ | $0.700_{\pm0.010}$ | $1.092_{\pm0.029}$ | $89.97_{\pm0.69}$ (✓) | $94.92_{\pm0.45}$ (✓) | $98.85_{\pm0.20}$ (✓) |
| facebook_1 $N$=40,948 $I$=53 | LA | $67.580_{\pm2.637}$ | $80.527_{\pm3.142}$ | $105.831_{\pm4.130}$ | $97.63_{\pm0.09}$ (✗) | $98.15_{\pm0.08}$ (✗) | $98.76_{\pm0.06}$ (✗) |
| | SCP | $18.756_{\pm0.625}$ | $42.110_{\pm1.381}$ | $176.332_{\pm5.531}$ | $89.98_{\pm0.12}$ (✓) | $95.08_{\pm0.09}$ (✓) | $99.03_{\pm0.04}$ (✓) |
| | CRF | $16.845_{\pm0.513}$ | $34.078_{\pm0.936}$ | $114.769_{\pm4.111}$ | $90.04_{\pm0.13}$ (✓) | $95.12_{\pm0.09}$ (✓) | $98.98_{\pm0.05}$ (✓) |
| | CQR | $21.120_{\pm1.049}$ | $27.024_{\pm1.543}$ | $43.095_{\pm2.232}$ | $90.05_{\pm0.14}$ (✓) | $95.04_{\pm0.08}$ (✓) | $98.95_{\pm0.03}$ (✓) |
| | ACP-GN | $17.986_{\pm0.480}$ | $41.063_{\pm0.770}$ | $199.331_{\pm10.821}$ | $90.36_{\pm0.16}$ (✓) | $95.56_{\pm0.17}$ (✗) | $99.37_{\pm0.08}$ (✗) |
| | SCP-GN | $18.696_{\pm0.612}$ | $39.460_{\pm1.552}$ | $130.967_{\pm6.779}$ | $90.04_{\pm0.13}$ (✓) | $95.04_{\pm0.10}$ (✓) | $99.03_{\pm0.04}$ (✓) |
| | ACP-GN (split + refine) | $24.639_{\pm1.195}$ | $51.864_{\pm2.687}$ | $181.585_{\pm11.155}$ | $90.27_{\pm0.13}$ (✓) | $95.29_{\pm0.09}$ (✓) | $99.14_{\pm0.04}$ (✓) |

Table 11: KFAC approximation to the Gauss-Newton evaluated on large UCI datasets

| | | Avg. Width | | | Avg. Coverage | | |
|---|---|---|---|---|---|---|---|
| | | 90% | 95% | 99% | 90% | 95% | 99% |
| bike $N$=10,886 $I$=18 | **LA** | $100.451_{\pm2.394}$ | $119.694_{\pm2.853}$ | $157.305_{\pm3.749}$ | $89.82_{\pm0.39}$ (✓) | $93.29_{\pm0.33}$ (✗) | $96.83_{\pm0.16}$ (✗) |
| | **LA**(*kfac*) | $91.501_{\pm3.058}$ | $109.030_{\pm3.644}$ | $143.290_{\pm4.789}$ | $87.24_{\pm1.01}$ (✗) | $91.07_{\pm0.84}$ (✗) | $95.20_{\pm0.55}$ (✗) |
| | **SCP** | $131.138_{\pm0.812}$ | $180.477_{\pm1.244}$ | $324.756_{\pm4.635}$ | $90.33_{\pm0.21}$ (✓) | $95.17_{\pm0.15}$ (✓) | $99.00_{\pm0.07}$ (✓) |
| | **SCP-GN** | $122.245_{\pm1.073}$ | $160.505_{\pm1.761}$ | $254.409_{\pm3.767}$ | $90.34_{\pm0.24}$ (✓) | $95.26_{\pm0.15}$ (✓) | $99.02_{\pm0.08}$ (✓) |
| | **SCP-GN**(*kfac*) | $121.401_{\pm0.708}$ | $160.214_{\pm1.328}$ | $259.909_{\pm3.247}$ | $90.46_{\pm0.21}$ (✓) | $95.22_{\pm0.18}$ (✓) | $98.99_{\pm0.07}$ (✓) |
| | **ACP-GN** | $102.401_{\pm6.037}$ | $133.781_{\pm7.394}$ | $212.144_{\pm11.677}$ | $89.27_{\pm0.35}$ (✗) | $94.45_{\pm0.24}$ (✗) | $98.69_{\pm0.10}$ (✗) |
| | **ACP-GN**(*kfac*) | $94.964_{\pm2.095}$ | $124.501_{\pm3.011}$ | $198.018_{\pm5.012}$ | $88.67_{\pm0.68}$ (✗) | $94.06_{\pm0.52}$ (✗) | $98.49_{\pm0.18}$ (✗) |
| | **ACP-GN** (split + refine) | $125.090_{\pm4.180}$ | $162.697_{\pm5.654}$ | $254.893_{\pm9.363}$ | $90.20_{\pm0.22}$ (✓) | $94.95_{\pm0.13}$ (✓) | $99.01_{\pm0.08}$ (✓) |
| | **ACP-GN**(*kfac*) (split + refine) | $162.693_{\pm2.312}$ | $211.953_{\pm2.906}$ | $334.975_{\pm5.205}$ | $91.31_{\pm0.19}$ (✗) | $95.81_{\pm0.11}$ (✗) | $99.24_{\pm0.05}$ (✗) |
| community $N$=1,994 $I$=100 | **LA** | $0.548_{\pm0.074}$ | $0.653_{\pm0.088}$ | $0.858_{\pm0.116}$ | $90.90_{\pm0.59}$ (✓) | $93.83_{\pm0.50}$ (✓) | $97.05_{\pm0.33}$ (✗) |
| | **LA**(*kfac*) | $0.472_{\pm0.003}$ | $0.562_{\pm0.004}$ | $0.739_{\pm0.005}$ | $90.45_{\pm0.42}$ (✓) | $93.55_{\pm0.41}$ (✗) | $96.95_{\pm0.30}$ (✗) |
| | **SCP** | $0.534_{\pm0.010}$ | $0.735_{\pm0.020}$ | $1.164_{\pm0.029}$ | $90.17_{\pm0.41}$ (✓) | $95.33_{\pm0.22}$ (✓) | $99.17_{\pm0.17}$ (✓) |
| | **SCP-GN** | $0.473_{\pm0.013}$ | $0.660_{\pm0.024}$ | $1.116_{\pm0.034}$ | $90.55_{\pm0.39}$ (✓) | $95.10_{\pm0.25}$ (✓) | $99.12_{\pm0.14}$ (✓) |
| | **SCP-GN**(*kfac*) | $0.476_{\pm0.012}$ | $0.657_{\pm0.023}$ | $1.095_{\pm0.030}$ | $90.62_{\pm0.41}$ (✓) | $95.30_{\pm0.23}$ (✓) | $98.90_{\pm0.18}$ (✓) |
| | **ACP-GN** | $0.611_{\pm0.151}$ | $0.784_{\pm0.191}$ | $1.222_{\pm0.289}$ | $90.92_{\pm0.62}$ (✓) | $95.28_{\pm0.45}$ (✓) | $99.25_{\pm0.13}$ (✓) |
| | **ACP-GN**(*kfac*) | $0.476_{\pm0.018}$ | $0.612_{\pm0.023}$ | $0.964_{\pm0.035}$ | $90.78_{\pm0.48}$ (✓) | $95.20_{\pm0.37}$ (✓) | $99.25_{\pm0.12}$ (✓) |
| | **ACP-GN** (split + refine) | $0.523_{\pm0.005}$ | $0.661_{\pm0.007}$ | $1.040_{\pm0.016}$ | $91.38_{\pm0.55}$ (✓) | $95.45_{\pm0.44}$ (✓) | $99.20_{\pm0.14}$ (✓) |
| | **ACP-GN**(*kfac*) (split + refine) | $0.530_{\pm0.005}$ | $0.674_{\pm0.008}$ | $1.067_{\pm0.019}$ | $91.20_{\pm0.64}$ (✓) | $95.62_{\pm0.44}$ (✓) | $99.35_{\pm0.12}$ (✓) |
| protein $N$=45,730 $I$=9 | **LA** | $9.385_{\pm0.022}$ | $11.183_{\pm0.027}$ | $14.697_{\pm0.035}$ | $85.43_{\pm0.18}$ (✗) | $89.69_{\pm0.15}$ (✗) | $94.81_{\pm0.10}$ (✗) |
| | **LA**(*kfac*) | $9.132_{\pm0.027}$ | $10.881_{\pm0.032}$ | $14.301_{\pm0.042}$ | $84.33_{\pm0.18}$ (✗) | $88.63_{\pm0.16}$ (✗) | $94.04_{\pm0.12}$ (✗) |
| | **SCP** | $13.041_{\pm0.088}$ | $17.161_{\pm0.098}$ | $26.181_{\pm0.119}$ | $89.78_{\pm0.08}$ (✓) | $94.83_{\pm0.06}$ (✓) | $98.94_{\pm0.04}$ (✓) |
| | **SCP-GN** | $12.426_{\pm0.085}$ | $16.102_{\pm0.096}$ | $24.032_{\pm0.138}$ | $89.78_{\pm0.10}$ (✓) | $94.86_{\pm0.08}$ (✓) | $98.94_{\pm0.03}$ (✓) |
| | **SCP-GN**(*kfac*) | $12.679_{\pm0.084}$ | $16.570_{\pm0.096}$ | $24.981_{\pm0.133}$ | $89.76_{\pm0.10}$ (✓) | $94.85_{\pm0.08}$ (✓) | $98.94_{\pm0.03}$ (✓) |
| | **ACP-GN** | $10.127_{\pm0.020}$ | $13.156_{\pm0.027}$ | $19.943_{\pm0.061}$ | $87.62_{\pm0.15}$ (✗) | $93.14_{\pm0.10}$ (✗) | $98.35_{\pm0.04}$ (✗) |
| | **ACP-GN**(*kfac*) | $9.695_{\pm0.034}$ | $12.595_{\pm0.041}$ | $19.093_{\pm0.064}$ | $86.02_{\pm0.18}$ (✗) | $91.92_{\pm0.15}$ (✗) | $97.77_{\pm0.06}$ (✗) |
| | **ACP-GN** (split + refine) | $12.614_{\pm0.034}$ | $15.961_{\pm0.044}$ | $23.188_{\pm0.069}$ | $89.88_{\pm0.10}$ (✓) | $94.90_{\pm0.09}$ (✓) | $98.97_{\pm0.05}$ (✓) |
| | **ACP-GN**(*kfac*) (split + refine) | $12.129_{\pm0.105}$ | $15.363_{\pm0.128}$ | $22.260_{\pm0.152}$ | $88.39_{\pm0.14}$ (✗) | $93.82_{\pm0.13}$ (✗) | $98.49_{\pm0.07}$ (✗) |
| facebook_1 $N$=40,948 $I$=53 | **LA** | $67.580_{\pm2.637}$ | $80.527_{\pm3.142}$ | $105.831_{\pm4.130}$ | $97.63_{\pm0.09}$ (✗) | $98.76_{\pm0.06}$ (✗) | $98.76_{\pm0.06}$ (✗) |
| | **LA**(*kfac*) | $67.534_{\pm3.806}$ | $80.471_{\pm4.535}$ | $105.757_{\pm5.960}$ | $97.39_{\pm0.15}$ (✗) | $97.88_{\pm0.12}$ (✗) | $98.57_{\pm0.09}$ (✗) |
| | **SCP** | $20.771_{\pm2.452}$ | $44.192_{\pm2.799}$ | $178.890_{\pm4.596}$ | $90.00_{\pm0.10}$ (✓) | $95.03_{\pm0.08}$ (✓) | $99.05_{\pm0.04}$ (✓) |
| | **SCP-GN** | $20.460_{\pm2.463}$ | $39.712_{\pm3.148}$ | $125.078_{\pm8.075}$ | $90.01_{\pm0.10}$ (✓) | $94.94_{\pm0.09}$ (✓) | $99.04_{\pm0.03}$ (✓) |
| | **SCP-GN**(*kfac*) | $20.303_{\pm2.476}$ | $40.531_{\pm2.973}$ | $128.583_{\pm5.746}$ | $90.01_{\pm0.11}$ (✓) | $95.04_{\pm0.09}$ (✓) | $99.03_{\pm0.03}$ (✓) |
| | **ACP-GN** | $30.466_{\pm7.858}$ | $62.895_{\pm13.942}$ | $223.609_{\pm45.677}$ | $90.56_{\pm0.19}$ (✗) | $95.61_{\pm0.16}$ (✗) | $99.26_{\pm0.05}$ (✗) |
| | **ACP-GN**(*kfac*) | $19.275_{\pm0.542}$ | $40.991_{\pm0.964}$ | $149.319_{\pm5.539}$ | $90.24_{\pm0.12}$ (✓) | $95.38_{\pm0.09}$ (✗) | $99.24_{\pm0.06}$ (✗) |
| | **ACP-GN** (split + refine) | $33.780_{\pm5.321}$ | $57.292_{\pm5.730}$ | $141.074_{\pm6.581}$ | $90.13_{\pm0.11}$ (✓) | $95.23_{\pm0.06}$ (✓) | $99.08_{\pm0.04}$ (✓) |
| | **ACP-GN**(*kfac*) (split + refine) | $23.732_{\pm1.180}$ | $45.239_{\pm2.220}$ | $121.631_{\pm3.699}$ | $89.81_{\pm0.10}$ (✓) | $94.86_{\pm0.08}$ (✓) | $98.90_{\pm0.04}$ (✓) |
| facebook_2 $N$=81,311 $I$=53 | **LA** | $66.088_{\pm2.760}$ | $78.749_{\pm3.289}$ | $103.493_{\pm4.322}$ | $97.47_{\pm0.12}$ (✗) | $98.01_{\pm0.09}$ (✗) | $98.65_{\pm0.06}$ (✗) |
| | **LA**(*kfac*) | $63.591_{\pm3.085}$ | $75.774_{\pm3.675}$ | $99.584_{\pm4.830}$ | $97.26_{\pm0.15}$ (✗) | $97.84_{\pm0.12}$ (✗) | $98.54_{\pm0.08}$ (✗) |
| | **SCP** | $16.387_{\pm0.208}$ | $35.387_{\pm0.462}$ | $152.706_{\pm1.591}$ | $89.97_{\pm0.07}$ (✓) | $95.00_{\pm0.06}$ (✓) | $99.06_{\pm0.03}$ (✓) |
| | **SCP-GN** | $16.287_{\pm0.202}$ | $33.655_{\pm0.563}$ | $118.489_{\pm4.305}$ | $89.99_{\pm0.07}$ (✓) | $94.98_{\pm0.06}$ (✓) | $99.00_{\pm0.03}$ (✓) |
| | **SCP-GN**(*kfac*) | $16.213_{\pm0.195}$ | $33.686_{\pm0.530}$ | $116.672_{\pm2.618}$ | $89.96_{\pm0.08}$ (✓) | $94.98_{\pm0.06}$ (✓) | $99.02_{\pm0.02}$ (✓) |
| | **ACP-GN** | $26.536_{\pm3.513}$ | $56.470_{\pm7.754}$ | $197.251_{\pm27.935}$ | $90.54_{\pm0.11}$ (✗) | $95.46_{\pm0.08}$ (✗) | $99.23_{\pm0.04}$ (✗) |
| | **ACP-GN**(*kfac*) | $20.497_{\pm1.859}$ | $43.246_{\pm3.955}$ | $147.832_{\pm13.223}$ | $90.34_{\pm0.08}$ (✗) | $95.22_{\pm0.05}$ (✗) | $99.18_{\pm0.03}$ (✗) |
| | **ACP-GN** (split + refine) | $23.095_{\pm1.120}$ | $42.653_{\pm1.161}$ | $123.337_{\pm2.954}$ | $90.17_{\pm0.08}$ (✓) | $95.15_{\pm0.05}$ (✓) | $99.08_{\pm0.03}$ (✓) |
| | **ACP-GN**(*kfac*) (split + refine) | $20.487_{\pm0.257}$ | $39.636_{\pm0.582}$ | $117.994_{\pm2.407}$ | $90.11_{\pm0.09}$ (✓) | $95.15_{\pm0.06}$ (✓) | $99.11_{\pm0.03}$ (✓) |

Table 12: Last-layer (LL) approximation to the Gauss-Newton evaluated on large UCI datasets

| | | Avg. Width | | | Avg. Coverage | | |
|---|---|---|---|---|---|---|---|
| | | 90% | 95% | 99% | 90% | 95% | 99% |
| bike $N$=10,886 $I$=18 | **LA** | $100.451_{\pm2.394}$ | $119.694_{\pm2.853}$ | $157.305_{\pm3.749}$ | $89.82_{\pm0.39}$ (✓) | $93.29_{\pm0.33}$ (✗) | $96.83_{\pm0.16}$ (✗) |
| | **LA**$_{(LL)}$ | $79.923_{\pm3.087}$ | $95.234_{\pm3.679}$ | $125.159_{\pm4.835}$ | $82.80_{\pm1.32}$ (✗) | $86.98_{\pm1.21}$ (✗) | $92.19_{\pm0.95}$ (✗) |
| | **SCP** | $131.138_{\pm0.812}$ | $180.477_{\pm1.244}$ | $324.756_{\pm4.635}$ | $90.33_{\pm0.21}$ (✓) | $95.17_{\pm0.15}$ (✓) | $99.00_{\pm0.07}$ (✓) |
| | **SCP-GN** | $122.245_{\pm1.073}$ | $160.505_{\pm1.761}$ | $254.409_{\pm3.767}$ | $90.34_{\pm0.24}$ (✓) | $95.26_{\pm0.15}$ (✓) | $99.02_{\pm0.08}$ (✓) |
| | **SCP-GN**$_{(LL)}$ | $130.965_{\pm0.796}$ | $180.227_{\pm1.213}$ | $324.427_{\pm4.404}$ | $90.26_{\pm0.22}$ (✓) | $95.17_{\pm0.15}$ (✓) | $99.00_{\pm0.07}$ (✓) |
| | **ACP-GN** | $98.813_{\pm2.485}$ | $130.893_{\pm3.231}$ | $213.131_{\pm5.630}$ | $89.36_{\pm0.43}$ (✓) | $94.41_{\pm0.27}$ (✗) | $98.67_{\pm0.09}$ (✗) |
| | **ACP-GN**$_{(LL)}$ | $76.808_{\pm2.933}$ | $100.920_{\pm4.078}$ | $160.448_{\pm6.648}$ | $81.69_{\pm1.31}$ (✗) | $88.20_{\pm1.23}$ (✗) | $95.17_{\pm0.78}$ (✗) |
| | **ACP-GN** (split + refine) | $128.336_{\pm4.336}$ | $170.782_{\pm5.859}$ | $281.632_{\pm10.176}$ | $89.98_{\pm0.22}$ (✓) | $94.94_{\pm0.16}$ (✓) | $99.01_{\pm0.06}$ (✓) |
| | **ACP-GN**$_{(LL)}$ (split + refine) | $128.853_{\pm1.101}$ | $178.079_{\pm1.464}$ | $325.171_{\pm3.916}$ | $90.12_{\pm0.23}$ (✓) | $95.12_{\pm0.20}$ (✓) | $99.02_{\pm0.07}$ (✓) |
| community $N$=1,994 $I$=100 | **LA** | $0.548_{\pm0.074}$ | $0.653_{\pm0.088}$ | $0.858_{\pm0.116}$ | $90.90_{\pm0.59}$ (✓) | $93.83_{\pm0.50}$ (✓) | $97.05_{\pm0.33}$ (✗) |
| | **LA**$_{(LL)}$ | $0.455_{\pm0.013}$ | $0.542_{\pm0.015}$ | $0.712_{\pm0.020}$ | $89.28_{\pm1.00}$ (✓) | $92.33_{\pm0.94}$ (✗) | $96.05_{\pm0.79}$ (✗) |
| | **SCP** | $0.534_{\pm0.010}$ | $0.735_{\pm0.020}$ | $1.164_{\pm0.029}$ | $90.17_{\pm0.41}$ (✓) | $95.33_{\pm0.22}$ (✓) | $99.17_{\pm0.17}$ (✓) |
| | **SCP-GN** | $0.473_{\pm0.013}$ | $0.660_{\pm0.024}$ | $1.116_{\pm0.034}$ | $90.55_{\pm0.39}$ (✓) | $95.10_{\pm0.25}$ (✓) | $99.12_{\pm0.14}$ (✓) |
| | **SCP-GN**$_{(LL)}$ | $0.533_{\pm0.010}$ | $0.730_{\pm0.020}$ | $1.157_{\pm0.028}$ | $90.33_{\pm0.42}$ (✓) | $95.50_{\pm0.24}$ (✓) | $99.15_{\pm0.16}$ (✓) |
| | **ACP-GN** | $0.570_{\pm0.108}$ | $0.755_{\pm0.158}$ | $1.224_{\pm0.285}$ | $90.90_{\pm0.62}$ (✓) | $95.30_{\pm0.47}$ (✓) | $99.25_{\pm0.14}$ (✓) |
| | **ACP-GN**$_{(LL)}$ | $0.438_{\pm0.012}$ | $0.564_{\pm0.016}$ | $0.887_{\pm0.026}$ | $88.83_{\pm1.06}$ (✓) | $93.78_{\pm0.99}$ (✗) | $98.42_{\pm0.65}$ (✗) |
| | **ACP-GN** (split + refine) | $0.519_{\pm0.006}$ | $0.652_{\pm0.008}$ | $1.010_{\pm0.016}$ | $90.97_{\pm0.61}$ (✓) | $95.28_{\pm0.41}$ (✓) | $99.12_{\pm0.16}$ (✓) |
| | **ACP-GN**$_{(LL)}$ (split + refine) | $0.495_{\pm0.005}$ | $0.648_{\pm0.006}$ | $1.040_{\pm0.014}$ | $90.75_{\pm0.40}$ (✓) | $95.70_{\pm0.29}$ (✓) | $99.35_{\pm0.12}$ (✓) |
| protein $N$=45,730 $I$=9 | **LA** | $9.385_{\pm0.022}$ | $11.183_{\pm0.027}$ | $14.697_{\pm0.035}$ | $85.43_{\pm0.18}$ (✗) | $89.69_{\pm0.15}$ (✗) | $94.81_{\pm0.10}$ (✗) |
| | **LA**$_{(LL)}$ | $8.667_{\pm0.026}$ | $10.328_{\pm0.031}$ | $13.573_{\pm0.041}$ | $82.51_{\pm0.17}$ (✗) | $87.08_{\pm0.17}$ (✗) | $92.81_{\pm0.12}$ (✗) |
| | **SCP** | $13.041_{\pm0.088}$ | $17.161_{\pm0.098}$ | $26.181_{\pm0.119}$ | $89.78_{\pm0.08}$ (✓) | $94.83_{\pm0.06}$ (✓) | $98.94_{\pm0.04}$ (✓) |
| | **SCP-GN** | $12.426_{\pm0.085}$ | $16.102_{\pm0.096}$ | $24.032_{\pm0.138}$ | $89.78_{\pm0.10}$ (✓) | $94.86_{\pm0.08}$ (✓) | $98.94_{\pm0.03}$ (✓) |
| | **SCP-GN**$_{(LL)}$ | $13.035_{\pm0.088}$ | $17.150_{\pm0.098}$ | $26.167_{\pm0.119}$ | $89.77_{\pm0.08}$ (✓) | $94.82_{\pm0.06}$ (✓) | $98.94_{\pm0.04}$ (✓) |
| | **ACP-GN** | $10.243_{\pm0.019}$ | $13.294_{\pm0.027}$ | $20.101_{\pm0.053}$ | $87.54_{\pm0.15}$ (✗) | $93.04_{\pm0.11}$ (✗) | $98.24_{\pm0.05}$ (✗) |
| | **ACP-GN**$_{(LL)}$ | $8.731_{\pm0.032}$ | $11.341_{\pm0.038}$ | $17.192_{\pm0.060}$ | $82.62_{\pm0.18}$ (✗) | $89.16_{\pm0.15}$ (✗) | $96.29_{\pm0.08}$ (✗) |
| | **ACP-GN** (split + refine) | $12.660_{\pm0.028}$ | $16.073_{\pm0.031}$ | $23.445_{\pm0.057}$ | $89.83_{\pm0.09}$ (✓) | $94.90_{\pm0.09}$ (✓) | $98.97_{\pm0.05}$ (✓) |
| | **ACP-GN**$_{(LL)}$ (split + refine) | $12.616_{\pm0.029}$ | $16.078_{\pm0.036}$ | $23.672_{\pm0.075}$ | $89.81_{\pm0.11}$ (✓) | $94.77_{\pm0.09}$ (✓) | $98.98_{\pm0.03}$ (✓) |
| facebook_1 $N$=40,948 $I$=53 | **LA** | $67.580_{\pm2.637}$ | $80.527_{\pm3.142}$ | $105.831_{\pm4.130}$ | $97.63_{\pm0.09}$ (✗) | $98.15_{\pm0.08}$ (✗) | $98.76_{\pm0.06}$ (✗) |
| | **LA**$_{(LL)}$ | $66.025_{\pm4.620}$ | $78.673_{\pm5.506}$ | $103.394_{\pm7.235}$ | $96.90_{\pm0.19}$ (✗) | $97.45_{\pm0.17}$ (✗) | $98.10_{\pm0.14}$ (✗) |
| | **SCP** | $20.771_{\pm2.452}$ | $44.192_{\pm2.799}$ | $178.890_{\pm4.596}$ | $90.00_{\pm0.10}$ (✓) | $95.03_{\pm0.08}$ (✓) | $99.05_{\pm0.04}$ (✓) |
| | **SCP-GN** | $20.460_{\pm2.463}$ | $39.712_{\pm3.148}$ | $125.078_{\pm8.075}$ | $90.01_{\pm0.10}$ (✓) | $94.94_{\pm0.09}$ (✓) | $99.04_{\pm0.03}$ (✓) |
| | **SCP-GN**$_{(LL)}$ | $20.735_{\pm2.454}$ | $44.237_{\pm2.780}$ | $177.595_{\pm4.404}$ | $90.02_{\pm0.11}$ (✓) | $95.03_{\pm0.08}$ (✓) | $99.04_{\pm0.03}$ (✓) |
| | **ACP-GN** | $17.986_{\pm0.480}$ | $41.063_{\pm0.770}$ | $199.331_{\pm10.821}$ | $90.36_{\pm0.16}$ (✓) | $95.56_{\pm0.17}$ (✗) | $99.37_{\pm0.08}$ (✗) |
| | **ACP-GN**$_{(LL)}$ | $17.305_{\pm0.514}$ | $36.951_{\pm1.220}$ | $134.876_{\pm6.983}$ | $89.54_{\pm0.12}$ (✗) | $94.53_{\pm0.09}$ (✗) | $98.63_{\pm0.09}$ (✗) |
| | **ACP-GN** (split + refine) | $29.006_{\pm4.472}$ | $54.099_{\pm4.576}$ | $172.252_{\pm6.758}$ | $90.06_{\pm0.10}$ (✓) | $95.20_{\pm0.08}$ (✓) | $99.14_{\pm0.04}$ (✓) |
| | **ACP-GN**$_{(LL)}$ (split + refine) | $19.198_{\pm0.585}$ | $42.501_{\pm0.841}$ | $175.527_{\pm2.964}$ | $90.02_{\pm0.11}$ (✓) | $95.12_{\pm0.08}$ (✓) | $99.06_{\pm0.03}$ (✓) |
| facebook_2 $N$=81,311 $I$=53 | **LA** | $66.088_{\pm2.760}$ | $78.749_{\pm3.289}$ | $103.493_{\pm4.322}$ | $97.47_{\pm0.12}$ (✗) | $98.01_{\pm0.09}$ (✗) | $98.65_{\pm0.06}$ (✗) |
| | **LA**$_{(LL)}$ | $64.783_{\pm4.014}$ | $77.194_{\pm4.783}$ | $101.450_{\pm6.286}$ | $97.02_{\pm0.15}$ (✗) | $97.56_{\pm0.13}$ (✗) | $98.23_{\pm0.09}$ (✗) |
| | **SCP** | $16.387_{\pm0.208}$ | $35.387_{\pm0.462}$ | $152.706_{\pm1.591}$ | $89.97_{\pm0.07}$ (✓) | $95.00_{\pm0.06}$ (✓) | $99.06_{\pm0.03}$ (✓) |
| | **SCP-GN** | $16.287_{\pm0.202}$ | $33.655_{\pm0.563}$ | $118.489_{\pm4.305}$ | $89.99_{\pm0.07}$ (✓) | $94.98_{\pm0.06}$ (✓) | $99.00_{\pm0.03}$ (✓) |
| | **SCP-GN**$_{(LL)}$ | $16.323_{\pm0.199}$ | $35.192_{\pm0.446}$ | $151.199_{\pm1.862}$ | $90.00_{\pm0.07}$ (✓) | $94.99_{\pm0.06}$ (✓) | $99.05_{\pm0.03}$ (✓) |
| | **ACP-GN** | $18.396_{\pm0.546}$ | $40.088_{\pm1.091}$ | $166.792_{\pm5.632}$ | $90.47_{\pm0.11}$ (✗) | $95.45_{\pm0.08}$ (✗) | $99.35_{\pm0.04}$ (✗) |
| | **ACP-GN**$_{(LL)}$ | $17.671_{\pm0.815}$ | $37.040_{\pm1.617}$ | $127.615_{\pm5.783}$ | $89.94_{\pm0.09}$ (✓) | $94.74_{\pm0.06}$ (✗) | $98.73_{\pm0.05}$ (✗) |
| | **ACP-GN** (split + refine) | $21.469_{\pm0.906}$ | $42.184_{\pm0.788}$ | $152.460_{\pm2.499}$ | $90.14_{\pm0.08}$ (✓) | $95.10_{\pm0.06}$ (✓) | $99.13_{\pm0.02}$ (✗) |
| | **ACP-GN**$_{(LL)}$ (split + refine) | $16.895_{\pm0.238}$ | $36.727_{\pm0.477}$ | $149.723_{\pm2.197}$ | $89.98_{\pm0.07}$ (✓) | $95.03_{\pm0.06}$ (✓) | $99.03_{\pm0.02}$ (✓) |

