# OpenReview forum: "Approximating Full Conformal Prediction for Neural Network Regression with Gauss-Newton Influence"
_ICLR.cc/2025/Conference — ICLR 2025 Poster_

### Official Review · Reviewer_EePj · 2024-10-26

**Soundness:** 3
**Presentation:** 3
**Contribution:** 2
**Rating:** 6
**Confidence:** 2

**Summary:**

The paper proposes an efficient conformal prediction method for regression problems based on neural networks. To do so, the paper suggests to use Gauss-Newton influence in order to approximate how the residue of the NN changes, which allows for the CP intervals to be computed without performing NN training in full again.

**Strengths:**

- The paper is well-written, especially in providing enough background for those who may be unfamiliar with the CP methods already.

**Weaknesses:**

- Although I think the paper is already well-written, I feel it would improve on the presentation further if more visualisations are provided, especially in trying to understand how the

- Even though the paper claims that their method should be more efficient than existing naive CP methods for NNs, it would still be interesting to see more results on how this compares. In particular, since the algorithm requires the inverse of a Hessian, it would be interesting to see how the method can scale to larger NNs or to cases with more training data. In particular, some results on running time that compares the proposed methods to other methods would be interesting.

- To also verify the use of Gauss-Newton method, it may also be an interesting demonstration to directly compare the naive method in Algorithm 1 and the approximation in Algorithm 2, to show that even using the approximation the tradeoff in accuracy is not so large but more gained in the running time (I am unsure if this is already shown in the SCP benchmark case already though, in which case the discrepancy in result would be interesting to discuss).

**Questions:**

1. Are there any possible theoretical guarantees for the achieved CP bounds? Given that the Gauss-Newton influence seems like a well-studied theory, and that linearised NNs are also common assumptions in the NTK theory (which itself provides extensive theoretical guarantees for NN predictions), I am wondering if those results can be used to show how accurate the residual estimates would be, and how they would affect the quality of the resulting interval predictions. I am unsure if these theoretical bounds are typical in CP works but it would make the work more theoretically sound.

2. How sensitive are the conformal intervals with respect to the initialisation of the neural network?

3. In the related works section, you have mentioned some more recent methods for conformal prediction on regression problems. Why were these benchmarks unsuitable for comparison in the experiments section as opposed to some of the older methods that were used in the experiments?

---

> ### Author Response · Authors · 2024-11-22
> **Rebuttal by Authors**
>
> Thank you for the positive feedback on our work. We provide below responses and clarifications for the specific questions raised.
>
> **Q1:**
> *Although I think the paper is already well-written, I feel it would improve on the presentation further if more visualisations are provided, especially in trying to understand how the*
>
> **A1:**
> We are more than happy to incorporate suggestions to improve the presentation and clarity of our paper. Can the reviewer elaborate on specific suggestions for the visualization which we will try to incorporate in the camera-ready.
>
> **Q2:**
> *Even though the paper claims that their method should be more efficient than existing naive CP methods for NNs, it would still be interesting to see more results on how this compares.*
> - *In the related works section, you have mentioned some more recent methods for conformal prediction on regression problems. Why were these benchmarks unsuitable for comparison in the experiments section as opposed to some of the older methods that were used in the experiments?*
>
> **A2:**
> See our response to R1 (Q1). Two popular methods (CRF & CQR) from the conformal prediction literature have been added to Table 1 (see revised version). We also compare with ACP in table 6, a recent approximate full-CP approach using influence function. Across all conformal methods, our ACP-GN remains the most efficient in limited data regimes and is overall competitive on larger datasets.
>
> **Q3:**
> *In particular, since the algorithm requires the inverse of a Hessian, it would be interesting to see how the method can scale to larger NNs or to cases with more training data.*
>
> **A3:**
> In appendix H.1 of the revised version, we have added an extensive investigation into scalable approximations, KFAC and last-layer (i.e. neural linear model), evaluated on the same UCI datasets from Table 1. Table 11 and 12 shows their resulting efficiency and coverage. We observe that both approximations frequently give tighter intervals but at the expense of coverage. However in the case of last-layer approximation, when evaluated with our split+refine variant, this always appears to give the correct coverage without any loss to efficiency.
>
> Furthermore, in Sec. 5.2. we have added a new experiment on bounding box localization involving the VGG-19 architecture. This uses the last-layer approximation and its effectiveness is demonstrated in Table 2.
>
> **Q4:**
> *In particular, some results on running time that compares the proposed methods to other methods would be interesting.*
> - *To also verify the use of Gauss-Newton method, it may also be an interesting demonstration to directly compare the naive method in Algorithm 1 and the approximation in Algorithm 2, to show that even using the approximation the tradeoff in accuracy is not so large but more gained in the running time (I am unsure if this is already shown in the SCP benchmark case already though, in which case the discrepancy in result would be interesting to discuss).*
>
> **A4:**
> We have added a running time comparison against Full-CP (naive method in Algorithm 1) and other competing methods in Table 7 in the appendix. For this synthetically-generated dataset, all conformal methods give the correct coverage but the naive method (Full-CP) is indeed the most efficient. However, the running time is several orders of magnitude slower than our method (Algorithm 2). Also see our response to R3 (Q1).
>
> **Q5:**
> *Are there any possible theoretical guarantees for the achieved CP bounds? Given that the Gauss-Newton influence seems like a well-studied theory, and that linearised NNs are also common assumptions in the NTK theory (which itself provides extensive theoretical guarantees for NN predictions), I am wondering if those results can be used to show how accurate the residual estimates would be, and how they would affect the quality of the resulting interval predictions. I am unsure if these theoretical bounds are typical in CP works but it would make the work more theoretically sound.*
>
> **A5:**
> This does seem like an interesting avenue for future work, but we believe is beyond the present scope of this work. It is likely that the consistency results (see Sec. 3.2. of [Martinez et. al., 2023]) can be extended to our case with Gaussian-Newton influence rather than (Hessian-based) influence function. In terms of finite-sample validity, we proposed a variant "ACP-GN (split+refine)" that is guaranteed to provide correct coverage. The residual nonconformity score expressions are *exact* with respect to the linearized neural network and therefore standard conformal guarantees carry over. We will improve the presentation and clarity in the camera-ready to better emphasise this contribution.
>
> [Martinez et. al., 2023] Martinez, J. A., Bhatt, U., Weller, A., & Cherubin, G. (2023, June). Approximating full conformal prediction at scale via influence functions. AAAI.
>
> <1/2>

---

> ### Author Response · Authors · 2024-11-22
> **Rebuttal by Authors**
>
> **Q6:**
> *How sensitive are the conformal intervals with respect to the initialisation of the neural network?*
>
> **A5:**
> All experiments are repeated 20 times with different seeds controlling the initialization of the neural network. All tables report the resulting mean and standard error.
>
> <2/2>

---

### Official Review · Reviewer_pv54 · 2024-10-28

**Soundness:** 3
**Presentation:** 3
**Contribution:** 4
**Rating:** 8
**Confidence:** 4

**Summary:**

The authors highlight in this paper the computational challenges associated with full conformal prediction (FCP), arguing that it is infeasible for practical applications due to needing to train a new instantiation of the model for every data point for every unique label. In an attempt to alleviate this issue and scale FCP to neural network (NN) regression, they attempt to approximate FCP through the use of Gauss-Newton influence and network linearization to form their method ACP-GN and an extension for split CP SCP-GN. The use of Gauss-Newton influence prevents the need to retrain the network as it allows an approximate solution to perturbation of the model parameters, while the use of network linearization prevents the need for an exhaustive grid search over the label space. The evaluation shows that ACP-GN provides a more well-calibrated method in terms of coverage compared to comparable methods in experiments tested.

**Strengths:**

- The proposed methodology to alleviate the issues with FCP is intuitive and logical. It is further backed up with well-written derivations provided in the appendix.
- Sections 1 to 4 are extremely well written: references and literature covered are correct/up-to-date and craft good motivation. Notation throughout is consistent and rigorous.
- The numerous datasets and evaluation splits tested upon are impressive and provide statistically rigorous results.
- Highlighting between Algorithms 1 & 2 is great for readability.
- It is clear originality is high.

**Weaknesses:**

- The main motivation behind the method is that utilising FCP is computationally infeasible. It then feels detached to not comment on the possible/potential computational savings between FCP and ACP-GN. Calculating an approximate training time that FCP would take on an example dataset and comparing it to ACP-GN's training time would be helpful and insightful in Appendix E.
- Realistically, between the handful of proposed methods, the evaluation compares against two methods: a Laplace approximation, and the base split CP. This feels a little weak; I would have preferred to see comparisons against works that use influence functions, or homotopy.
- In the conclusion, the authors state - 'our approximate full-cp methods provide tighter prediction intervals in limited data regimes'. When looking at the small datasets, this statement is true when alpha=(0.1). But when alpha=(0.05 or 0.01), the Laplace method consistently outperforms all proposed variations.

Small weaknesses:
- Captions for tables and figures throughout the paper are used for discussion instead of describing what the table is showing specifically.
- Bolding in Table 1 is potentially misleading. Even though you declare what definition for bolded results. Typically, bolding is used for top-performing results. In Table 1, in numerous cases, bolded results are not best performing. This is a shame as it takes away from the great results reported in the 'coverage' column.

**Questions:**

Can the authors comment on the following:
- The lack of evaluation to previously proposed approximate FCP methods.
- The potentially misleading statement in the conclusion of the paper.
- Why you did not approximate the saved training time between FCP and ACP-GN as computational infeasibility is the main motivation behind the paper.

---

> ### Author Response · Authors · 2024-11-22
> **Rebuttal by Authors**
>
> Thank you for the positive feedback on our work. We provide below responses and clarifications for the specific questions raised.
>
> **Q1:**
> *The main motivation behind the method is that utilising FCP is computationally infeasible. It then feels detached to not comment on the possible/potential computational savings between FCP and ACP-GN. Calculating an approximate training time that FCP would take on an example dataset and comparing it to ACP-GN's training time would be helpful and insightful in Appendix E
> Why you did not approximate the saved training time between FCP and ACP-GN as computational infeasibility is the main motivation behind the paper.*
>
> **A1:**
> We have added a running time comparison to FCP on a synthetic dataset in appendix H.3 in the revised version. The synthetic dataset has 500 train points and 100 test points. We use a uniform grid of 50 points for the target discretization in FCP which is on the lower end as suggested by previous works [Chen et. al., 2018] and enough to give valid coverage. As table 7 shows, we find ACP-GN to be several orders of magnitude faster than FCP.
>
> We have also included a time complexity analysis in App. F of the revised version.
>
> [Chen et. al., 2018] Chen, W., Chun, K. J., & Barber, R. F. (2018). Discretized conformal prediction for efficient distribution‐free inference. Stat.
>
> **Q2:**
> *Realistically, between the handful of proposed methods, the evaluation compares against two methods: a Laplace approximation, and the base split CP. This feels a little weak; I would have preferred to see comparisons against works that use influence functions, or homotopy.*
> - *The lack of evaluation to previously proposed approximate FCP methods.*
>
> **A2:**
> In appendix H.2, we added a comparison against Approximate full Conformal Prediction (ACP) [Martinez et. al., 2023]. Also see our response to R1 (Q1) where we added comparisons against CRF and CQR.
>
> [Martinez et. al., 2023] Martinez, J. A., Bhatt, U., Weller, A., & Cherubin, G. (2023, June). Approximating full conformal prediction at scale via influence functions. AAAI.
>
> **Q3:**
> *In the conclusion, the authors state - 'our approximate full-cp methods provide tighter prediction intervals in limited data regimes'. When looking at the small datasets, this statement is true when alpha=(0.1). But when alpha=(0.05 or 0.01), the Laplace method consistently outperforms all proposed variations.*
> - *Bolding in Table 1 is potentially misleading. Even though you declare what definition for bolded results. Typically, bolding is used for top-performing results. In Table 1, in numerous cases, bolded results are not best performing. This is a shame as it takes away from the great results reported in the 'coverage' column.*
> - *The potentially misleading statement in the conclusion of the paper.*
>
> **A3:**
> We apologise for the potentially misleading conclusion and table 1. The conclusion should have been framed as "our approximate full-cp methods often provide tighter prediction intervals in limited data regimes across well-calibrated methods". We have added coverage checks (ticks/crosses) in the revised version and the table bolding indicates the smallest average width over methods with valid coverage. The table caption has been updated to reflect this. Table 1 now emphasises that Laplace's method miscovers on all datasets at alpha=0.01 and all except boston for alpha=0.05.
>
> **Q4:**
> Captions for tables and figures throughout the paper are used for discussion instead of describing what the table is showing specifically.
>
> **A4:**
> Thank you for pointing this out. We will correct this in the camera ready.
>
> <1/1>

---

> > ### Comment · Reviewer_pv54 · 2024-11-26
> >
> > Dear Authors,
> >
> > Thank you for your detailed response and for adding to the paper as suggested by all the reviewers, not just myself.
> >
> > After considering my previous review and the changes made, I believe this work contributes significantly to the field of uncertainty quantification and the literature on conformal prediction.
> >
> > As a result, I will update my score to reflect this and wish the authors luck with this paper's submission.

---

### Official Review · Reviewer_mcn6 · 2024-10-30

**Soundness:** 2
**Presentation:** 1
**Contribution:** 1
**Rating:** 5
**Confidence:** 5

**Summary:**

This paper proposes to use influence function to approximate full conformal prediction for regression tasks. Like a recent literature, it perturbs the model in the prediction space, and allows for training the model once instead of carrying out the actual costly full conformal prediction procedures.

**Strengths:**

This paper extends a recent work on approximate FCP on classification to regression. FCP is indeed expensive and, if to be applied on large modern datasets with NNs, needs to be made more efficient one way or another.

**Weaknesses:**

1. Inappropriate literature review: This paper didn't cite important (although un-sound, see Appendix C of https://dl.acm.org/doi/abs/10.5555/3540261.3540902) previous research https://proceedings.mlr.press/v119/alaa20a.html despite the very similar idea (and they are also studying regression). Similarly, even though (Martinez et al. 2023) is classification, the current draft severely underplays the clear similarity. While this probably does not constitute a research integrity issue yet, in my opinion this MUST be fixed.

2. Validity of the method: Although I'm very glad that the authors didn't make claims about validity of the approximated FCP method, this paper also lacks an appropriate discussion on the *invalidity* like (Martinez et al. 2023, section title "Validity of ACP"). I highly recommend the authors include a similar section, as in my opinion, the whole point of conformal prediction is about "validity", and approximate CP methods are, to some extent, closed to a "calibrated" non-CP method. Alternatively, I'm hoping to see some "worse case" guarantee when we make additional assumptions about the data distribution. Either way, a discussion on validity is needed.

3. Experiments:
	a. While it is obvious that for very small datasets SCP is very wasteful (due to the sample splitting), I think to use half of the training data as the calibration set is also misleading. Yes, this was what was proposed decades ago, but in practice no one actually reserve half of the data as the calibration set. I suspect with a more appropriate data splitting, we wouldn't see much difference between SCP and ACP on bike.
	b. The lack of validity of ACP is very concerning, as it lacks both theoretical and empirical validity.
	c. I don't see why SCP-GN is related to this paper. It seems the same as "Locally-Weighted Conformal Inference" in (Lei et al., 2018).

**Questions:**

1. Do the author have any thoughts on the validity?

2. Why do we need to approximate FCP in "low-data regime"? Related to this, maybe an actual FCP baseline should be included for yacht, boston and energy.

---

> ### Author Response · Authors · 2024-11-22
> **Rebuttal by Authors**
>
> Thank you for the feedback on our work. Below we provide responses and clarifications for the specific questions raised.
>
> **Q1:**
> *Inappropriate literature review: This paper didn't cite important (although un-sound, see Appendix C of https://dl.acm.org/doi/abs/10.5555/3540261.3540902) previous research https://proceedings.mlr.press/v119/alaa20a.html despite the very similar idea (and they are also studying regression). Similarly, even though (Martinez et al. 2023) is classification, the current draft severely underplays the clear similarity. While this probably does not constitute a research integrity issue yet, in my opinion this MUST be fixed.*
>
> **A1:**
> We will fix the missing reference and better emphasize the relation to [Martinez et al., 2023] in the camera-ready.
> Whilst [Alaa & van der Schaar, 2020] is certainly related work, we would like to push back against the remark that it is a "very similar idea".
> [Alaa & van der Schaar, 2020] approximates jackknife+ which is closely related to cross-conformal methods and has assumption-free coverage guarantee of (1 − 2*alpha) rather than the target level 1-alpha. Our work like [Martinez et al., 2023] approximates the full conformal approach which has coverage at the target level.
>
> In the regression setting our work proposes additional innovations over [Martinez et al., 2023].
> - Firstly, the use of Gauss-Newton influence rather than the more common influence function is a crucial one to enable a grid-free approach and recovers conformal least-squares as a special case (e.g. see Sec. 9.2.1 in https://arxiv.org/abs/2411.11824). To apply [Martinez et al., 2023] to the regression setting requires discretizing the (continuous) target space which introduces an additional computational burden and must be done in a careful way to avoid further increasing the coverage gap [Chen et. al., 2018].
> - Secondly, the extension to studentized nonconformity score was left as future work in [Martinez et al., 2023] which we address.
> - Thirdly, we demonstrate the efficacy of scalable approximations such as Kronecker-factored approximate curvature (KFAC) and showcase the method on deep architectures such as VGG19 in the revised version.
> - Fourthly, we show that our construction of approximate full-CP can be interpreted as conformalizing Linearized Laplace, a popular post-hoc Bayesian Deep Learning method and is therefore relevant to the growing community of Conformal Bayes.
>
> [Chen et. al., 2018] Chen, Wenyu, Kelli‐Jean Chun, and Rina Foygel Barber. "Discretized conformal prediction for efficient distribution‐free inference." Stat 7, no. 1 (2018): e173.
>
> [Alaa & van der Schaar, 2020] Alaa, A., & Van Der Schaar, M. (2020). Discriminative jackknife: Quantifying uncertainty in deep learning via higher-order influence functions. ICML.
>
> **Q2:**
> *Validity of the method: Although I'm very glad that the authors didn't make claims about validity of the approximated FCP method, this paper also lacks an appropriate discussion on the invalidity like (Martinez et al. 2023, section title "Validity of ACP"). I highly recommend the authors include a similar section, as in my opinion, the whole point of conformal prediction is about "validity", and approximate CP methods are, to some extent, closed to a "calibrated" non-CP method. Alternatively, I'm hoping to see some "worse case" guarantee when we make additional assumptions about the data distribution. Either way, a discussion on validity is needed.*
> - *Do the author have any thoughts on the validity?*
>
> **A2:**
> Thank you for the suggestion, we will add a similar section to [Martinez et al., 2023]'s "Validity of ACP". It cannot be assured (nor do we claim) that our ACP-GN satisfies the validity guarantee of full-CP. That being said in a majority of settings (datasets and target coverage levels) we empirically observe that the validity of ACP-GN does hold in practice. To address concerns about validity, we proposed a variant "ACP-GN (split+refine)" that is guaranteed to provide correct coverage. The residual nonconformity score expressions are *exact* with respect to the linearized neural network and therefore standard conformal guarantees carry over. We will improve the presentation and clarity in the camera-ready to better emphasise this contribution.
>
> We will investigate whether the consistency results of ACP (Sec. 3.2. of [Martinez et. al., 2023]) can be extended to our ACP-GN for the camera-ready. It should be possible since the Gauss-Newton influence considered here can be viewed as a specific type of influence function.
>
> <1/2>

---

> ### Author Response · Authors · 2024-11-22
> **Rebuttal by Authors**
>
> **Q3:**
> *While it is obvious that for very small datasets SCP is very wasteful (due to the sample splitting), I think to use half of the training data as the calibration set is also misleading. Yes, this was what was proposed decades ago, but in practice no one actually reserve half of the data as the calibration set. I suspect with a more appropriate data splitting, we wouldn't see much difference between SCP and ACP on bike.*
>
> **A3:**
> In revised version, we repeated all UCI experiments with 25% calibration split -- see Tables 9 and 10. On the small datasets (yacht, boston, energy), ACP-GN remains the most efficient amongst the conformal methods. Furthermore, on bike ACP-GN remains the most efficient amongst conformal methods although the coverage check indicates it miscovers at the 95% and 99% confidence levels. Generally across both settings of the calibration set size we observe our ACP-GN is more efficient than SCP. That being said normalized varieties of split-CP and particularly CQR can close the gap and in some cases be more efficient.
>
> **Q4:**
> *The lack of validity of ACP is very concerning, as it lacks both theoretical and empirical validity.*
>
> **A4:**
> With regards to theoretical validity, we reiterate our response to Q2. We proposed a variant "ACP-GN (split+refine)" that is guaranteed to provide finite-sample validity. The residual nonconformity score expressions are *exact* with respect to the linearized neural network and therefore standard conformal guarantees carry over. For the concern of empirical validity, we have improved the presentation of our tables to show coverage checks (ticks/crosses) in the revised version. A method is reported as satisfying validity if its empirical coverage lies within the 1% and 99% quantiles of the exact marginal coverage distribution as given by the train/calibration set size (depending on the method) (e.g. see Sec. 3.2 in [Angelopoulos & Bates, 2021]). We observe that across the 12 UCI datasets and 3 target coverage levels (see Tables 1 and 6), ACP-GN is empirically valid in a majority of cases (23 out of 36).
>
> [Angelopoulos & Bates, 2021] Angelopoulos, A. N., & Bates, S. (2021). A gentle introduction to conformal prediction and distribution-free uncertainty quantification. arXiv preprint.
>
> **Q5:**
> *I don't see why SCP-GN is related to this paper. It seems the same as "Locally-Weighted Conformal Inference" in (Lei et al., 2018).*
>
> **A5:**
> The "Locally-Weighted Conformal Inference" in (Lei et al., 2018) is conformalized residual fitting (CRF) [Papadopoulos et al., 2002]. As is common with all locally-adaptive/weighted split-CP methods, the residual is normalized by a measure of example difficulty. In the case of CRF, this measure of difficulty is the output of an additional model trained on the absolute residuals. Our SCP-GN does not train an additional model, instead the normalization is computed from the Gauss-Newton Hessian and NN Jacobians of the original model at that point.
>
> Whilst not our main contribution, SCP-GN is relevant to this paper since the normalization can be interpreted as approximating the in-sample residual and recovers as a special case a normalized nonconformity score for least squares proposed in [Vovk et. al., 2005] (see Eq. 4.10 on page 100).
>
> Across all our experiments we have added CRF as a baseline (see revised version). We observe that our SCP-GN improves over CRF for most datasets and settings of the target coverage.
>
> [Papadopoulos et al., 2002] Papadopoulos, H., Proedrou, K., Vovk, V., & Gammerman, A. (2002). Inductive confidence machines for regression. ECML 2002.
>
> [Vovk et. al., 2005] Vovk, V., Gammerman, A., & Shafer, G. (2005). Algorithmic learning in a random world. Springer.
>
> **Q6:**
> *Why do we need to approximate FCP in "low-data regime"? Related to this, maybe an actual FCP baseline should be included for yacht, boston and energy.*
>
> **A6:**
> One could make the case for FCP in low-data regimes in classification settings with small label spaces. However, this quickly becomes prohibitively expensive for larger label spaces, or in a regression setting where a fine grid of candidate targets are needed. In the regression case that we consider this leads to a trade-off between running time and prediction interval efficiency.
>
> We have added a running time comparison to FCP on a synthetic dataset in appendix H.3 in the revised version. We may consider adding a FCP baseline to the small UCI datasets (yacht, boston and energy) in the camera-ready. See also our response to R3 (Q1).
>
> <2/2>
>
> We hope to have sufficiently addressed your comments and concerns. Please let us know if you have any follow-up questions.

---

### Official Review · Reviewer_6yBk · 2024-11-02

**Soundness:** 3
**Presentation:** 3
**Contribution:** 3
**Rating:** 6
**Confidence:** 3

**Summary:**

The authors developed a new and scalable full-CP method considering the Gauss-Newton influence. The paper is well-organized.

**Strengths:**

The theoretical analysis is thorough.

**Weaknesses:**

Some more state-of-the-art algorithms are expected to be adopted for comparison to illustrate the effectiveness.

**Questions:**

1. A complexity analysis of the proposed method should be included to demonstrate its efficiency relative to the standard Full-CP approach.
2. Several state-of-the-art algorithms from the past three years are expected to be compared to further demonstrate the advantages of the proposed method.

---

> ### Author Response · Authors · 2024-11-22
> **Rebuttal by Authors**
>
> Thank you for the positive feedback on our work. We provide below responses for the specific questions raised.
>
> **Q1:**
> *Some more state-of-the-art algorithms are expected to be adopted for comparison to illustrate the effectiveness.
> Several state-of-the-art algorithms from the past three years are expected to be compared to further demonstrate the advantages of the proposed method.*
>
> **A1:**
> We have added comparisons against two popular baselines from the conformal prediction literature: conformalized residual fitting (CRF) [Papadopoulos et. al., 2002] [Johansson et. al., 2014] [Lei et. al., 2018] and conformalized quantile regression (CQR) [Romano et. al., 2019] -- see table 1 in the newest revision.
> We have also added a comparison to Approximate full Conformal Prediction (ACP) [Martinez et. al., 2023] in the appendix (see table 6 in the newest revision).
> Across all conformal methods, our ACP-GN remains the most effective (in terms of prediction interval efficiency) in limited data regimes and is overall competitive on larger datasets.
>
> [Papadopoulos et. al., 2002] Papadopoulos, H., Proedrou, K., Vovk, V., & Gammerman, A. (2002). Inductive confidence machines for regression. ECML 2002.
>
> [Lei et. al., 2018] Lei, J., G’Sell, M., Rinaldo, A., Tibshirani, R. J., & Wasserman, L. (2018). Distribution-free predictive inference for regression. Journal of the American Statistical Association.
>
> [Johansson et. al., 2014] Johansson, U., Boström, H., Löfström, T., & Linusson, H. (2014). Regression conformal prediction with random forests. Machine learning.
>
> [Martinez et. al., 2023] Martinez, J. A., Bhatt, U., Weller, A., & Cherubin, G. (2023, June). Approximating full conformal prediction at scale via influence functions. AAAI.
>
> **Q2:**
> *A complexity analysis of the proposed method should be included to demonstrate its efficiency relative to the standard Full-CP approach.*
>
> **A2:**
> We have added a time complexity analysis in App. F of the revised version. Unlike FCP, much of the cost in our ACP-GN can be amortized, that is computation can be re-used when constructing prediction intervals for new batches of test points. This complexity however is cubic in the number of parameters but we show in Table 4 that this can be reduced considerably when certain approximations to the Gauss-Newton Hessian are used. In contrast, the complexity of FCP is multiplicative in the number of test points, grid points, train points (twice) and epochs whereas for ACP-GN, it is just multiplicative in the number of test points given the upfront cost. See also our response to R3 (Q1) on the running time comparison between ACP-GN and FCP.
>
> <1/1>

---

### Author Response · Authors · 2024-11-26
**Author response**

We thank reviewers for their thoughtful feedback for which we have provided responses. We hope to have addressed your concerns satisfactorily and if so, we would appreciate if you could consider increasing your score. To reiterate, in the revised version we have:

- included additional baselines: conformalized residual fitting (CRF) and conformalized quantile regression (CQR) *[R1,R4]*
- added a time complexity analysis against full-CP and ACP in addition to a running time comparison on a synthetic dataset *[R1,R2,R3,R4]*
- repeated experiments with a 25% calibration split in the case of split CP methods *[R2]*
- added coverage checks to all tables to address concerns of empirical validity *[R2,R3]*
- added ablations using Gauss-Newton Hessian approximations such as KFAC in addition to a new experiment on bounding box localization using a VGG19 backbone *[R4]*

---

### Public Comment · ~Valery_Manokhin1 · 2025-04-21
**Code**

Where is the code?

---

> ### Public Comment · ~Dharmesh_Tailor1 · 2025-04-24
> **Code repository**
>
> The code has been released here: https://github.com/Qualcomm-AI-research/newton-influence-conformal/

---

### Meta-Review · Area_Chair_yzG7 · 2024-12-10

**Metareview:**

The main computational challenge in evaluating full conformal prediction lies in the requirement to retrain the model for each test point and candidate label. This work addresses the issue by proposing a more efficient approach: training the model only once and approximating the effects of retraining through local perturbation of its parameters using Gauss-Newton influence. Additionally, through linearisation of the network, they represent the absolute residual nonconformity score as a piecewise linear function of the candidate label. This formulation enables an efficient procedure that avoids exhaustive searches over the output space.

The paper addresses an important problem, is well-written, presents interesting ideas, and includes numerous experiments. While some concerns have been raised regarding the degree of novelty and connections to prior work, most reviewers agree that the paper offers a meaningful contribution to the field.

**Additional Comments On Reviewer Discussion:**

Some concerns were initially raised regarding the connections of the present work to relevant prior research and the lack of discussion surrounding the issue of "validity." However, the authors have since addressed many of these issues.

---

### Decision · Program_Chairs · 2025-01-22

Accept (Poster)